# Population discontinuity in the Paris Basin linked to evidence of the Neolithic decline

At the transition between the third and the fourth millennium BC, there is evidence for a population decline concurrent with the end of megalith building across continental northwestern Europe. In Scandinavia this 'Neolithic decline' is followed by a massive population turnover, as farming communities disappeared and were replaced by people with steppe ancestry. In western Europe, however, ancestry associated with Neolithic farmers persisted beyond the Neolithic decline, and it remains unclear whether a similar demographic replacement occurred. To investigate the population dynamics around the Neolithic decline in present-day France, we sequenced 132 ancient genomes from the *allée sépulcrale* at Bury. Located in the Paris area, Bury spans two burial phases separated by a hiatus with no burial activity: one phase directly preceding the Neolithic decline in the late fourth millennium BC, ending around 3000 BC, and a later phase some time after the Neolithic decline in the early- to mid-third millennium BC. Our analysis revealed that the two burial phases at Bury represented largely discontinuous genetic groups of a markedly different social organization as inferred from three large pedigrees. We show that the difference between the two burial phases can be linked to a northwards movement of Neolithic ancestry from the south, which only spread into the Paris Basin after the Neolithic decline, at around 2900 BC. Together with genetic evidence of various infectious diseases in the dataset, such as *Yersinia pestis* and *Borrelia recurrentis*, as well as evidence for forest regrowth between the two phases, these findings detail a population turnover at the end of the fourth millennium BC, offering a possible explanation for the cessation of megalith building.

The construction of complex megalithic tombs, a hallmark feature of the Neolithic time period, ceased across continental northwestern Europe at the end of the fourth millennium BC for unknown reasons[1]. Radiocarbon dates indicate a period of construction for these collective tombs around 4300–3100 BC, followed by a decline of burial activities in general between 3000 and 2600 BC, depending on the area[2–4]. These tombs are numerous throughout northwestern Europe, with very high concentrations in places such as the Paris Basin, central Germany and southern Scandinavia. They are collective burials that accommodated deceased as they died, and thus collected tens of thousands of dead

in the second half of the fourth millennium BC[5–7]. Constructions vary from so-called passage graves made from large boulders mainly found in northwestern Europe and southern Scandinavia, to long cist graves (gallery graves) made of stone slabs more common in the Paris Basin (*allées sépulcrales*) and central Germany (*Galeriegräber*), which were used by different archaeological groups: Seine-Oise for the *allées sépulcrales*[8], Wartberg for the *Galeriegräber*[9], Bernburg for the *Totenhutten*[10] and the Funnelbeaker–Trichterbecker cultural complex for the Scandinavian passage graves[11]. The large-scale decline in the construction of these megaliths towards the end of the fourth millennium BC, could,

✉e-mail: ewillerslev@sund.ku.dk; ew482@cam.ac.uk; martin.sikora@sund.ku.dk

**Fig. 1 | Overview of the Bury grave. a**, Location of Bury and similar sites with genetic data available, with the geographical extent of the Paris Basin highlighted. **b**, Schematic overview of the Bury grave during Phase 1. p.slab, porthole slab. **c**, Schematic overview of the Bury grave during Phase 2 (adapted from ref. 12, De Gruyter). Basemap data from Natural Earth (https://www.naturalearthdata.com/).

in principle, reflect either a shift in cultural behaviour or a demographic decline. However, demographic analyses[12] and recent genetic results[13,14] in combination with data from distributions of radiocarbon dates[3,15] have provided increasing support for the latter hypothesis. One of the theories put forward to explain this so-called Neolithic decline is that environmental exploitation brought about by farming, such as soil degradation and deforestation, reduced the land's capacity to support agriculture and livestock and, thus, its ability to support local populations[15]. Others argue that the close contact between humans and animals in the Neolithic increased the risk of pathogen emergence, which, together with the increased population density, increased the risk of transmission[16,17].

Located 50 km north of Paris, the Neolithic burial site of Bury is from a region where many other collective graves have been recorded[4]. As a semi-underground monument of rectangular shape, Bury is a classic example of the *allées sépulcrales* found northwest of Paris, built with a combination of megalithic slabs and other techniques, like drystone walls. The Bury grave held primary burials of 316 individuals, divided in two main burial phases[18]. The first phase was used over a relatively short period at the end of the fourth millennium (around 3200–3100 BC)[19], and represents burials in extended body positions oriented by the main axis of the grave[18] (Fig. 1b). Phase 2, on the other hand, covered several centuries over the third millennium until 2470 BC, and is characterized by flexed body positions with no preferred orientation[18] (Fig. 1c). The demographic profile of the 180 individuals buried during the first phase does not correspond with the normal age–mortality pattern expected in such a population[12] (Supplementary Note 1). Rather, the demographic profile is suggestive of excess mortality, particularly affecting juvenile individuals, perhaps indicating a catastrophic event, such as war, famine or a disease outbreak or, on the contrary, a rapid increase in the population. Phase 2, on the other hand, shows no indication of elevated mortality.

Here we present a population-scale study of this megalithic site, integrating analyses of burial practices, social structure and kinship, ancestry and ancient pathogen genomics to determine the importance of infectious diseases in these mortality patterns, and their wider implications in the Neolithic decline on a broad geographical scale across northern and western Europe.

## Results

We sampled the cementum layer of 182 teeth excavated from Bury (Supplementary Table 1) and sequenced these to a depth of coverage ranging from ×0.001 to ×4.6 (median ×0.126) with 132 genomes above ×0.01, which we term sufficient coverage for further analysis (Supplementary Table 2). The remains were categorized into two chronological phases based upon the stratigraphy and the location of the remains, burial behaviours and the radiocarbon program made in parallel with the excavation[18] (Fig. 1 and Supplementary Table 3). These results were then completed with data from the European Research Council-funded advanced grant 'The times of their lives'. Phase assignments were later confirmed and refined, in particular for isolated mandibles, using the results of genetic kinship analysis. With this combined procedure, we assigned 74 and 51 samples with coverage over ×0.01 to Phases 1 and 2, respectively, while 7 samples could not be assigned to a phase (Fig. 2b).

Genetic sexing confirmed the high predominance of males in both phases[20] (Fig. 2b and Supplementary Note 1), similar to reports from other Neolithic sites from present-day France and Germany[21–24]. Of the 131 individuals where genetic sex could be reliably determined, 71% and 73% males were observed in Phases 1 and 2, respectively. This imbalance concerns both adult and non-adult individuals, and is incompatible with a natural population, thus suggesting differential burial treatment between males and females at Bury—for some reason, more than half of the females in the community were excluded from being buried in the grave.

Mitochondrial DNA haplogroup analysis revealed a diverse set of maternal lineages over both phases (Supplementary Table 2), whereas Y chromosome haplogroups showed lower diversity among males, but with distinct patterns between phases (Supplementary Table 4 and Extended Data Fig. 1). While men from Phase 1 generally carried haplogroups H2a1 or I2a1a2, men from Phase 2 carried haplogroup I2a1a1 almost exclusively.

## Genetic discontinuity between Phases 1 and 2

To determine genetic ancestry of the individuals at Bury, we analysed them within the context of a reference panel of ~4,700 previously published ancient genomes (Supplementary Table 5). From our principal component analysis (PCA), we found that all individuals from Bury fell within the broader diversity of 'Neolithic farmers' from western Eurasia (Fig. 2c). Furthermore, we note that the Phase 1 individuals generally appear more diverse, with a wider distribution on the PCA, as opposed to the Phase 2 individuals, which are more concentrated

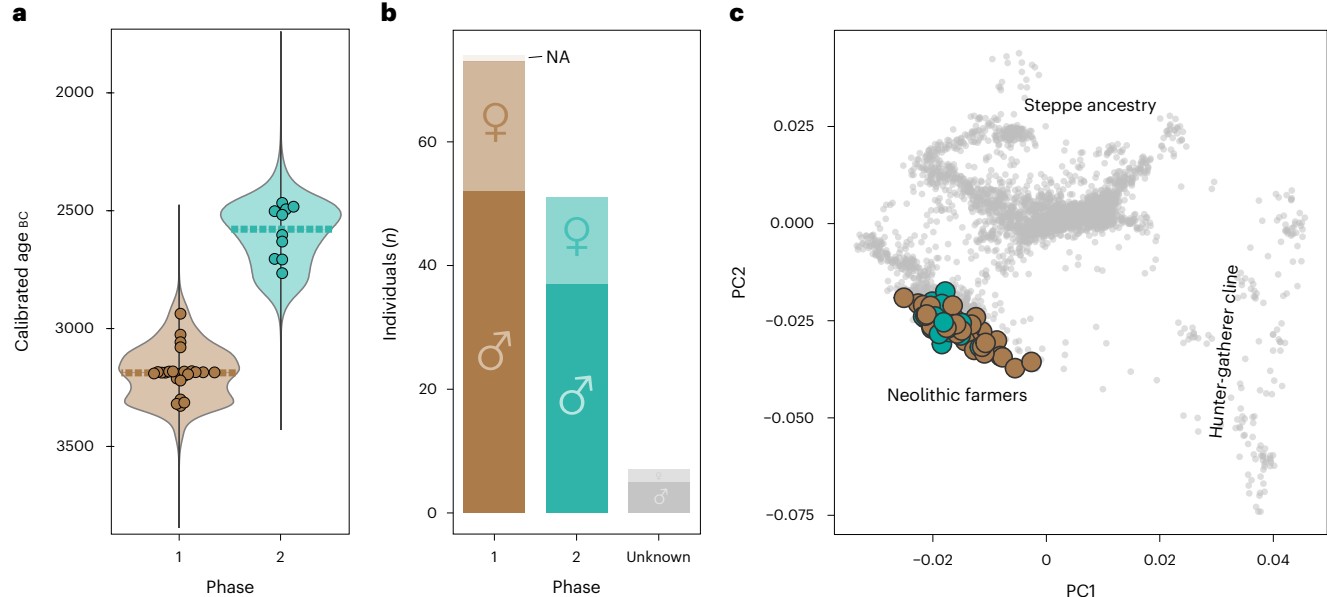

**Fig. 2 | Genetic results from Bury. a**, Kernel density calibration curves for all radiocarbon-dated individuals included in the genetic program in Phase 1 (brown) and Phase 2 (teal). The stippled line represents the median of the group. **b**, Number of males and females buried in the different use phases with coverage >×0.01. NA represents individuals where sex could not be reliably determined. **c**, PCA showing the genetic ancestry of the samples from Bury with coverage >×0.01 (in colour) in relation to published west Eurasian ancient genomes (grey).

on the plot (Extended Data Fig. 2). The higher diversity during Phase 1 is also reflected by the proportion of hunter-gatherer ancestry, which is elevated in a handful of samples from Phase 1, resulting in a slight shift towards the hunter-gatherer cline of the PCA (Fig. 2c and Extended Data Fig. 3a,b).

To further investigate how the populations of Phases 1 and 2 in Bury relate to each other and to other Neolithic groups of Europe, we analysed identity-by-descent (IBD) segments shared between pairs of individuals. Using hierarchical graph clustering on the resulting IBD-sharing network between individuals, we found that the individuals from Bury generally clustered based on their phase. While individuals from Phase 1 cluster on their own or with individuals from the Paris Basin and western Germany, the Phase 2 individuals display strong ties to southern France and Iberia (Supplementary Fig. 2.3).

Next, we investigated total pairwise IBD sharing between the Bury populations and all other clusters to determine how genetically different the two populations at Bury were (Supplementary Fig. 2.4). We found that the two Bury populations did not share the most genetic segments with each other; instead, each phase appeared to have more total IBD sharing with several other Neolithic groups. For the population in Phase 1, 12 other groups share a higher mean total IBD length than the Phase 2 population. For Phase 2, ten groups are more similar to that population than the Phase 1 population. We also estimated the effective population sizes for each phase projected back in time using this IBD data (Extended Data Fig. 4c and Supplementary Note 2.4). From this analysis, we found that the two phases followed separate population size trajectories, and that the effective population size of Phase 1 was markedly smaller than that of Phase 2 at the point of sampling. Importantly, these estimates also revealed a very recent population size contraction in Phase 1. Taken together, these IBD-based findings provide further support for considering the individuals from the two phases as separate populations.

Having established that the two phases at Bury form two separate populations, we decided to explore whether these findings are compatible with either genetic continuity or discontinuity between the two phases (Methods). To do this, we simulated the following two different scenarios: discontinuity (blue; Extended Data Fig. 4a,b) and continuity (pink; Extended Data Fig. 4a,b). We compared patterns of IBD sharing between the simulated populations and the actual data, and found no overlap between the real data and the continuity scenario. Accordingly, these simulations rule out the scenario of complete continuity between the two populations at Bury. However, the simulation does not exclude the possibility of some amount of gene flow between the two burial phases.

## Neolithic ancestry from the south could explain the shift at Bury

To put our findings from Bury into context with other Neolithic people from western Europe, we modelled the proportion of various ancestries in all individuals from Bronze Age and Neolithic Europe (Methods and Supplementary Tables 6 and 7). In agreement with our PCA analysis, we found high diversity in the Phase 1 individuals with varying proportions of modelled ancestries from Early Neolithic France and from a group of Neolithic Iberians dated to the fourth millennium BC (Supplementary Note 2.1 and Extended Data Fig. 5). This pattern is reflected in other contemporaneous individuals from the Paris Basin, from the sites Mont Aimé hypogee (I + II)[25], Wettolsheim[26] and Pont-sur-Seine[26]. For Phase 2, on the other hand, the modelling revealed a more homogenous population, with over 80% (mean 83.8% ± 0.1% s.d.) ancestry from Iberia.

When visualizing major ancestry groups on a map (Fig. 3 and Extended Data Fig. 4), the mixture modelling reveals a stepwise northwards spread of this Neolithic Iberian ancestry (Supplementary Note 2.1). By 2900 BC, populations across southern France and Iberia all constituted a large fraction of Iberian ancestry, while people in the Paris Basin still comprised mixed ancestry proportion, as represented by the Phase 1 individuals. At some point after 2900 BC, a final northwards push of the Iberian ancestry partially replaces the existing local ancestry in the Paris Basin, resulting in the homogenous population we observe in Phase 2. After the end of Phase 2, around 2500 BC, people with steppe ancestry first appear in the Paris Basin[27], where they mix with the local population to form the genetic profile typically associated with Bell Beakers (Fig. 3). As such, these results readily explain the difference between the populations of Phases 1 and 2, and could suggest an event that facilitated the northwards expansion of Neolithic Iberian ancestry at around 2900 BC.

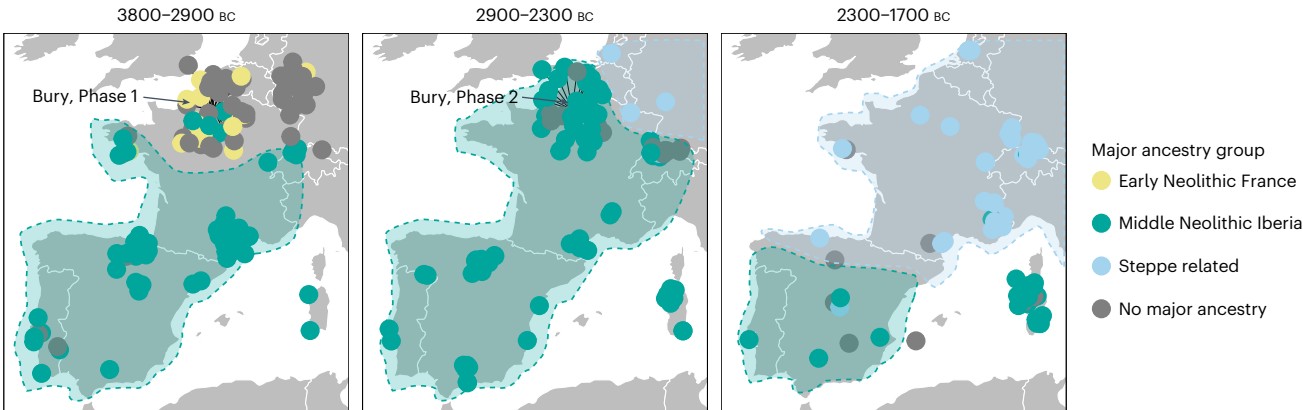

**Fig. 3 | The spread of Neolithic ancestry from Iberia.** Map of genomes from western Europe coloured by the major modelled ancestry group in each individual, split by time period. Grey ('no major ancestry') represents individuals where no ancestry group makes up over 60% of the total ancestry. Shaded areas represent our interpretation of the geographical spread of steppe ancestry and Neolithic Iberians. Basemap data from Natural Earth (https://www.naturalearthdata.com/).

The scenario outlined above represents our interpretation of this mixture modelling data. An alternative explanation could be that the few individuals who already had high proportions of 'Middle Neolithic Iberia' DNA in Phase 1 proliferated and came to dominate the population in Phase 2. However, if the descendants of a few Phase 1 individuals had come to dominate in Phase 2, we would expect to see a strong bottleneck in Phase 2, more similarity in the population size trajectories for both phases (Extended Data Fig. 4c) and simulation results indicative of higher levels of population continuity. As none of these patterns are present in our data, we find this explanation less likely. Instead, we view the individuals with Iberian ancestry in Phase 1 as early arrivals originating from outside the local region. This interpretation is supported by the observation that half of the unrelated Phase 1 individuals carried high proportions of 'Middle Neolithic Iberia' ancestry, while only three individuals within the pedigrees exhibited this genetic profile.

## A shift in societal structure inferred by genetic links

To examine differences in the social organization between the two phases at Bury, we investigated genetic relatedness with NgsRelate[28,29] and KIN[30] (Supplementary Table 8). We found that three quarters of individuals in Phase 1 (55 out of 72 (76%); Fig. 4c) have at least one first- or second-degree relative also buried at the site, whereas Phase 2 displays a larger fraction of unrelated individuals (only 21 related among 53 individuals (40%); Fig. 4c). Furthermore, we found no trace of relatedness between individuals across the two phases. To investigate whether there were differences in societal structure between the two phases, we combined these results with Y chromosome and mitochondrial haplogroups to reconstruct a total of 14 pedigrees. From each phase, seven pedigrees could be reconstructed varying in size, from large networks spanning up to five generations to small pedigrees consisting of only two individuals (Fig. 4 and Extended Data Figs. 6 and 7).

In the first phase of the burial, the sampled material is dominated by large biological groups spanning several generations, with several cases of three or four full siblings (for example, siblings sharing both parents, as opposed to half siblings) or their offspring being buried in the grave (Fig. 4a and Extended Data Fig. 6). We term the seven groups in Phase 1 '1.A–1.G'. We observe no cases where males from outside the family line entered it by having children with its members. Aside from the founding generation of each pedigree, all subsequent males are genetically descended from earlier members. On the other hand, all females except one are related exclusively to their offspring, suggesting a high level of female exogamy for the females buried in the tomb. Interestingly, the male sex bias observed within families is not as pronounced for unrelated individuals. This observation aligns well with reports from sites with high levels of female exogamy, where unrelated individuals tend to be dominated by females, perhaps representing females that never produced offspring or whose offspring were buried outside of the tomb[13].

Pedigree 1.A is the largest of the Phase 1 pedigrees (Fig. 4a and Extended Data Fig. 6a), containing 29 sequenced individuals and a further 19 inferred individuals who were either not sampled or buried outside the grave. All males in this group have Y chromosome haplogroup H2a1. Three brothers (BUR222, BUR174 and BUR343) form the first generation. Of these, BUR343 does not have offspring that were buried in the grave, while BUR174 only has a single granddaughter. However, BUR222 fathers four children (three sons, two of which sequenced, and one daughter), and all of these children have children, grandchildren and, in one case, great-grandchildren who were buried within the grave.

The two sequenced sons of BUR222, BUR291 and BUR275, both show long runs of homozygosity, with similar length profiles as expected from parents that are first cousins (Extended Data Fig. 8a), suggesting that BUR222 and the unsampled mother were third-degree relatives. One of these sons, BUR291, has a further son (unsampled), who conceived three children with one female, BUR262, who is a third- or fourth-degree relative to BUR291, her reproductive partner's biological father. The only sequenced son of this union (BUR316) also exhibits long runs-of-homozygosity segments. Incidentally, BUR262 is also the only female buried in the grave who has both ancestors and children also buried at Bury.

One of the three brothers, BUR174, in the first generation of Phase 1 is also of some note. He was found as an almost complete skeleton, in a seated position in the northeastern corner of the grave. The skull had three perimortem lesions, each of which was caused by a blow with a heavy and sharp instrument, theorized to be an axe[19]. Around BUR174, three children and one perinatal individual were deposited in a seemingly deliberate pattern. One of these children, BUR189, was successfully sequenced and found to be an avuncular relation to the three brothers. As they share the same mitochondrial haplogroup, it is likely that BUR189 is the offspring of a sister to the three-brother group. It is not known if the other children in the burial arrangement are relatives, but given the high predominance of related individuals in Phase 1, it is likely that these individuals were related to others. This arrangement was previously hypothesized to be part of a 'founding act' of the monument because of the position and characteristics of BUR174 in the corner of the rear end of the grave[19] (Fig. 1a). Given our genetic results and BUR174's position in the first generation of pedigree group 1.A, this hypothesis seems likely.

The other large pedigree in the first phase, pedigree 1.B, spans four generations, with all individuals descending from a 'founder'

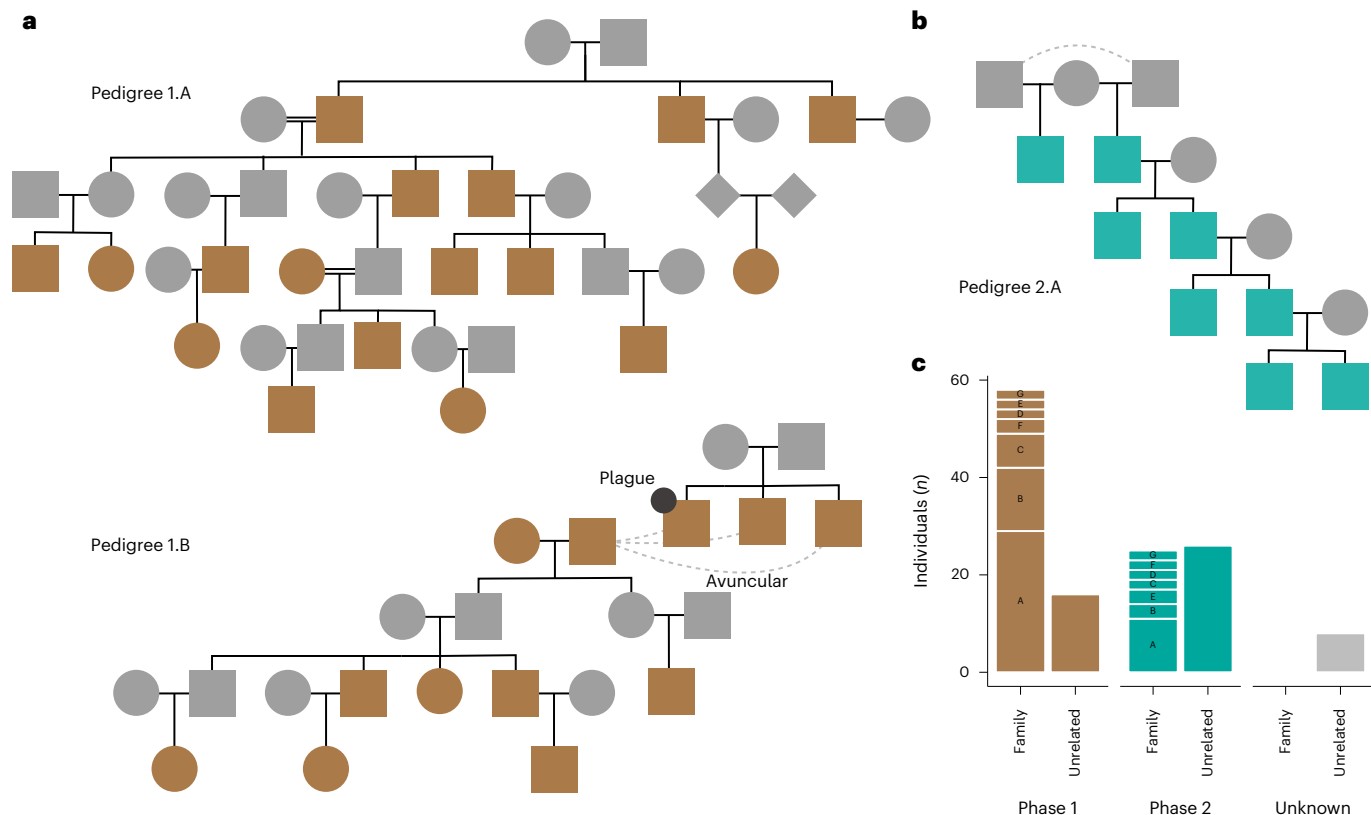

**Fig. 4 | Familial relations in Phases 1 and 2. a**, Pedigree 1.A and 1.B from Phase 1. Double black lines indicate mating between related individuals. The stippled lines in **a** and **b** represent unknown second degree relationships (avuncular: relationship between uncle/aunt and nephew/niece). **b**, Pedigree 2.A from Phase 2. **c**, Bar plot depicting the number of individuals within each pedigree group compared to unrelated individuals, stratified into phases. For visual clarity, each pedigree group is abbreviated with a single letter (A, B, C and so on), where, for example, 'A' in Phase 1 corresponds to 'pedigree 1.A', while 'B' in Phase 2 represents 'pedigree 2.B', and so on (Supplementary Note 2.2, Extended Data Figs. 6 and 7 and Supplementary Table 2).

reproductive couple, BUR257 and BUR266. Their children, an unsampled female and male, went on to have a total of at least five children and three grandchildren. All of these individuals but one son were included in our analysis. The remaining son was either not buried at Bury or not sampled. One of the children, the only offspring of the unsampled daughter, BUR185, is a juvenile male who shares the same chromosome Y haplogroup H2a1 as pedigree group 1.A. This is remarkable, as the rest of pedigree group 1.B males have the I2a1 haplogroup. The second phase lacks the large pedigrees that dominated the first phase (Fig. 4c and Extended Data Fig. 7). There are seven distinct pedigrees, which we term pedigree 2.A–2.G, the largest of which is pedigree group 2.A. Pedigree group 2.A spans four generations in a patrilineal line, with each of the four generations constituting exactly two brothers (or half-brothers), only one of which in turn has offspring buried in the grave. This consistent pattern is suggestive of a hereditary network.

There are only two females in the second phase with first-degree relatives—one mother of a son buried in the grave (pedigree 2.F) and one female with a full brother (pedigree 2.C). On the one hand, as the second phase lacks the wide pedigrees that characterize the first phase, it seems as though the burials were more selective. However, the higher fraction of totally unrelated individuals could speak to the monument following a different kind of organization with a much higher degree of non-biological relatives included in the grave. Altogether, the differences in the pedigrees and the distribution of related and unrelated individuals between the two phases of Bury imply that the grave was used differently through time—a finding which is corroborated by demographic analysis[12]; in the first phase it was used for wider families and in the second phase it was used for smaller patrilineal lines and unrelated individuals.

There is no correlation between burial position within the grave and genetic relationship, except in the founding deposit (Supplementary Note 7). Without comparable studies of similar collective burials, it is difficult to interpret such a result. However, it confirms the community character of the tomb. Slightly older, the two chambers of the long barrow of Hazleton North[31], with fewer individuals, nevertheless show that the internal distribution of the individuals is not guided by genetic links; conversely, these links decide the distribution of individuals in either chamber. The predominance of a large genetic group during the first phase, probably related to the other small groups, suggests a strong association between this group and the collective grave, or to the links that could exist between the inhabitants of the same place. Phase 2, on the other hand, is characterized by a single paternal family line, and a handful of small pedigrees scattered throughout the different spaces of the tomb. As this phase is also represented by a substantially higher proportion of unrelated individuals, this could imply that social or cultural ties were more important during this time. It should also be noted that the longer duration of Phase 2, compared to Phase 1 (ref. [19]), may have contributed to the different selection of the dead.

## Evidence for the earliest diverging lineage of *Yersinia pestis* at Bury

While no published record on the general health of the populations in Bury exist, an analysis of the oral-dental area did not find evidence of poor conditions[20]. To investigate if this pattern could be corroborated by genetic data, we screened all samples for pathogen DNA (Methods). In total, we detected four different microbes that most likely stem from a disease in the host: *Yersinia enterolitica* (n = 8), *Yersinia pestis* (n = 4), *Borrelia recurrentis* (n = 2) and *Human alphaherpesvirus 1* (that

is, Herpes simplex virus 1, $n = 1$; Supplementary Table 9). Of these four pathogens, only *Yersinia pestis*, the aetiological agent of plague[32,33], displays an uneven distribution between the two burial phases: three individuals are from Phase 1, whereas only one individual with plague is detected in Phase 2. The plague-positive individuals do not appear to be related; one male comes from pedigree 1.E (Extended Data Fig. 6e), another male comes from pedigree 1.B (Fig. 4a), while one female from Phase 1 and a male from Phase 2 are both unrelated (Extended Data Figs. 6h and 7h). Of the other pathogens detected, *Yersinia enterocolitica*, the cause of yersiniosis, and *Borrelia recurrentis*, the causative agent of louse-borne relapsing fever, are both noteworthy. While the mortality of yersiniosis today is low, almost a third of cases need hospitalization. Furthermore, the mortality of untreated louse-borne relapsing fever ranges from 15% to 40%, and has been associated with poor hygiene and, in historic times, with catastrophic events such as war or famine.

The coverage of *Yersinia pestis* in the four plague-positive individuals is generally low. With a coverage of ×0.43, BUR218 is our best sample, while the remaining samples are significantly lower (×0.010, ×0.019 and ×0.002; Supplementary Table 10). Using phylogenetic placement analyses, we found that the three plague-positive samples over ×0.01 were placed in the pre-Late Neolithic/Bronze Age (preLNBA) cluster (Extended Data Fig. 9). Furthermore, due to the higher coverage of our best sample, we were able to place it in the phylogeny with higher confidence. We found that the plague form from BUR218 (3339–3042 BC) was placed at the branch basal to the genome RV2039[34] (3350–3100 BC; Fig. 5b). This finding suggests that the plague form from Phase 1 of Bury is similar, and perhaps slightly older, than that of the RV2039 genome isolated from a hunter-gatherer in present-day Latvia.

Given that we find three plague cases in Phase 1 (3 out of 74, 4% of Phase 1 samples with coverage ≥×0.01) and only a single case in Phase 2 (1 out of 51, 2% of Phase 2 samples with coverage ≥×0.01), one could hypothesize that an outbreak of the plague caused the end of Phase 1, leading to the cessation of burials. However, two hallmark features that would be expected for a deadly plague outbreak are missing at Bury: (1) at 4% the plague prevalence in Phase 1 is still relatively low (at the Swedish site Frälsegården, for example, a prevalence of 28% was reported[13]) and (2) the plague-positive cases do not appear to be distributed in the last generations of the pedigree, although they are among the last deposits of Phase 1 in the anterior part of the grave. However, the grave only represents a subset of the population and it is perfectly possible that a severe plague epidemic would leave very little evidence behind, if the entire population perished without being buried or if they were buried differently, presumably elsewhere, as was the case for many plague victims during the second pandemic[35,36].

### Environmental data on the Neolithic decline

Lastly, we investigated pollen data from the Paris Basin to assess if the population collapse observed between Phase 1 and Phase 2 could be linked to any vegetational changes. Of the seven Neolithic temporal windows analysed by David et al.[37], the interval from 2900 to 2500 BC shows evidence of forest regeneration, which is typically linked to a decrease in human activity (Supplementary Note 3). A similar pattern was observed, both in Scania, Sweden (Supplementary Note 3), where forest regrowth reached a climax around 3100 BC, and in Zealand, Denmark, showing a climax between 3000 and 2800 BC (Supplementary Note 4). Similar results have previously been documented in northern Germany[38], and in central Europe where the decline period could be dated to between 3300 and 2950 BC[39] based on a combination of summed probability distributions of radiocarbon dates and palaeoecological proxies. As such, these observations from a number of well-documented regions can be interpreted as resulting from abandonment of grazing lands and fields, implying settlements were given up. Accordingly, they describe a significant decline in human activity, and are in agreement with similar observations after the Justinian plague[40] and the Black Death[41].

## Discussion

The data from the Bury grave provides important new evidence regarding both the 'Neolithic decline' and the general population dynamics in the fourth and third millennium BC. Since the beginning of the twentieth century, many authors have argued for an influx from Iberia to northwestern Europe according to the Bell Beaker pottery diffusion during the third millennium BC (as reported in ref. 42; Fig. 3). Our data demonstrate an influx of genetic ancestry from Iberia before the Bell Beaker phenomenon[43]: we show that the builders and first users of the tomb were largely replaced by a Neolithic population coming from southern France and Iberia as early as 2900 BC. While this date precedes the first confirmed Bell Beaker influx in the Paris Basin by several hundred years[44], it fits well with both the timing of the Neolithic decline (3100–2900 BC)[1] and the first archaeological influence from the south of France in the region at around 2800 BC[18]. Importantly, this Middle Neolithic spread of Iberian ancestry does not explain the previously established link between Early Neolithic farmers from Iberia and the British Isles[45], as individuals from modern day Britain and Ireland are modelled almost exclusively as 'Early Neolithic France' (Extended Data Fig. 5). Instead, the arrival of the first farmers to the British Isles may reflect an earlier migration from mainland Europe.

As opposed to Scandinavia where steppe-related groups replaced the local farmers 4,700 years ago[46], the population dynamics in the Paris Basin are characterized by the replacement (or partial replacement) with another group of Neolithic people, who persisted for around 500 years until the arrival of steppe-related populations. Our modelling results demonstrate that these groups mixed with the local 'Bury-Phase-2-like' population, as genomes from 2300–1700 BC are modelled with varying proportions of 'Iberian Neolithic' and 'steppe' ancestries (Extended Data Fig. 5). Furthermore, our strontium isotope data depict a more stable and sedentary population in Bury than earlier Neolithic groups from the area (Supplementary Note 6 and Supplementary Table 11). Tied in with our findings on the full or partial replacement of the population between Phase 1 and Phase 2, this suggests that newcomers did not immediately reuse the tomb. Still, we do observe some mobility, with a total of 14 strontium outliers identified (six from Phase 1, and eight from Phase 2), most of which are either unrelated individuals or exogamous females.

When we compare Bury with Hazelton[31] in England, a double-chambered megalithic burial place, we note both similarities and differences. Our findings of the community-like social structure of the buried population in Phase 1, with little or no in-marriage from individuals outside the Bury region, correspond in several aspects with the community-like organization also in Hazelton, as well as western Sweden[13]. This pattern is mirrored in earlier Middle Neolithic populations from present-day France, at sites such as Gurgy[21] and Fleury-sur-Orne[23], which exhibit similar characteristics. Phase 2, on the other hand, stands out with its longer duration, hereditary family structure and high number of unrelated individuals. The clear differences between the phases at Bury suggest that a different population reused a tomb built by the people that came before them. A similar reuse was observed at the site Frälsegården from Sweden[13] where there is evidence for four individuals with steppe DNA interred in the tomb, in addition to the 47 Neolithic farmers who presumably built the grave. Such observations warrant questions about how the people of the second phase of Bury regarded the original builders of the tomb from Phase 1. Could a small amount of continuity between the phases exist, which justified the appropriation of the grave in Phase 2? How did the transmission of the burial place occur? These reflections highlight broader uncertainties about how ancestry and ownership were understood in these societies.

Given the temporal hiatus between the two Bury phases, the genetic evidence suggesting a large amount of discontinuity between the phases and the evidence of reforestation (Supplementary Note 3), we can tie this into a wider European picture to trace potential signals of demographic decline in the Late Neolithic[47]. The same hiatus in burial

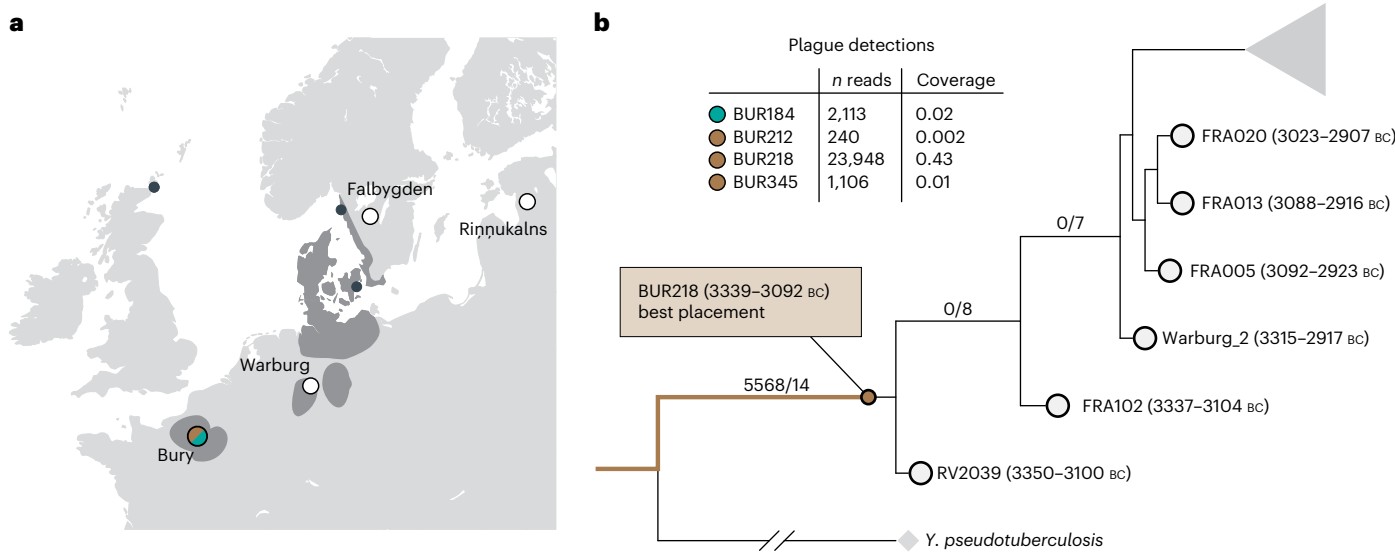

**Fig. 5 | Phylogenetic placement of the Bury plague. a**, Map of pre-Late Neolithic/Bronze Age plague cases. Areas in dark grey correspond to the main concentrations of collective graves in the second half of the fourth millennium, with bones preserved. **b**, Phylogenetic placement of the highest coverage plague sample (BUR218, depth of coverage: ×0.43). Basemap data from Natural Earth (https://www.naturalearthdata.com/).

sequence is known in similar graves from Germany[48,49]. There is also evidence from megalithic tombs in northern Germany, which ended around 3100–3000 BC, followed by decreasing open land[11,12,50,51]. Additionally, we present evidence of precise construction dates for Danish passage graves ending around 3000 BC (Supplementary Note 5). When we add the anomalies in the demographic structure of the buried population in Phase 1 of Bury, as well as other megaliths from northern France and Germany (Supplementary Note 1), there are further indications of demographic change, supporting a more widespread phenomenon covering most of continental northwestern Europe. The construction of collective burials corresponded to a demographic increase and high population density among Neolithic societies in northwest Europe[12], which stopped around 3100 BC, as exemplified in Bury. After a couple of centuries of interruption, the use of the Bury grave during the whole third millennium is different, both in ritual and population structure, as in most of northwest Europe. It should be stressed, however, that the Neolithic decline is not synchronous across Europe, even if it falls within a timespan from roughly the late fourth to the beginning of the third millennium BC. Within this time span there are temporal and geographical variations[52], and we should expect that a gradual reduction of populations due to both pathogens and other environmental factors may have contributed by making Neolithic populations more vulnerable[47,52].

We may thus consider the possibility that both the Iberian northward migration and the expansion from the steppe were related responses to the Neolithic decline, as widespread demographic contraction would have created a vacuum that neighbouring groups could expand into.

## Methods
### Sampling strategy
The sampling was made in 2013 under the supervision of L. Salanova, the scientific authority of the Bury project. In total, we collected 181 ancient human teeth samples, representing 179 archaeological individuals (57% of the total of 316 individuals estimated to have been buried at the site). We decided to sample mandibular teeth, as the skulls could not be linked to the identified archaeological individuals (Supplementary Table 1). In total, we sampled more than 71% of the testable mandibles, including all but one of the mandibles from identified archaeological individuals. Lastly, we also sampled 21 loose teeth to ensure the broadest representation possible. Along with all other samples, these loose teeth were subsequently screened genetically to identify any duplicates (see 'Kinship analysis and pedigree reconstruction').

### Library preparation and sequencing
The processes of sample drilling, DNA extraction and library preparation were carefully performed in the dedicated ancient DNA clean facility at the Globe Institute, University of Copenhagen, following strict ancient DNA protocols[14,53]. Both manual and automated techniques were applied. The DNA extracted was then used to create double-stranded, blunt-end sequencing libraries[54]. To confirm the authenticity of the ancient DNA, initial libraries were prepared without uracil-specific excision reagent (USER) treatment, enabling the detection of post-mortem damage. When feasible, uracil-specific excision reagent (USER)-treated libraries were later constructed using the NEB M5505L enzyme (New England Biolabs) to remove deaminated cytosines and reduce the influence of post-mortem damage on subsequent analyses[55], particularly during deeper sequencing. These libraries were sequenced on either the Illumina HiSeq 2500 for single-end reads or the NovaSeq 6000 for paired-end reads.

### Basic bioinformatics
The sequencing data were demultiplexed using the Illumina software BCL Convert, allowing for one mismatch in the index. Adaptor sequences were trimmed and overlapping reads were collapsed (using the '--collapse-conservatively' option) using AdapterRemoval (2.3.2)[56]. Reads below 30 base pairs were discarded. Paired-end and collapsed reads were mapped to the human reference genome build 38 using bwa software (0.7.17)[57] with seeding enabled (-l 32). Paired- and single-end reads for each library and lane were merged, and duplicates were marked using Picard MarkDuplicates (2.25.0) with a pixel distance of 12,000. Mitochondrial haplogroups were called using Haplogrep 2 (v.2.1.25) on variants identified using mutserve (v.1.3.0). Chromosome Y haplogroups, on the other hand, were called using an in-house script detailed in Supplementary Note 2 of ref. 13.

### DNA authentication
To determine the authenticity of the ancient reads, post-mortem DNA damage patterns were quantified using mapDamage2.0[58]. Next, two different methods were used to estimate the levels of contamination.

Firstly, we applied ContamMix to quantify the fraction of exogenous reads in the mitochondrial reads by comparing the mitochondrial DNA consensus genome to possible contaminant genomes[59]. For each library, an in-house perl script was used to construct two different versions of the endogenous mitochondrial genome. The first approach, aimed at low-coverage data, used sites with at least ×1 coverage, and at each position a base was only called if it was observed in at least 50% of reads covering the site. The second approach, for high-coverage libraries, only considered sites with at least ×5 coverage and 70% of reads agreeing. Lastly, we applied Analysis of Next Generation Sequencing Data (v.0.931)[60] to estimate nuclear contamination by quantifying heterozygosity on the X chromosome in males. Both contamination estimates only used filtered reads with a base quality of ≥20 and mapping quality of ≥30. We retained libraries with at least 95% authentic mitochondrial reads (maximum a posteriori estimate of contamination), and with <2% X chromosome contamination in the case of genetic males. Finally, in cases where we had more than one library per individual, we merged them using SAMtools[61].

## Human population genetics

For basic human population genetic analysis, we called pseudohaploid genotypes by randomly selecting alleles at all positions in the reference genome (hg38) and merging this dataset with a panel of 4,748 imputed ancient human genomes (Supplementary Table 5). We characterized genetic diversity using the PCA implemented in PLINK[62] and the Smart-PCA[63] approach from the Eigensoft tool[64], focusing on a subset of 2,011,119 transversion-only single-nucleotide polymorphisms with a minor allele frequency over 0.1%. For the SmartPCA plot in Fig. 2c, a sub-panel of 3,779 ancient west Eurasians was used (panel: 'Eur' in Supplementary Table 5), with samples from this study projected onto the reference panel diversity. Furthermore, to enhance genotype accuracy for high-coverage samples, we imputed genotypes for samples with over ×0.1 coverage. Imputation was carried out on all genomes generated for this study with GLIMPSE[65] using the phased data from the 1000 Genomes Project[66] dataset as reference. To investigate fine-scale genetic structures within each population, we merged these imputed genomes with our imputed reference panel (Supplementary Table 5) and filtered genotypes, retaining only sites with a minor allele frequency over 1% and imputation quality (INFO score) over 0.5. We used IBDseq (r.1206)[67] to characterize regions shared IBD between all pairs of individuals with default settings. We chose IBDseq for calling IBD segments because of its advantage in detecting shorter segments, which are integral for our downstream clustering and mixture modelling analysis (refer to Supplementary Note 5.3.5 in ref. 40).

To filter out closely related individuals, we converted all pairs of individuals sharing more than 800 cM IBD into a graph object and calculated the maximal set of independent vertices using the max_ivs function from the R package igraph. For each graph, we calculated maximal sets of unrelated individuals and retained the set with the highest number of individuals. In cases where multiple sets were of equal size, we retained the set with the highest mean coverage. Lastly, using this dataset of unrelated individuals, we removed short IBD segments (<2 cM) and clustered genomes of similar ancestry using hierarchical clustering on a Euclidean distance matrix of total IBD-sharing vectors between all pairs of individuals. Final cluster memberships were extracted using dynamic branch cutting implemented in the DynamicTreeCut[68] package in R, with a cluster height cut-off of 4,000.

## Simulations

To investigate whether continuity or discontinuity between the two phases at Bury best explained our data, we used msprime[69] to simulate each scenario. Specifically, we simulated two populations (A and B) that shared a last common ancestor 1,000 years ago, assuming a constant population size of both populations of 6,000 (based on IBDNe estimates). We sampled population A in the present, representing Phase 2 under a discontinuity scenario, while population B was sampled in the present, representing Phase 2 under continuity, and 400 years ago, representing Phase 1 (as shown in the following table). To reflect the actual data as closely as possible, we sampled the same number of individuals (>×0.1 coverage) as identified in each phase: 52 individuals in Phase 1 and 42 individuals in Phase 2.

| Simulated population | Sampling time | Phase, scenario | *n* samples |
|---|---|---|---|
| popA | Present | Phase 2, discontinuity | 42 |
| popB | Present | Phase 2, continuity | 42 |
| popB | 400 years ago | Phase 1 | 52 |

Furthermore, to simulate erroneous base calls arising as a result of the fragmented, damaged and low-coverage nature of ancient DNA, we estimated the frequency of incorrectly called alleles using real data. We subsampled an Iron Age sample (CGG023681[40], average coverage ×19.2) down to various coverages and counted the frequency of incorrectly called single-nucleotide polymorphisms at each coverage. From this analysis we inferred a relationship between error rate and coverage, which we used to infer the estimated error rates for all bury samples, given their coverage. Using this approach, we estimated that the mean error rate for all Bury samples is 0.00473. Next, we added error to the simulated data at a rate of 0.00473 using the R script vcfErr.R (https://github.com/spTallman/vcfErr). Lastly, we called segments of IBD shared between pairs of simulated individuals using IBDseq[67], and calculated basic IBD-sharing statistics between Phase1, and Phase2_discontinuity and Phase2_continuity, respectively.

## Mixture modelling

Raw IBD coordinates from IBDseq were first converted into centiMorgans using the genetic map for build 38 of the human genome (https://storage.googleapis.com/broad-alkesgroup-public/Eagle/downloads/tables/genetic_map_hg38_withX.txt.gz). Next, we removed IBD tracts with excess sharing across all individuals and filtered out segments shorter than 2 cM, as described in ref. 70. We modelled all individuals from this study as mixtures of eight source groups (Supplementary Note 2.1) using non-negative least squares, implemented in the mixmodel_ibd.R script (mixmodel_ibd) and plotted the resulting ancestry proportions in an admixture style plot using the script plot_mixmodel.R.

## Kinship analysis and pedigree reconstruction

To investigate close relatives among the Bury individuals, we employed five different methods to detect close relatives: (1) kinship-based inference for genome-wide association studies[71], (2) NgsRelate[28,29], (3) Relationship Estimation from Ancient DNA version 2 (READ2)[72], (4) KIN[73] and (5) the relationship inference algorithm based on pairwise IBS sharing presented in ref. 74. For methods (1)–(3) we used imputed genotypes as input data, whereas for methods (4) and (5) we used mapped reads as the starting point for kinship inference. Using this combination of algorithms, we manually inferred the most likely relationship between all related pairs of individuals and reconstructed the pedigree using a combination of the automated tool PRIMUS[75] and the manual triangulation approach described in ref. 31. We found a very high level of agreement between all approaches for high-coverage samples when comparing the performance from each of the five kinship tools (Supplementary Figs. 2.7 and 2.8). However, for lower coverage samples, we observe greater variability in the results. We generally found that KIN, NgsRelate2 and READ2 produced more consistent results than the other two approaches; therefore, we placed the highest confidence in predictions derived from these tools. In cases where these three tools did not agree, we generally followed the prediction from KIN, unless

contradicted by other data, such as mitochondrial haplogroups, sex or relatedness to other individuals (Supplementary Table 8).

We acknowledge that these analyses only reflect organization amongst the dead, not the living. However, we also note that the two are tightly linked and one is very likely to have implications for the other. Furthermore, we use the term 'pedigree', along with kinship terms such as 'mother', 'daughter', 'full brother' and 'half-sister', to describe biological relatedness, and do not imply that these terms, nor the notion of family, was understood by the population using the grave at Bury during the Neolithic times in the same way as it is understood in Western societies today.

## Pathogen screening
To screen sequencing data for human pathogens, we characterized microbial diversity using a previously described approach[76]. Briefly, we pre-processed DNA reads using KrakenUniq[77] against a custom database of full microbial genomes. Based on this screening, we identified genera present in each sample. Next, we analysed reads belonging to each identified genus individually using traditional Bowtie2 mapping. Using this approach, we leverage the efficiency of the $k$-mer-based classification employed in KrakenUniq, while retaining the sensitivity of traditional mapping. For each alignment, we then calculated standard mapping statistics such as depth of coverage, breadth of coverage, duplication rate, read mapping quality, average nucleotide identity and coverage evenness (actual breadth of coverage/expected breadth of coverage given the depth of coverage). Lastly, to characterize the best hit within each genus, we ranked each alignment based on the number of unique $k$-mers identified in each sample. We characterize a positive microbial detection as alignments with over 50 reads that have the highest number of unique $k$-mers within their genus, an average number of soft clippings under 8, an average nucleotide identity of over 0.97 and a coverage evenness score of over 0.8 (Supplementary Table 9). This approach ensures that all closely related species within each genus are compared with competitive mapping after the initial KrakenUniq step. A microbial hit is only considered a true positive detection when a given microbial species is detected with higher similarity than all of its close relatives.

## Plague phylogenetics
We analysed the four plague-positive samples identified in the pathogen screening using the workflow described in ref. 13. Briefly, we mapped all pre-processed reads from each plague-positive sample against the plague reference genome (CO92; GCA_000009065.1). Next, duplicate reads or reads mapping to the reference with low confidence (mapping quality < 30) were removed using MarkDuplicates from Picard tools and SAMtools, respectively. We used bedtools genomecov (v2.31.0)[78] to calculate the average depth of coverage for each sample. For the three samples with a depth of coverage over 0.01 we carried out phylogenetic placements using EPA-ng[79] based on major allele genotypes, which were called by selecting the most frequent allele at each variant position from a reference panel. Subsequently, placements were visualized in R using a combination of the ggtree package[80] and the treeio package[81].

## Reporting summary
Further information on research design is available in the Nature Portfolio Reporting Summary linked to this article.

## Data availability
Fastq files with collapsed and trimmed reads from this study have been deposited in the European Nucleotide Archive under accession number PRJEB95770 (Supplementary Tables 12 and 13). Furthermore, human (hg38) bam alignment files and imputed VCF genotype data files are available from https://doi.org/10.17894/ucph.2d60146b-f518-4b79-8896-8f10c3037684.

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

## Acknowledgements

The Bury excavation was made possible thanks to the support of the Ministère de la Culture (service régional de l'Archéologie de Picardie), the CNRS, the Bury council and private owners of the field. We thank A. Bayliss and A. Whittle, European Research Council advanced grant 'The times of their lives' (2012–2017), for the radiocarbon dating of BUR184. We acknowledge the support of the GeoGenetics Sequencing Core (University of Copenhagen), and L.D.K. Hansen, M. Madrona, A.B. Pørksen and M.B. Hjort for their technical assistance. E.W. thanks St John's College, Cambridge, for providing a stimulating environment of discussion and learning. This study was supported by The Swedish Foundation for Humanities and Social Sciences (Riksbankens Jubileumsfond, grant no. M16-0455, to K.K., K.-G.S. and T.D.P.) and the project 'COREX: from correlations to explanations: towards a new European prehistory' funded by the European Research Council under the European Union's Horizon 2020 research and innovation programme (grant agreement no. 95138) to K.K. and R.F. Furthermore, the Lundbeck Foundation GeoGenetics Centre is supported by grants from the Lundbeck Foundation (R302-2018-2155, R155-2013-16338 and R491-2024-1351), the Novo Nordisk Foundation (NNF18SA0035006 and NNF25SA0103965), the Wellcome Trust (214300), the Carlsberg Foundation (CF18-0024), the Danish National Research Foundation (DNRF94, DNRF174), the University of Copenhagen (KU2016 programme) and Ferring Pharmaceuticals A/S (to A.R., F.V.S., J.C., F.D., C.G., L.V., J.S., H.M., G.S., T.S.K., M.E.A., K.K., E.W., M.S.). F.V.S was supported by the Lundbeck Foundation (grant no. R322-2019-2610). T.S.K was supported by Carlsberg Foundation Young Researcher Fellowship awarded by the Carlsberg Foundation in 2019. M.S. was supported by the Riksbankens Jubileumsfond grant M 21-0018 for the six-year programme MARITIME ENCOUNTERS: a counterpoint to the dominant terrestrial narrative of European prehistory.

## Author contributions

L.S., K.K., E.W. and M.E.A. conceptualized this study. F.V.S., A.R., J.C., C.G., L.V., J.S., H.M., G.S., R.F., M.F.M., S.K., K.V., T.S.K., L.S., P.C. and M.E.A generated the data. A.R., J.C. and F.D. curated the data. Plans of the tomb were created by P.C. and L.S. Formal analysis was carried out by F.V.S., A.R., J.C., R.F., T.D.P., M.F.M., S.K., T.D. and S.I.H. M.E.A., K.K. and E.W. acquired funding for the project. L.S. and P.C. provided resources for the study. Original draft preparation was carried out by F.V.S., A.R., L.S., K.K., J.C., P.C., E.W. and M.S. Manuscript review and editing were carried out by F.V.S., A.R., J.C., K.-G.S., L.V., G.S., K.V., M.E.A., K.K., E.W., M.S., L.S. and P.C. F.V.S., P.C., L.S., R.F., T.D.P., M.F.M., S.K., T.D. and S.I.H. wrote the Supplementary Information. Supervision was provided by L.S., T.S.K., K.K., E.W. and M.S.

## Competing interests

The authors declare no competing interests.

## Additional information

**Extended data** is available for this paper at https://doi.org/10.1038/s41559-026-03027-z.

**Correspondence and requests for materials** should be addressed to Eske Willerslev or Martin Sikora.

Frederik V. Seersholm [1,16], Abigail Ramsøe[1,16], Jialu Cao[1,16], Philippe Chambon[2], Karl-Göran Sjögren [3], Hugh McColl [1,3], Fabrice Demeter [1,4], Charleen Gaunitz[1], Lasse Vinner[1], Jesper Stenderup[1], Gabriele Scorrano [1,5], Ralph Fyfe [6], T. Douglas Price [7], Morten Fischer Mortensen[8], Sascha Krüger[8], Torben Dehn[9], Svend Illum Hansen[9], Kristine Vesterdorf [10], Thorfinn Sand Korneliussen [1], Morten E. Allentoft [1,11], Kristian Kristiansen[1,3], Laure Salanova[12], Eske Willerslev [1,13,14,15,17] ✉ & Martin Sikora [1,17] ✉

[1]Section for GeoGenetics, Globe Institute, University of Copenhagen, Copenhagen, Denmark. [2]CNRS-UMR 7206, Musée de l'Homme, Paris, France. [3]Department of Historical Studies, University of Gothenburg, Gothenburg, Sweden. [4]Eco-anthropologie (EA), Dpt ABBA, Muséum National d'Histoire Naturelle, CNRS, Université Paris Cité, Musée de l'Homme, Paris, France. [5]Center for Molecular Anthropology for the Study of Ancient DNA, Department of Biology, University of Rome Tor Vergata, Rome, Italy. [6]School of Geography, Earth and Environmental Sciences, University of Plymouth, Plymouth, UK. [7]Laboratory for Archaeological Chemistry, Department of Anthropology, University of Wisconsin–Madison, Madison, WI, USA. [8]The National Museum of Denmark, Environmental Archaeology and Materials Science, I.C. Modewegs Vej, Kongens Lyngby, Denmark. [9]Dehn Archaeological Consulting, Copenhagen, Denmark. [10]School of Human Sciences, University of Western Australia, Perth, Western Australia, Australia. [11]Trace and Environmental DNA (TrEnD) Laboratory, School of Molecular and Life Sciences, Curtin University, Perth, Western Australia, Australia. [12]CNRS, Paris, France. [13]Geogenetics Centre for Ancient Environmental Genomics, Globe Institute, University of Copenhagen, Copenhagen, Denmark. [14]GeoGenetics Group, Department of Genetics, University of Cambridge, Cambridge, UK. [15]MARUM Center for Marine Environmental Sciences, University of Bremen, Bremen, Germany. [16]These authors contributed equally: Frederik V. Seersholm, Abigail Ramsøe, Jialu Cao. [17]These authors jointly supervised this work: Eske Willerslev, Martin Sikora. ✉e-mail: ewillerslev@sund.ku.dk; ew482@cam.ac.uk; martin.sikora@sund.ku.dk

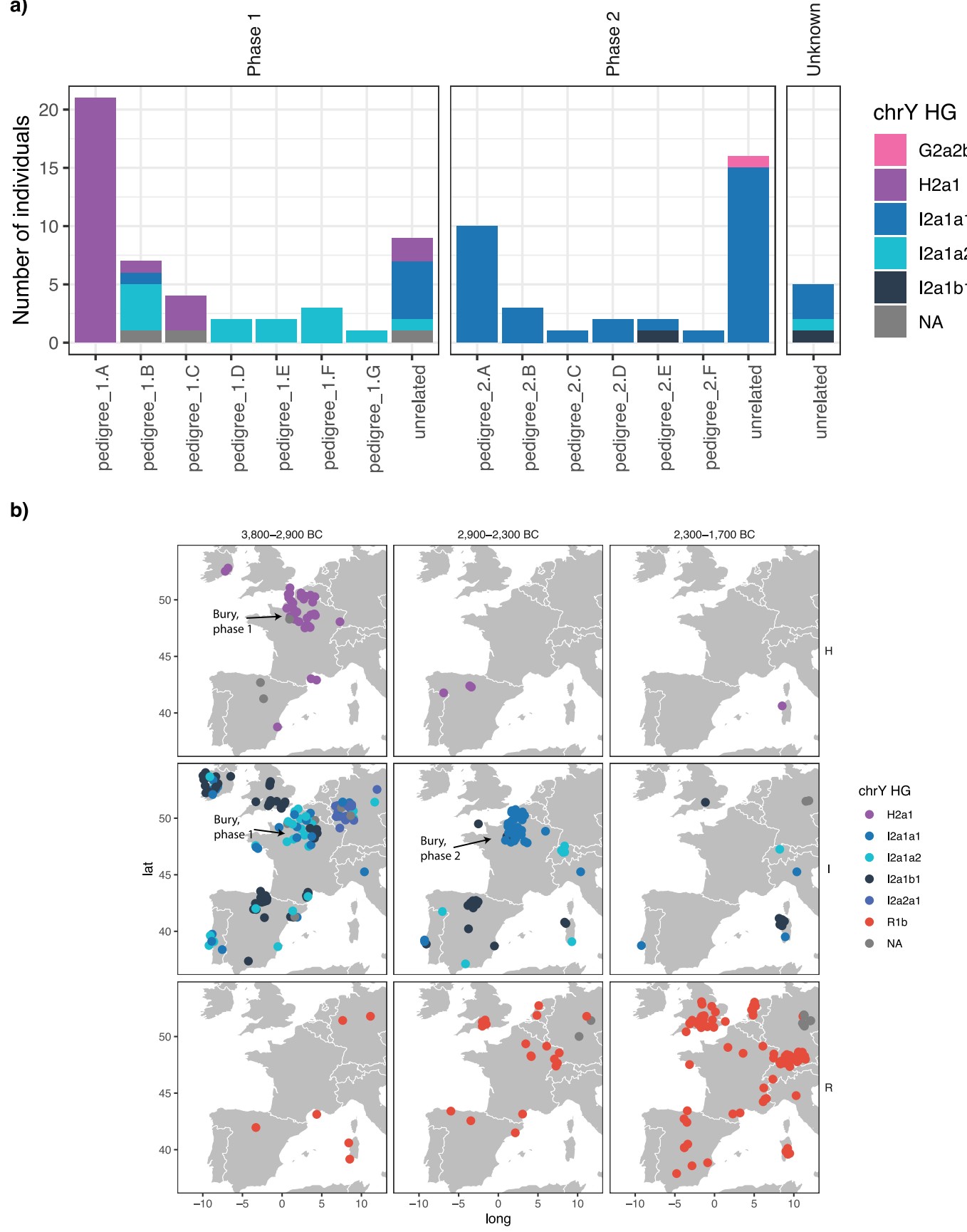

**Extended Data Fig. 1 | Chromosome Y haplogroup diversity. a)** Distribution of chromosome Y haplogroups within Bury. Variants of haplogroup I upstream of the main three "I" haplogroups were not coloured and were classified as 'NA'. **b)**. Map of Y chromosome haplogroups distribution in Neolithic Western Europe. Haplogroups were called using the ISOGG database. Only individuals with chromosome Y haplogroups within H, I and R are shown. Basemap data from Natural Earth (https://www.naturalearthdata.com/).

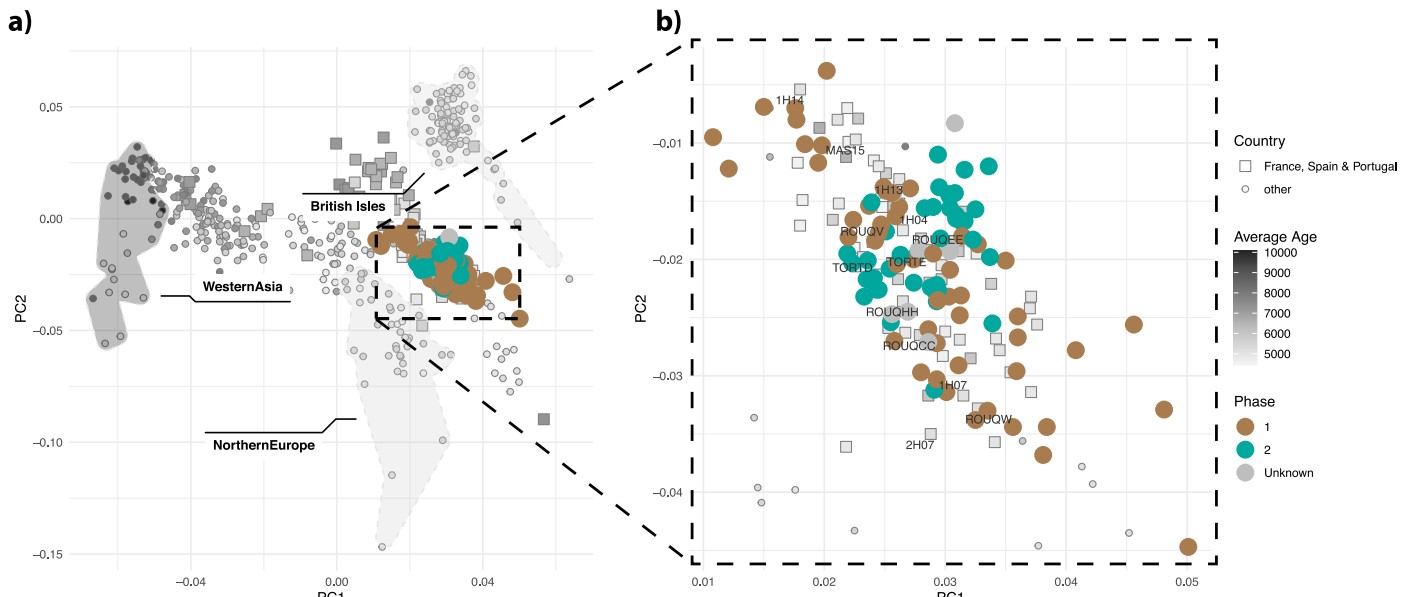

**Extended Data Fig. 2 | PCA of Neolithic individuals.** Individuals with more than 50% Neolithic ancestry dated to before 2,550 BC were plotted. Reference data from France, Spain and Portugal are represented by squares, whereas all other individuals are represented by circles. Phase 1 and Phase 2 individuals from this study were highlighted in green/brown color, respectively. **a**) Full plot with groups of individuals from the British Isles, Northern Europe and Western Asia highlighted. **b**) Insert focusing on individuals from Bury, with reference data from Seguin-Orlando et al. 2021 highlighted[25].

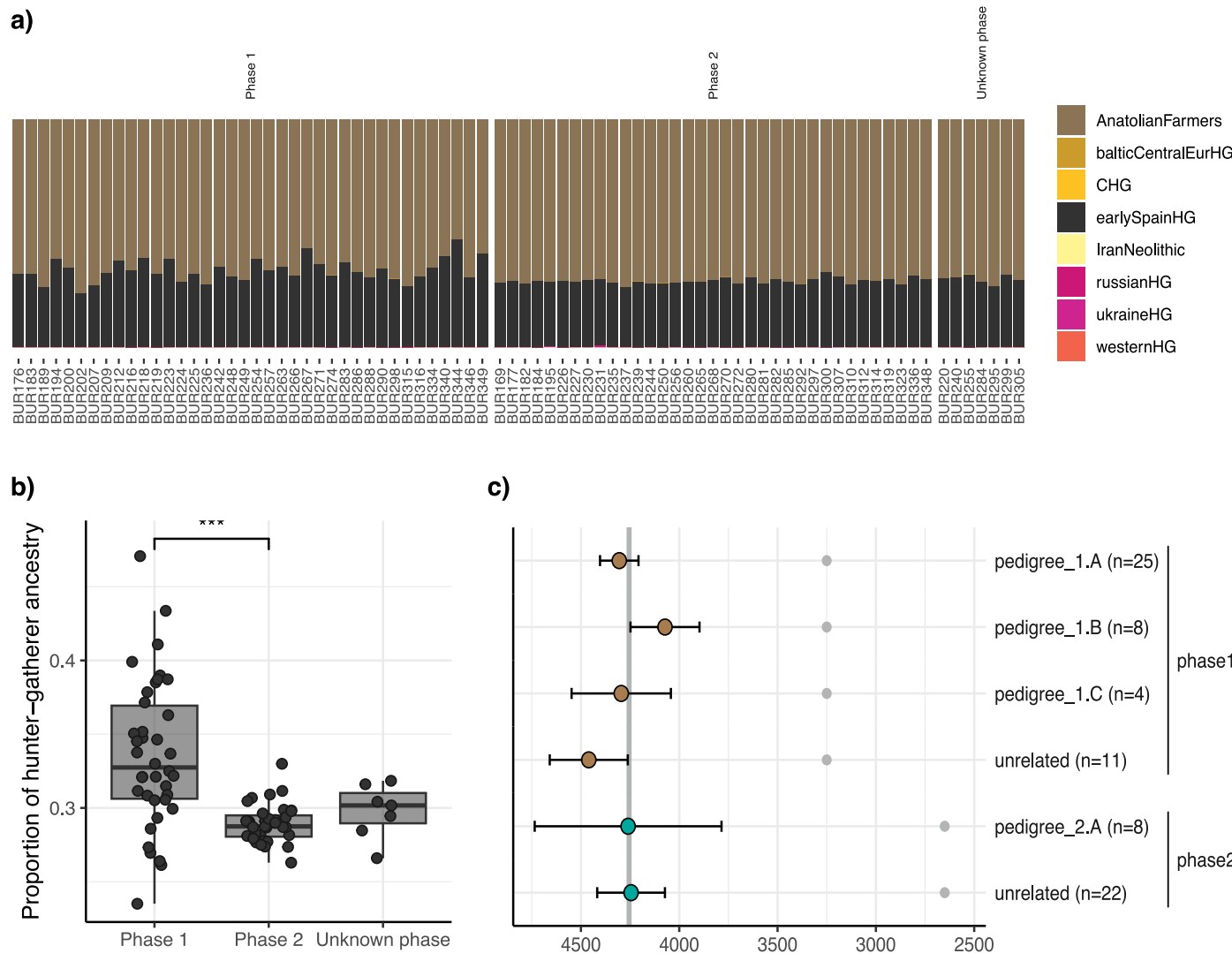

**Extended Data Fig. 3 | Hunter-gatherer ancestry at Bury. a)** Mixture modelling results using the following groups as source populations: AnatolianFarmers, CHG, IranNeolithic, balticCentralEurHG, earlySpainHG, russianHG, ukraineHG, westernHG. All individuals were modelled exclusively with varying proportions of ancestry from 'earlySpainHG' and 'AnatolianFarmers', suggesting that these two populations are the best sources for the hunter-gatherer and Neolithic ancestries in the Bury individuals. See Allentoft et al. (2024) for details.
**b)** Proportion of hunter-gatherer (source: earlySpainHG) ancestry between the two phases at Bury. ***Wilcoxon rank-sum test (one sided): p-value = 2.281e-07,

W = 1,100, effect size=0.4. **c)** Estimates of the timing of hunter-gatherer and Neolithic admixture in individuals from Bury using DATES[82]. As source groups we used the two populations 'earlySpainHg' and 'earlySpainFranceNeolithic' which were identified in our mixture modelling analysis as the best sources for HG and Neolithic ancestry, respectively, in the Bury individuals (see 'Mixture modelling' above). Most likely sample ages are depicted as grey circles (phase1: 3,250 BC, phase2: 2,650 BC). While no time point exists where all confidence intervals overlap, we have highlighted the most likely admixture time in the grey rectangle (4,248-4,262 BC).

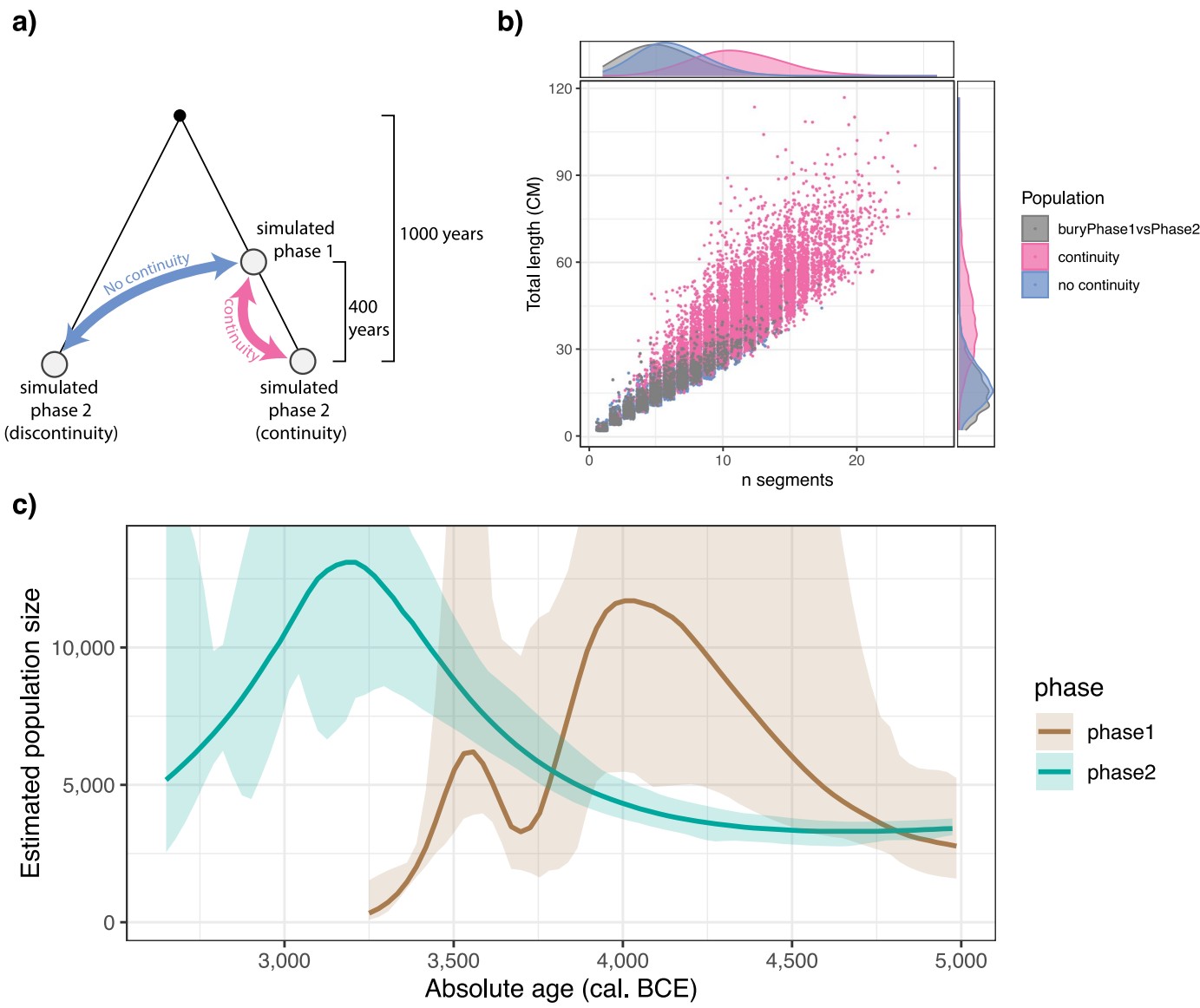

**Extended Data Fig. 4 | Distinct population histories between Phase 1 and Phase 2. a)** Overview of the simulation strategy. **b)** IBD sharing between the two phases at Bury (grey) compared to simulated data of two distinct scenarios: continuity (pink) and discontinuity (blue). **c)** IBDne population size estimates for phase 1 and 2, respectively plotted against absolute age (ca. BC).

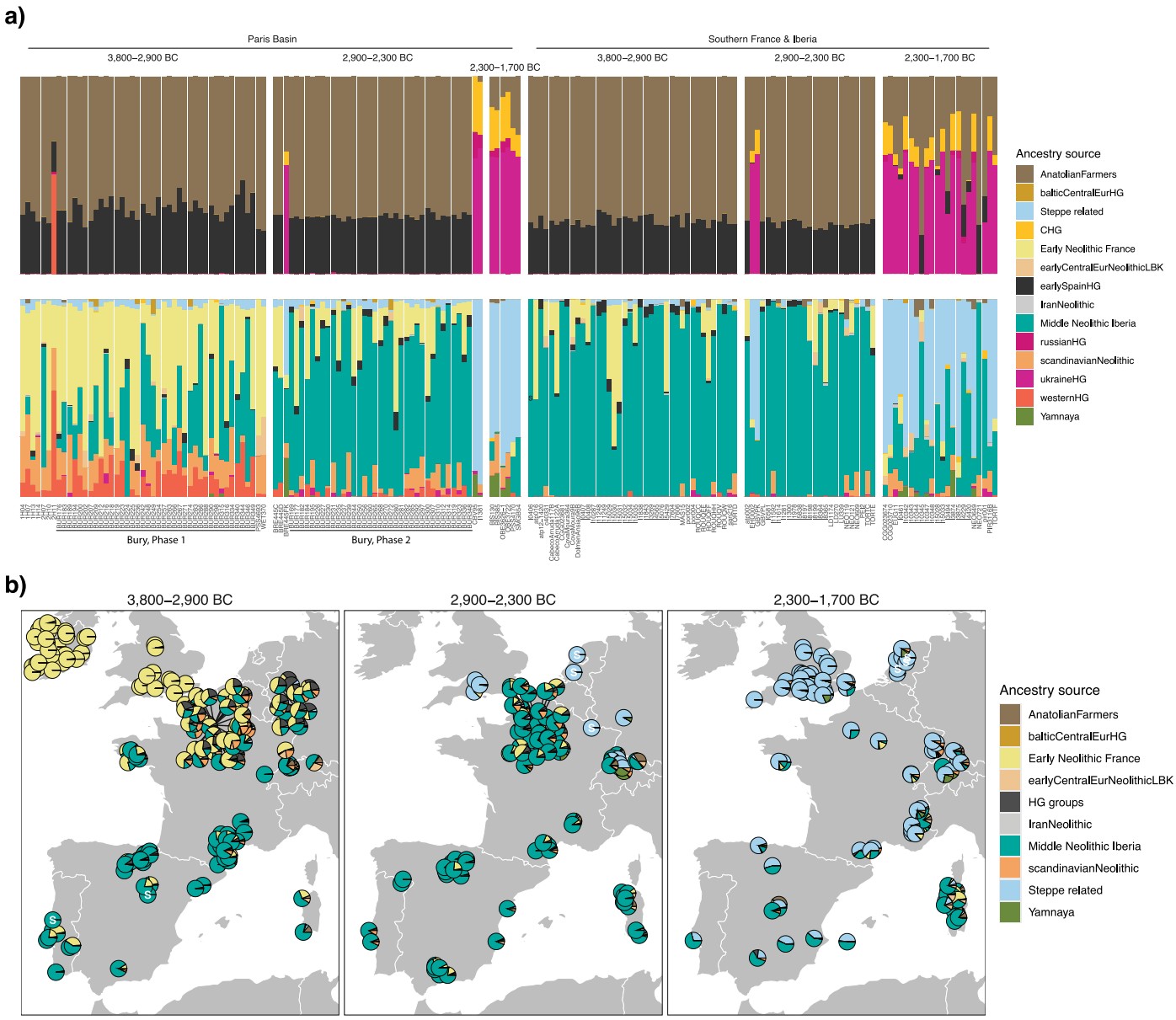

**Extended Data Fig. 5 | Mixture modelling. a**) Proportions of modelled ancestries per individual stratified by time period and region ("Paris Basin" and "Southern France & Iberia"). Top panel is modelled using only distal sources, while the bottom panel is modelled using both distal and proximal sources. **b**). Pie charts of ancestry proportions per individual modelled using both distal and proximal sources (lower part of panel **a**). Basemap data from Natural Earth (https://www.naturalearthdata.com/).

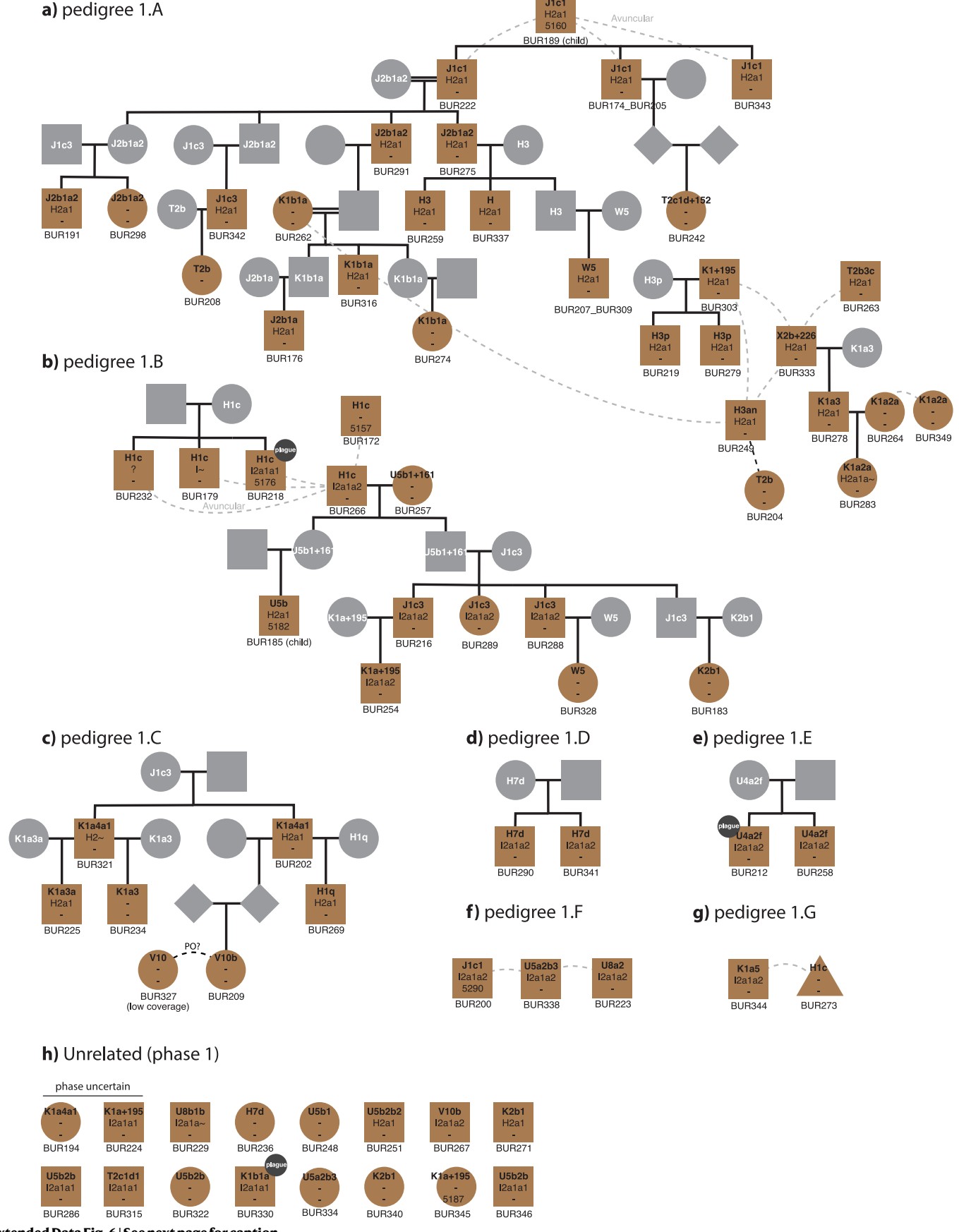

**Extended Data Fig. 6 | See next page for caption.**

**Extended Data Fig. 6 | Pedigrees from Phase 1.** Circles and squares represent females and males, respectively, while triangles represent unknown sex. Inside each shape, individuals are labelled with their mitochondrial haplogroup (first line), chromosome Y haplogroup (second line) and calibrated median age (last line). Furthermore, solid black lines between shapes indicate well defined first degree relationships, while stippled black and grey lines specify unknown or uncertain first and second degree relationships, respectively. Panels a-g represent Pedigrees 1.A to 1.G, respectively, whereas h) represents unrelated individuals from Phase 1.

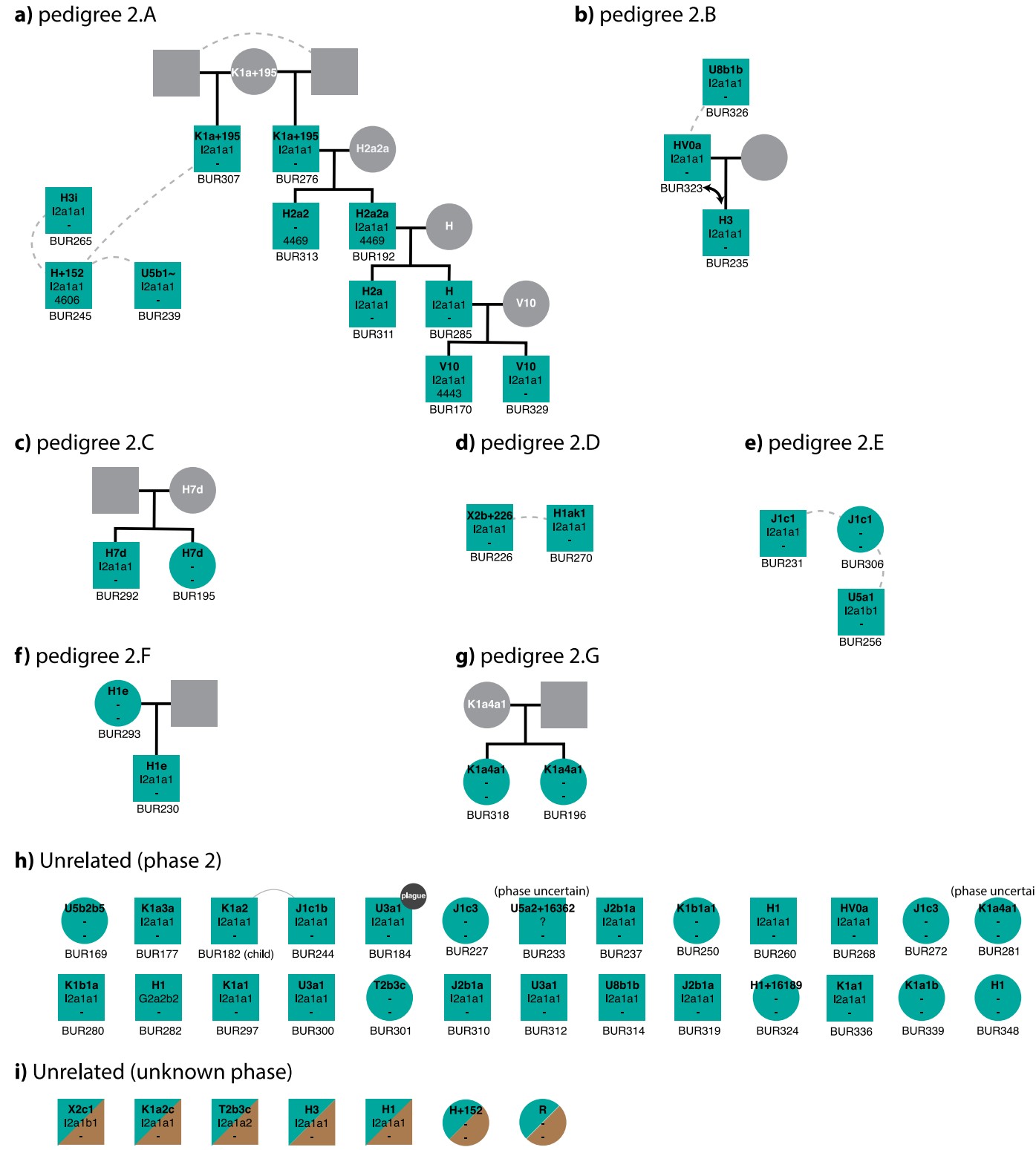

**Extended Data Fig. 7 | Pedigrees from Phase 2.** Circles and squares represent females and males, respectively, while color specifies burial phase (green: phase 2, green/brown: unknown phase). Inside each shape, individuals are labelled with their mitochondrial haplogroup (first line), chromosome Y haplogroup (second line) and calibrated median age (last line). Furthermore, solid black lines between shapes indicate well defined first degree relationships, while stippled black and grey lines specify unknown or uncertain first and second degree relationships, respectively. Panels a-g represent pedigrees 2.A to 2.G, respectively, whereas panels h and i represent unrelated individuals from Phase 2 and unrelated individuals where phase could not be determined, respectively.

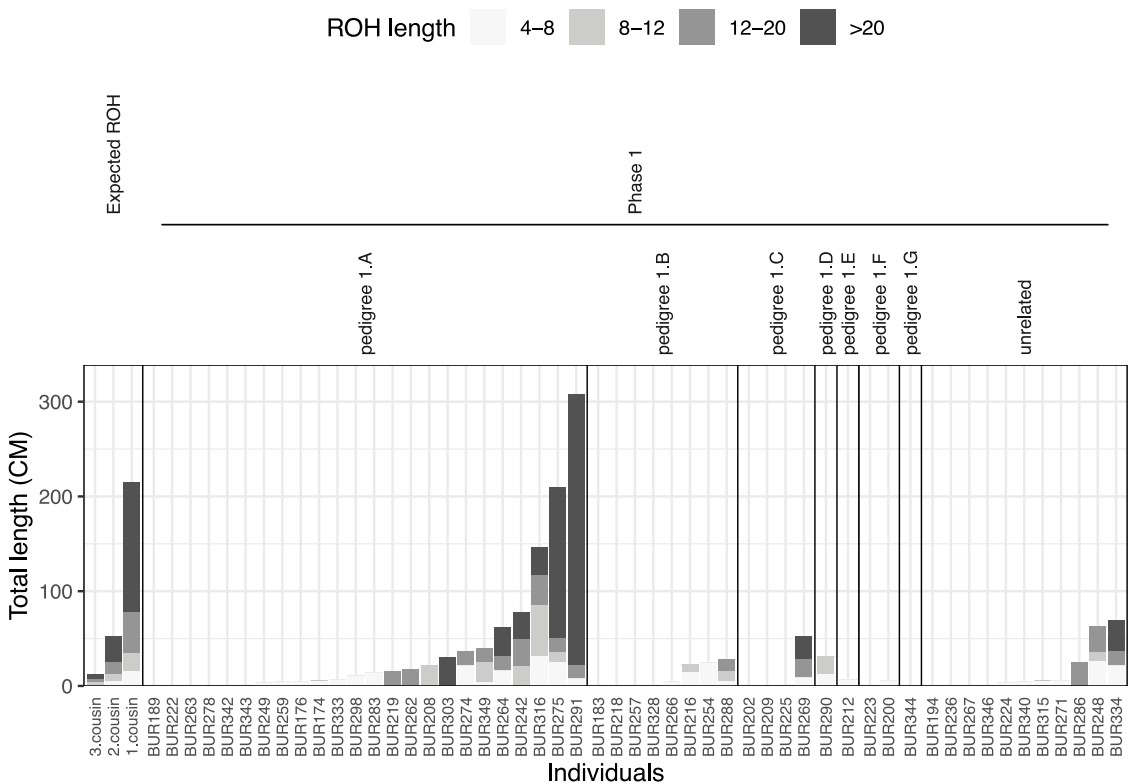

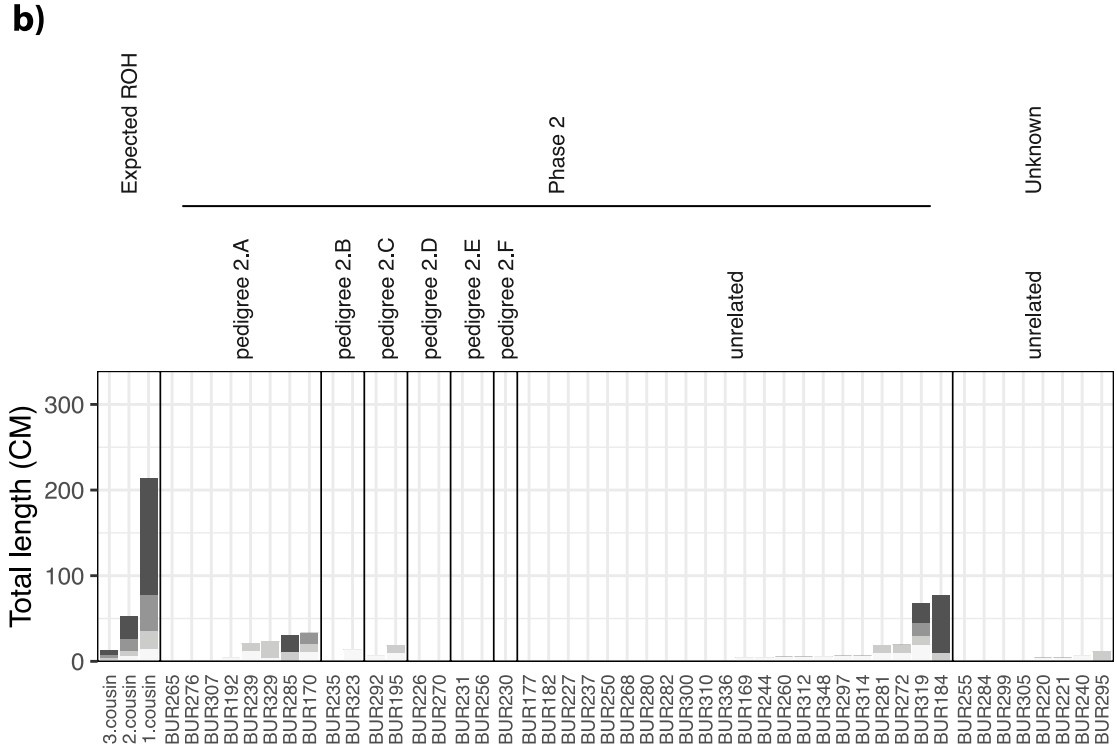

**Extended Data Fig. 8 | Runs of homozygousity (RoH). a)** Sum of RoH lengths coloured by segment size per individual at Phase 1. The plot is stratified by burial phase and family line. 'Expected ROH' represents estimated ROH profiles of 1st. 2nd and 3rd cousins from hapROH[83]. **b)** Sum of RoH lengths coloured by segment size per individual at Phase 2 and unknown phase. 'Expected ROH' represents estimated ROH profiles of 1st. 2nd and 3rd cousins from hapROH[83].

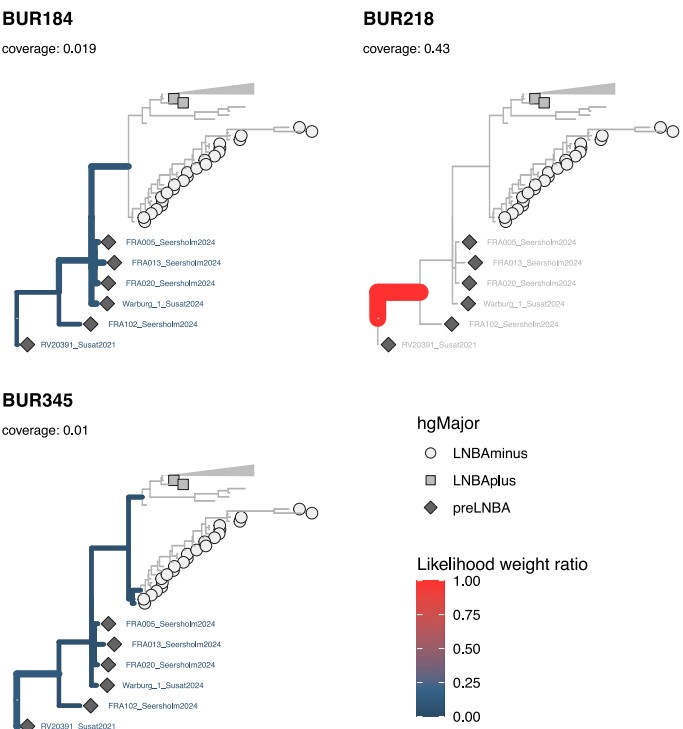

**Extended Data Fig. 9 | Phylogenetic placements of plague cases over 0.01X.** Each phylogenetic tree represents one placement, with sample names indicated at the top. Color and branch widths indicate support for phylogenetic placements.

# Reporting Summary

## Statistics

For all statistical analyses, confirm that the following items are present in the figure legend, table legend, main text, or Methods section.

| n/a | Confirmed | |
|---|---|---|
| ☐ | ☒ | The exact sample size (*n*) for each experimental group/condition, given as a discrete number and unit of measurement |
| ☐ | ☒ | A statement on whether measurements were taken from distinct samples or whether the same sample was measured repeatedly |
| ☐ | ☒ | The statistical test(s) used AND whether they are one- or two-sided<br>*Only common tests should be described solely by name; describe more complex techniques in the Methods section.* |
| ☒ | ☐ | A description of all covariates tested |
| ☒ | ☐ | A description of any assumptions or corrections, such as tests of normality and adjustment for multiple comparisons |
| ☐ | ☒ | A full description of the statistical parameters including central tendency (e.g. means) or other basic estimates (e.g. regression coefficient) AND variation (e.g. standard deviation) or associated estimates of uncertainty (e.g. confidence intervals) |
| ☐ | ☒ | For null hypothesis testing, the test statistic (e.g. *F*, *t*, *r*) with confidence intervals, effect sizes, degrees of freedom and *P* value noted<br>*Give P values as exact values whenever suitable.* |
| ☒ | ☐ | For Bayesian analysis, information on the choice of priors and Markov chain Monte Carlo settings |
| ☒ | ☐ | For hierarchical and complex designs, identification of the appropriate level for tests and full reporting of outcomes |
| ☒ | ☐ | Estimates of effect sizes (e.g. Cohen's *d*, Pearson's *r*), indicating how they were calculated |

*Our web collection on statistics for biologists contains articles on many of the points above.*

## Software and code

Policy information about availability of computer code

| | |
|---|---|
| Data collection | Illumina NovaSeq system |
| Data analysis | ANGSD (0.931)<br>Bcftools (1.16)<br>bedtools (v2.31.0)<br>bwa (0.7.17)<br>convertf(version: 5722)<br>dates (Version 4010)<br>decluster<br>EPA-ng v0.3.8<br>Gappa (v0.8.0)<br>GATK (v4.3.0.0)<br>gcta64 (v1.94.1)<br>glimpse (v1.1.1)<br>preseq (v 3.2.0)<br>ibdseq (r1206)<br>picard (v 3.1.1)<br>HAPLOGREP (2.1.25)<br>MUTSERVE (1.3.0)<br>Jvarkit (v dbdbed3a9)<br>Java (v 17.0.3) |

KIN (0.1.0)
KINgaroo (0.1.0)
King (v 2.3.0)
Krakenuniq (v 1.0.4)
mapDamage (2.2.0-86-g81d0aca)
METADMG (v 0.2-86-gcba5d46)
Mosdepth (v 0.3.3)
ngsRelate
plink (v1.90b6.21)
ngsngs (v0.9.0)
python (3.10.8)
PRIMUS (v1.9.0)
msPrime (1.2.0)
READ2.py (v2.00)
raxml-ng (v. 1.2.0)
realSFS
samtools (v 1.21)
seqtk (1.3-r106)
smartpca (eigensoft v. 8.0.0)

R packages:
Argparse (v 2.2.2)
data.table (v 1.14.8)
doParallel (1.0.17)
dplyr (1.1.1)
forcats (v 1.0.0)
foreach (v1.5.2)
furrr (v 0.3.1)
ggplot2 (v3.4.2)
ggtree (v3.6.2)
ggraph (v 2.1.0)
ggVennDiagram (v 1.5.0)
gridExtra (v 2.3)
igraph (v 1.4.2)
plyr (v 1.8.8)
purrr (v 1.0.1)
readr (v 2.1.4)
scales (v 1.4.0)
stringr (v 1.5.0)
tidyverse (v 2.0.0)
tidygraph (v 1.2.3)
vcfR (v 1.14.0)

For manuscripts utilizing custom algorithms or software that are central to the research but not yet described in published literature, software must be made available to editors and reviewers. We strongly encourage code deposition in a community repository (e.g. GitHub). See the Nature Portfolio guidelines for submitting code & software for further information.

# Data

Policy information about availability of data

All manuscripts must include a data availability statement. This statement should provide the following information, where applicable:
- Accession codes, unique identifiers, or web links for publicly available datasets
- A description of any restrictions on data availability
- For clinical datasets or third party data, please ensure that the statement adheres to our policy

Fastq files with collapsed and adapter trimmed reads from this study have been deposited in the European Nucleotide Archive under accession number PRJEB95770 (Supplementary Table 12).

# Research involving human participants, their data, or biological material

Policy information about studies with human participants or human data. See also policy information about sex, gender (identity/presentation), and sexual orientation and race, ethnicity and racism.

| | |
|---|---|
| Reporting on sex and gender | Not appllicable |
| Reporting on race, ethnicity, or other socially relevant groupings | Not appllicable |
| Population characteristics | Not appllicable |
| Recruitment | Not appllicable |

| Ethics oversight | Not appllicable |
|---|---|

Note that full information on the approval of the study protocol must also be provided in the manuscript.

# Field-specific reporting

Please select the one below that is the best fit for your research. If you are not sure, read the appropriate sections before making your selection.

☒ Life sciences ☐ Behavioural & social sciences ☐ Ecological, evolutionary & environmental sciences

For a reference copy of the document with all sections, see nature.com/documents/nr-reporting-summary-flat.pdf

# Life sciences study design

All studies must disclose on these points even when the disclosure is negative.

| | |
|---|---|
| Sample size | No tests were carried out to predetermine sample size. Sample size was determined by the availability of archaeological material, and on the DNA preservation in these samples. |
| Data exclusions | Samples with a final depth of coverage under 0.01X or libraries with high contamination estimates as determined by ContamMix (v1.0.10) were excluded from downstream analyses. This cutoff was predetermined and follows Seersholm et al. 2024 |
| Replication | Out of the 181 samples analysed in this study, 145 are represented by more than one sequencing library. Having multiple sequencing libraries for each sample serves to validate sequencing results and to pinpoint potential sample swaps. Of the 145 samples represented by more than one sequencing library, 82 samples have multiple sequencing libraries of sufficient coverage to assess whether these belong to the same genetic individual (>0.01). Except from one library pair, all of these where characterised as coming from the same individual using READ2. The library pair not characterised as coming from the same individual, was characterised as "first degree relatives", presumably due to the low coverage of both libraries (0.01X and 0.01X). Apart from this type of replication, replication of experimental findings is generally not applicable for this kind of ancient DNA study because of the unique nature of ancient human remains. |
| Randomization | Not relevant. Sample allocation was not random, but followed archaeological burial phase and biological kinship. |
| Blinding | Not applicable. Ancient DNA research is observational, with samples defined by archaeological context rather than experimental assignment, and all laboratory and bioinformatic procedures were applied uniformly using predefined protocols. As there were no interventions or subjective outcome assessments, blinding was not relevant to data collection or analysis. |

# Reporting for specific materials, systems and methods

We require information from authors about some types of materials, experimental systems and methods used in many studies. Here, indicate whether each material, system or method listed is relevant to your study. If you are not sure if a list item applies to your research, read the appropriate section before selecting a response.

## Materials & experimental systems

| n/a | Involved in the study |
|---|---|
| ☒ | ☐ Antibodies |
| ☒ | ☐ Eukaryotic cell lines |
| ☐ | ☒ Palaeontology and archaeology |
| ☒ | ☐ Animals and other organisms |
| ☒ | ☐ Clinical data |
| ☒ | ☐ Dual use research of concern |
| ☒ | ☐ Plants |

## Methods

| n/a | Involved in the study |
|---|---|
| ☒ | ☐ ChIP-seq |
| ☒ | ☐ Flow cytometry |
| ☒ | ☐ MRI-based neuroimaging |

## Palaeontology and Archaeology

| | |
|---|---|
| Specimen provenance | The sampling was made in 2013 under the supervision of Laure Salanova, the scientific authority of the Bury project. In total, we collected 181 ancient human teeth samples, representing 179 individuals (57% of the total of 316 individuals estimated to have been buried at the site). We decided to sample mandibular teeth, since the skulls could not be linked to the identified archaeological individuals (Supplementary table 1). In total, we sampled more than 71% of the testable mandibles, including all but one of the mandibles from identified archaeological individuals. Lastly, we also sampled 21 loose teeth to ensure the broadest representation possible. Along with all other samples, these loose teeth were subsequently screened genetically to identify any duplicates. |
| Specimen deposition | Leftover DNA digests, extract and sequencing libraries are stored at the DNA laboratory facilities at Globe Institute, Copenhagen. Upon completion of this project, leftover bone material will be returned to respective museum or university collections from which they were sampled. |

| Dating methods | Radiocarbon dating was performed at the Keck carbon cycle AMS facility, University of California, Irving. The samples were decalcified in 1N HCl, gelatinized at 60°C and pH 2, and ultrafiltered to select a high molecular wt fraction (>30kDa). δ13C and δ15N values were measured to a precision of <0.1‰ and <0.2‰, respectively, on aliquots of ultrafiltered collagen, using a Fisons NA1500NC elemental analyzer/Finnigan Delta Plus isotope ratio mass spectrometer. Datings were calibrated in Oxcal 4.4.4 using the Intcal20 calibration curve. |

☒ Tick this box to confirm that the raw and calibrated dates are available in the paper or in Supplementary Information.

| Ethics oversight | No ethical approval was required for this study. |

Note that full information on the approval of the study protocol must also be provided in the manuscript.

## Plants

| Seed stocks | *Report on the source of all seed stocks or other plant material used. If applicable, state the seed stock centre and catalogue number. If plant specimens were collected from the field, describe the collection location, date and sampling procedures.* |
| Novel plant genotypes | *Describe the methods by which all novel plant genotypes were produced. This includes those generated by transgenic approaches, gene editing, chemical/radiation-based mutagenesis and hybridization. For transgenic lines, describe the transformation method, the number of independent lines analyzed and the generation upon which experiments were performed. For gene-edited lines, describe the editor used, the endogenous sequence targeted for editing, the targeting guide RNA sequence (if applicable) and how the editor was applied.* |
| Authentication | *Describe any authentication procedures for each seed stock used or novel genotype generated. Describe any experiments used to assess the effect of a mutation and, where applicable, how potential secondary effects (e.g. second site T-DNA insertions, mosiacism, off-target gene editing) were examined.* |

