## [Peer Review File · Nature Ecology & Evolution]

Population discontinuity in the Paris Basin linked to evidence of the Neolithic Decline

Corresponding Author: Dr Martin Sikora

Version 0:

Decision Letter:

17th April 2025

Dear Dr Sikora,

Your manuscript entitled "Plague and the European Neolithic Decline: Population Discontinuity and Shifts in Societal Structure" has now been seen by four reviewers, whose comments are attached. The reviewers have raised a number of concerns which will need to be addressed before we can offer publication in Nature Ecology & Evolution. We will therefore need to see your responses to the criticisms raised and to some editorial concerns, along with a revised manuscript, before we can reach a final decision regarding publication.

As you'll see from the reviewer reports, while some aspects of the data and analyses come in for praise, the reviewers are near-unanimous in their feeling that the narratives related to plague and population decline are overstated based on the available data and far from the strongest aspects of the manuscript. We recognise that in submitting this research to Nature Ecology & Evolution, you may have felt that this overtly ecological framework would appeal to the journal's scope, and that removing this narrative (as requested by the reviewers) might undermine the appropriateness of the submission to this journal. However, editorially we feel that there is enough interest in what the analyses reveal about changing social structure and relationships (an aspect of human ecology) to warrant further consideration by the journal even if the plague narrative is downplayed. Therefore we encourage you to revise according to the reviewer reports and resubmit to Nature Ecology & Evolution.

Please highlight all changes in the manuscript text file [OPTIONAL: in Microsoft Word format].

* If you have not done so already please begin to revise your manuscript so that it conforms to our Article format instructions at <http://www.nature.com/natecolevol/info/final-submission>. Refer also to any guidelines provided in this letter.

* Extended Data Figures - please ensure that any supplementary figures and tables that are crucial to the manuscript's conclusions are converted into Extended Data figures and tables to increase visibility of these data. Extended Data figures and tables are online-only (present in the online PDF and full-text HTML versions of the paper), peer-reviewed display items that provide essential background to the article but are not included in the main article due to space constraints. A maximum of ten Extended Data display items (figures and tables) is permitted.

Link Redacted

Nature Ecology & Evolution is committed to improving transparency in authorship. As part of our efforts in this direction, we are now requesting that all authors identified as 'corresponding author' on published papers create and link their Open Researcher and Contributor Identifier (ORCID) with their account on the Manuscript Tracking System (MTS), prior to acceptance. ORCID helps the scientific community achieve unambiguous attribution of all scholarly contributions. You can create and link your ORCID from the home page of the MTS by clicking on 'Modify my Springer Nature account'. For more information please visit www.springernature.com/orcid.

[redacted]

Reviewer expertise:

Reviewer #1: Neolithic bioarchaeology

Reviewer #2: palaeopathogen genomics and *Yersinia pestis*

Reviewer #3: Neolithic/Bronze Age bioarchaeology

Reviewer #4: ancient genomics

Reviewers' comments:

Reviewer #1 (Remarks to the Author):

Nature review

The paper titled "Plague and the European Neolithic Decline: Population Discontinuity and Shifts in Societal Structure" presents compelling paleogenomic results from the collective burial site of Bury (Paris Basin) in the Late Neolithic. The study identifies two distinct phases of body deposition, each exhibiting different genetic pedigree patterns. Additionally, four individuals were found to carry the *Yersinia pestis* pathogen. This finding is discussed in relation to population decline at the end of the Neolithic and the cessation of megalithic construction.

While the paper presents robust and novel data, it cannot be published in its current form and requires major revisions. My primary concern is the focus on the plague pathogen as the key argument for positioning Bury as the missing link in explaining Neolithic population decline. This claim is not sufficiently supported, given that only four individuals tested positive for the pathogen.

This overemphasis on plague diminishes the significance of the truly groundbreaking findings from Bury, which deserve greater attention. These include:

- Rare evidence of half-sibling relationships
- Clear indications of a shift in social organisation over 100–200 years

Rather than prioritising the plague narrative, more space should be dedicated to properly presenting the site and incorporating archaeoanthatological analysis to contextualise these important discoveries.

General comments

The dates used throughout the paper are inconsistent, alternating between BCE and BP. Additionally, some individuals are presented with a date range, while others are listed with a standard deviation (\pm). A uniform approach should be applied. The authors use "pedigree" and "family" interchangeably, despite acknowledging that kinship and family structures in the Neolithic period remain unknown. It would be more appropriate to consistently use "pedigree", as it is a more neutral term.

Terminology Clarifications

- PCA (Principal Component Analysis): This should be defined in the first paragraph of the section Genetic discontinuity between two phases.
- "Founding act" and "founder": The reasoning behind these terms is not clearly explained. The authors should clarify whether this designation is based on C14 dating, stratigraphic position, or other criteria to justify their status as the source of the burial. If Pedigree 1 represents the earliest burial phase, this should be explicitly stated. A reference to Rivollat et al. (2023) could help strengthen the argument.
- "Avuncular" (Figure 3): This term should be explicitly defined.
- "Dark Age": This term should be avoided, as it is heavily associated with the Middle Ages. Since the paper consistently uses "Neolithic decline", this term should be maintained for consistency.

Additional References Needed

More references should be included to support key discussions in the paper:

- Demographic profile: Additional references are needed to contextualise the demographic data and their interpretation, in the text and in supplementary information 1.
- Reference panel of ancient genomes: The discussion on IBD (identity by descent) and farming ancestry across Europe should be supported by recent studies.
- Comparative studies on social structure: It is surprising that the authors do not reference recent publications addressing similar topics in either the same geographic area in earlier periods (e.g., Rivollat et al., 2023) or the same period and burial structure in a different region (Cassidy et al., 2020). Additionally, the social aspect of the population dynamic should be also backed up by social anthropology references and archaeoethnography studies (e.g. the work of Jeunesse and coll. or Gally).
- Simulation of two scenarios (discontinuity vs. continuity): These should be explicitly detailed with appropriate references or a reference to the methods section.
- Social and cultural ties in Phase 2: The argument that longer occupation suggests a greater role for social or cultural continuity should be supported with literature.
- Plague outbreaks and site abandonment: The authors suggest that plague outbreaks may have led to site abandonment without proper burial of the last deceased. However, this claim requires references to comparable archaeological evidence of epidemic-driven abandonment.
- The Scandinavian Neolithic decline: This parallel is mentioned but lacks proper referencing and context.
- Strontium analysis and broader mobility context (Supplementary Information 5): More references should be included to strengthen the interpretation of these findings.

Summary

Some editing is required. One sentence remains incomplete: “Recent research that is ‘Neolithic decline’ ...” This should be rephrased for clarity.

Additionally, *allée sépulcrale* should be written in italics.

Introduction

The authors should provide a more detailed definition of what constitutes a collective burial site, along with an overview of the main theories regarding this type of site. This would strengthen the subsequent arguments about population decline and provide a clearer framework for discussion. What are the prevailing interpretations of collective burial sites in archaeological research?

Regarding the evidence for population decline, the authors should reference relevant literature, including archaeological and demographic sources, to support their claims. What types of evidence are typically used to identify such declines, and how do they apply to this study?

The description of the semi-megalithic *allée sépulcrale* of Bury requires additional context and a more detailed stratigraphic account to demonstrate the presence of two distinct occupations. This could be including as supplementary information, such as field documentation like section drawings. It would allow for a clearer evaluation and validation of the authors' interpretations. Additionally, the concept of a semi-megalithic structure should be explicitly defined for clarity.

The demographic profile for the first phase is discussed, but there is no mention of the second phase. A similar analysis should be provided to ensure a comprehensive understanding of the demographic changes over time.

Results

The authors present a sample of 182 teeth from 179 individuals. How was the selection made to ensure that the same individual was not tested twice? In commingled contexts, we usually sample the bone that give the best MNI (Minimum Number of Individuals), but this does not seem to be the case here. The sampling strategy should be detailed.

In the Excel sheet presenting the sample, some individual numbers are listed alongside loose teeth. However, this aspect has not been explained in the context of the burial, highlighting the need for a clearer presentation of the site and a clear definition of the sampling strategy.

The statement, “genetic sexing [confirmed] the high predominance of males in both phases,” is made without a full biological profile for the entire site or for each phase. A comprehensive biological profile is crucial for a proper presentation of the site.

Figure 1: The publication date is missing in the caption.

Genetic discontinuity between two phases

The genetic ancestry determination should be contextualised more thoroughly. Given that this study focuses on the end of the Neolithic (even during its decline), why should we still be looking for Anatolian farmer origins and Mesolithic populations? Providing more context on recent research in this area would help guide the reader through the authors' line of reasoning and strengthen their argument.

A shift in societal structure inferred by genetic links

“we found that three-quarters of individuals...” Including precise percentages or actual numbers would enhance clarity.

A more detailed explanation of the sampling strategy would clarify why some individuals were not sampled or did not yield results. What were the selection criteria? For example, in Pedigree 1.A, the authors mention 29 sequenced individuals, while 21 were not sampled or were buried outside the grave. This highlights two key concerns:

1. The lack of contextual information regarding burial practices and funerary customs of the time.
2. The absence of a clear explanation of the sampling strategy.

Regarding Pedigree 1.A, a summary would be helpful—such as the number of generations represented and a general biological profile of the individuals within this pedigree. This should be done for each pedigree (in supplementary information).

For ID174, the reference to body position is noted, but the paper does not provide an overview of funerary practices, the bone assemblage at the site, or how these vary across different phases. This broader context is necessary for proper interpretation.

The results for ID185 are particularly intriguing and should definitely be developed further. However, the only hypothesis proposed by the authors is that the genetic link belongs to an unsampled individual. Could alternative explanations, such as

adoption or fostering, be considered? While rarely addressed in archaeological samples, insights from social anthropology could support this possibility. If proven accurate, this would actually be a major discovery.

For ID307 and ID276, the identification of half-siblings is noteworthy—not only within this burial site but also in the archaeological record more broadly. This rare finding warrants a more detailed discussion and comparison with other known case studies, as it represents a significant outcome.

Finally, the study does not present body positions for each identified individual, making it difficult to evaluate the claim that there is no correlation between genetic relationships and burial placement. Providing this data would strengthen the argument.

Evidence for the earliest diverging lineage of *Y. pestis* at Bury

The paper does not provide any data on the general health status of the population(s). Including such information would offer important context and potentially strengthen the authors' claims.

Additionally, I was unable to locate Figure 7.7.

Regarding the number of cases reported across the two phases, several points require clarification:

- The reported numbers ($85 + 56 = 141$) do not match the total population ($n = 316$), the sampled population ($n = 179$), or the number of teeth sampled ($n = 182$). Further explanation is needed to clarify these discrepancies.
- The authors effectively argue against their own hypothesis—that a plague outbreak led to the end of Phase 1—by presenting strong counterarguments. However, the alternative explanations they propose—stratigraphic position and a significant population increase—are not well documented and require further evidence.

Environmental data on the Neolithic decline

This paragraph is poorly documented and not well connected to the paper's results, as it does not include any direct data from Bury. Additionally, this section, along with Supplementary Information 3 and 4, does not appear to be relevant to the paper.

Discussion

I disagree with the statement that Bury is the missing link in demonstrating the "Neolithic decline", as no convincing evidence has been presented to establish a plague outbreak, its environmental impact, or other contributing factors. While I agree that these elements could be explored further using the new data from Bury, the current evidence is not compelling enough to support this claim.

The parallel drawn with Scandinavian data is weak, lacking sufficient references and clear examples that directly relate to what happened at Bury.

Overall, the discussion comes across as a tentative and unsubstantiated attempt to link Bury to broader patterns of Neolithic decline in Europe, rather than a well-supported argument grounded in robust comparative evidence.

Methods

Paleogenetics is not my area of expertise, so I have no specific comments on this section. However, the sample selection process should be clearly outlined here.

Supplementary Information 1

While I agree that the integration of juveniles into archaeological studies is not yet standard in biological anthropology, some relevant work has been published in the last 10 years that would be worth mentioning.

Supplementary Information 4

Although this section presents an interesting topic, its connection to Bury is not clearly explained, making it seem irrelevant to the paper. Additionally, the references are not cited in the main text.

Supplementary Information 5

This section is confusing, as it does not clarify why Le Tumulus des Sables is relevant for comparison with the Bury site. Further explanation is needed to establish its significance.

Reviewer #2 (Remarks to the Author):

The paper *Plague and the European Neolithic Decline* presents a highly speculative and poorly substantiated claim: that a genetic discontinuity at the Bury burial site reflects a widespread population collapse, potentially linked to *Yersinia pestis*. However, this interpretation suffers from major methodological flaws, weak genetic evidence, incomplete pathogen identification, and an overall lack of transparency in analytical procedures. The conclusions drawn are not supported by the data or methods described.

Major Issues

1. Population Genetics: Incomplete Comparisons and Lack of Continental Context

The paper argues that a break in identity-by-descent (IBD) sharing between burial phases at Bury reflects a broader demographic collapse. This argument is seriously undermined by several critical omissions:

No Pan-European IBD Framework:

The authors fail to provide a continent-wide IBD-sharing analysis to place the Bury discontinuity in context. Without comparison to other contemporaneous Neolithic populations, the claim of a continent-wide collapse remains speculative and unsupported.

Unclear IBD Comparisons with Other Sites:

The methodology for comparing IBD-sharing between Bury and other Neolithic sites is not transparently described. It remains unclear whether the genetic differences at Bury are unique or fall within expected regional variation. This lack of comparative clarity casts doubt on the robustness of the conclusions.

Missing or Misreferenced Figures:

Several figures mentioned in the main and supplementary texts are either missing or incorrectly referenced. For example, Supplementary Figure 7.4 (referenced for hierarchical clustering of IBD) is not present; Suppl. Information 6 only includes figures 6.1–6.7. Such inconsistencies undermine confidence in the visual evidence supporting key claims.

Lack of Effective Population Size (N_e) Estimates:

The authors do not include demographic modeling such as N_e estimates, genome-wide heterozygosity analyses, or coalescent simulations. These are essential to support claims of population decline. Instead, they rely on burial pattern interpretations, which could reflect social, ritual, or taphonomic changes rather than demographic collapse.

Cultural Transitions Misinterpreted as Demographic Events:

The genetic shifts around 4500 BP coincide with significant cultural transformations and changing burial customs. The persistence of distinct genetic profiles at Bury (e.g., minimal Steppe ancestry) argues against a collapse scenario and instead points to cultural continuity or interaction.

2. Kinship Analysis: Overreach from Limited and Ambiguous Data

The kinship analysis identifies three extended families within the burial site, yet the authors overgeneralize these findings to the entire population.

Inadequate Justification of Methodological Choices:

The authors used five different programs to estimate kinship—potentially increasing robustness—but failed to explain how conflicting outputs were reconciled. These tools often yield differing results, so clarity about which results were prioritized, and why, is essential for transparency and reproducibility.

Arbitrary Population-Level Extrapolations:

Most individuals are not closely related, contradicting the notion of a kinship-based burial pattern. Nevertheless, the authors generalize from a few family groups to the whole population without considering alternative explanations (e.g., social organization, mortuary variability).

Speculative Interpretations of Sex Ratios and Kinship Structures:

The underrepresentation of females is presented as evidence of patrilineal burial, but other plausible factors—such as preservation bias, differential burial treatment, or sampling artifacts—are not addressed.

3. *Yersinia pestis* Detection: Insufficient Evidence for Epidemic or Mortality Impact

The study's most dramatic claim—that *Y. pestis* contributed to population collapse—is based on questionable evidence.

Inadequate Pathogen Identification:

The study does not provide rigorous validation of *Y. pestis* reads. No mapping statistics, depth coverage data, or comparison with related environmental bacteria (e.g., *Y. pseudotuberculosis*) are presented. Contamination or misidentification cannot be ruled out.

Lack of Evidence for Virulence:

There is no reported coverage of key virulence factors (*pla*, *caf*, *ymt*), making it impossible to assess whether the strains could cause disease.

No Osteological or Burial Context Support:

There is no skeletal evidence of rapid death, nor signs of epidemic burial (e.g., mass graves, irregular positioning).

Furthermore, the low prevalence of *Y. pestis*—4% in Phase 1 and 2% in Phase 2—is far too low to suggest a significant outbreak.

Absence of Epidemiological Modeling:

The study lacks any modeling to assess the demographic impact of *Y. pestis*, leaving its role speculative and unconvincing.

4. Missing Methodological Details and Reproducibility Failures

The paper fails to provide critical methodological information necessary for independent verification and replication:

Mapping and Genotype Calling:

The reference genome used for mapping (e.g., hg19) is not stated. It is also unclear which positions were used for pseudohaploid calling—was the 1240K panel used?

Dataset Merging:

The authors state that they merged their data with ~6,000 samples, but do not specify which samples, from which release, or on what criteria (e.g., geography, time period). The current AADR release includes many more samples.

Principal Component Analysis (PCA):

The strategy for PCA is ambiguous. Was it performed using modern West Eurasian populations with projection of ancient individuals, or using a combined dataset?

IBD Analysis Parameters:

The IBD analysis was conducted using IBDseq, but no parameter details are provided. More suitable tools for ancient, low-coverage DNA (e.g., anclBD) are not considered, nor is a rationale given for choosing IBDseq.

Haplogroup Calling:

mtDNA and Y-chromosome haplogroups are reported without any details on the calling procedure, QC measures, or tools used. If performed manually, the criteria and process should be described.

Ancestry Estimation:

The study reports ancestry proportions (e.g., 90% farmer, 10% hunter-gatherer) but does not state which method was used (e.g., supervised ADMIXTURE, qpAdm). Modeling details, parameter choices, and full results (e.g., in Supplementary Tables) are missing.

5. Supplementary Materials: Missing Figures and Unclear Tables

Figure Inconsistencies:

As mentioned, several referenced figures are missing or misnumbered, including Supplementary Figure 7.4. This raises concerns about data accessibility and manuscript coherence.

Unclear Table Headers and Acronyms:

Supplementary Tables lack clear introductory descriptions. Column headers such as “nDer_woDam” are cryptic. If this refers to the number of derived alleles excluding those with deamination (i.e., only transitions), it should be explicitly stated. All acronyms should be written out to ensure accessibility for readers beyond the immediate research community.

This study does not provide the data quality, analytical transparency, or methodological rigor required to support its dramatic claims of a *Yersinia pestis*-driven demographic collapse in Neolithic Europe.

The genetic discontinuity observed at Bury is not convincingly linked to broader European trends.

The kinship analysis is based on limited and ambiguously interpreted data.

The pathogen evidence lacks rigor and plausibility.

The methods are insufficiently described to allow for replication or proper scrutiny.

Supplementary materials contain missing and confusing elements that further weaken the study's credibility.

In sum, what is interpreted here as population collapse is more plausibly explained by cultural and mortuary transitions around 4500 BP. Future studies must apply far more robust analytical frameworks and uphold higher standards of scientific transparency before advancing such consequential conclusions

Reviewer #3 (Remarks to the Author):

Ramsoe et al. present archaeogenetic data from 133 human burials recovered from a the Bury megalithic tomb in northern France where there were two distinct phases of construction and burial separated by one to two centuries: at the end of the fourth Millennium BC and at the beginning of the 3rd millennium BC. They find that while the people buried in both phases showed a bias towards males, patterns of relatedness differed, suggesting different social rules dictating whose remains were placed in the tombs, with the first phase including selected members of wider genealogies, and the second including more infrequent groups of close genetic relatives. They also find that ancestries carried by groups in either phase varied, and patterns of IBD DNA segment sharing suggests that the people from the second phase were largely not descendants of those in the first phase and represented a new community of people. Previous osteological analysis of the assemblage had found that the demographic profile of the people in the first phase was indicative of a catastrophic mortality profile or one where the population was growing rapidly. Finally, they found that four individuals, three of the final burials from the first phase and one from the second phase showed evidence for having been infected with *Yersinia pestis* bacteria (plague) when they died.

The authors' interpretation of this sequence as the community represented in the first phase of activity having been largely killed off by plague before a new, mostly unrelated community with different patterns of social organisation moved in two centuries later and remodelled the tomb to bury their own dead. The authors argue this supports a model where disease and specifically a plague pandemic substantially contributed to the 'Late Neolithic decline' in Europe a period which sees evidence for population decline which they specifically link to the decline in the construction of 'complex megalithic architecture'. Similarly-dated individuals infected with *Yersinia pestis* from Orkney and Scandinavia have been argued to be the result of localised outbreaks related to zoonotic spillovers rather than indicative of a widespread pandemic, but the authors argue their evidence from Bury suggests that plague was widespread and deadly enough to be responsible for significant demographic decline across Europe.

Clearly an incredible amount of effort has gone into the data generation and computational analysis for this paper and the authors should be commended for this. In this aspect alone, this paper would be an impressive addition to the published literature. Archaeogenetics papers presenting results or even (as in this case) hundreds of ancient genomes from single sites have rapidly become the norm more recently, but the number of individuals included here, the age of the samples and the variability in patterns of organisation they find mean that the authors' paper is a cut above, providing some truly novel insights into communities who used the Bury tomb for burying some of their dead. It is the first time, to my knowledge that the differences between two different phases of deposition at a tomb like this have been explored in this much detail, and it really provides fascinating evidence of reuse by a second community largely unrelated to the first. The authors use of IBD segment-sharing to look at continuity between the phase 1 and phase 2 population in the tomb is ingenious and provides a way forward for assessing these sorts of questions at other sites/monuments used by communities with similar broad ancestries. The paper draws upon standard established methods of data generation and analysis, is well written and the figures are all clear and appropriate.

While the standard of data and analysis is exceptional, I'm far from convinced by the interpretation, particularly in terms of broader arguments about a Late Neolithic decline and the association with disease and the decline of megalithic structures. The authors assert in no uncertain terms that the results from Bury when considered alongside recent related results indicate an 'event' at the end of 4th Millennium BC whereby a pan-European plague pandemic caused the collapse of megalithic tomb-building societies. In my opinion the authors' (still impressive) study of a human remains from a single megalithic tomb in the Paris Basin cannot sustain such an unequivocal and sweeping conclusion. The continental-scale framing of the significance of the results from Bury is, I think, unjustified and unnecessary. There is absolutely a wonderful paper in here which sets out quite a compelling narrative of the use of monument through time and the differing natures of the societies that used the tomb to bury their dead. Certainly given how much ink has been spilt on the reverence of ancestors in the European Neolithic, the finding that a tomb was remodelled and reused by a community that was largely not descended from the original builders is a fascinating and novel counterpoint. Similarly, the authors' study adds to the evidence for regional deviations in funerary behaviour in different tombs/regions, potentially suggesting that each community of tomb users

organised themselves slightly differently, a tantalizing insight, However these aspects are underexplored in favour of a bigger narrative of disease-driven decline in Neolithic Europe.

I would certainly agree with the authors that their results from Bury are consistent with the 'Neolithic decline' scenario the authors describe, but given the low detection rate of *Yersinia pestis* at Bury and other possible reasons for communities stopping burying their dead in specific tombs, I don't think the new data the authors present here in of itself provides strong evidence in favour of a pan-European 'Late Neolithic decline fuelled by plague. The catastrophic mortality profile identified by osteological analysis at Bury is interesting and provides important context for the disease results, but these patterns are only straightforwardly interpretable in cases where the community are well-represented in the burial assemblage. Given at Bury there are numerous reasons (male bias, occurrence of close genetic relatives) to think this is not a representative sample of a community, the catastrophic mortality profile is less easy to interpret and doesn't translate straightforwardly into rapid population rise or mass death.

Given the number of samples the authors have analysed from Bury, the rate of *Yersinia pestis* infection is very low and unless the authors can provide analysis to the contrary it looks to me like the varied occurrence in different phases could have occurred by chance. In addition, the factors affecting the within-site, between-phase and between-site (and sometimes intra-skeletal) preservation of pathogen DNA are poorly understood and it is difficult to know to what extent within-site and between site comparisons of rates of disease represent genuine variation in prevalence or just preservation. I agree that the authors findings of *Yersinia pestis* at Bury adds to the impression that this disease was widespread, but given, as the authors reference, it has already been found in Orkney, a location that is exceptionally geographically peripheral, Therefore, I don't think the authors' results from Bury necessarily adds so much to the discussion of how widespread plague was in Late Neolithic Europe.

I found the authors' discussion of the broader archaeological context of the 'Neolithic decline' and disuse of megalithic structures inadequate and simplistic. In the initial discussion it is unclear whether the authors are discussing megalithic structures generally (which would include things like stone circles and menhirs) or megalithic tombs specifically. Even if the latter, the situation is much more complex than the authors make out, with regional differences in use of megaliths and measure of demographic change. The obvious example is Britain and Ireland. In Britain, stone circles and Stonehenge specifically are being built during this supposed decline. Megalithic tombs are still being built in Orkney and Ireland into 3rd Millennium BC. I had wondered whether a more detailed discussion was included in the supplementary information. There was a well-written and interesting discussion of the development of Funnelbeaker-associated megalithic monuments, but it is unclear how this related to the Bury tomb specifically. The authors make no effort to understand their results in their local or regional contexts. Given the regionality in markers of a Neolithic demographic decline, it would be interesting to interrogate how the results from Bury match regional trends, but this aspect is neglected in favour of grand narratives of disease and decline across Europe. The authors do allude to this regional variability, but this is very cursory.

The authors' discussion also seems to conflate, the 'Late Neolithic decline' which is usually derived from measures of demographic decline with changes in the disuse/abandonment of megalithic tombs. I think it is reasonable to argue that the disuse of megalithic tombs is related to population decline, but given there could be lots of reasons why communities stopped using these monuments to bury their dead I don't think the authors can take this connection for granted. While their data and analysis are excellent, I don't think the authors have presented any definitive evidence for population collapse at Bury, only that one groups ceased using the tomb to bury their dead.

In sum, I don't think the results from Bury presented by the authors are enough to substantially tip the balance in discussions of the role of plague in the Neolithic decline and the abandonment of megalithic tombs in the way the authors argue. Normally I would say that the paper needs a major revision and reframing, but without the grander applications to disease and demographic decline I'm not sure a revised version of this paper would be well-suited to Nature Ecology and Evolution and perhaps would be better placed elsewhere.

Some more specific comments related to the above:

Page 2, Line 1: Around the Late 4th Millennium'..., a significant population decline in Europe and halt in megalith building occurred...'

I appreciate this is only part of a summary, but this is too simplistic and should be toned down significantly. Megalithic building in Britain, for instance, continues well into the 3rd Millennium BC, for instance. If the authors are only referring to tombs (which they don't specify clearly here or later), than this is also not accurate – e.g. developed passage tombs in Ireland and Orkney. This statement is also somewhat contradicted by their own paper given that Bury is a megalithic monument used and remodelled in 3rd Millennium BC.

Page 2, Line 2: 'Recent research that this 'Neolithic decline' linked to a plague outbreak and shifts in societal structure and genetic ancestry.'

This sentence is incomplete. Also 'associated with' rather than 'linked to' might be a more appropriate phrase here as 'linked to' implies some level of causality, which is debated.

Page 2, line 4: 'To investigate, we sequenced 133 ancient genomes from the French semi megalithic allée sépulcrale at Bury, which spans two burial phases—...'

I think it would be useful for the authors to include which specific region of France. Also the authors could consider whether 'French' works when discussing a Neolithic monument rather than 'in present-day France.'

Page 2, Line 7: 'Our analysis revealed societal changes and two distinct genetic groups between phases.'

I think it's always important in these sorts of studies to acknowledge that these are investigations of organisation amongst the dead, not the living. While certainly one is likely to have implications for the other, there is not necessarily a straightforward link. The authors find evidence for changes in the relationships between people buried in the tomb in different phases. I think there is a reasonable case to be made that this difference reflects differences in social organisation but I don't think the authors can say that their study directly reveals societal changes, rather evidence for differences in deposition and organisation which might reflect changes in society. I think it is necessary for the authors to lay out this reasoning, otherwise the assumptions they are making are not so transparent.

Page 2: 'The end of complex megalithic architectures in northwestern Europe has never been properly explained. Radiocarbon dates indicate a period of construction for these collective tombs around 4,300-3,100 BCE, followed by a pan-European decline of burial activities between 3,000 BCE to 2,900, depending on the area (Hinz et al., 2012; Shennan et al., 2013). These tombs are numerous throughout northwestern Europe, with very high concentrations in places such as the Paris Basin, Central Germany and southern Scandinavia. Many possible reasons have been put forward as contributing to this large-scale 'Neolithic decline'. Among the most prominent theories is that environmental exploitation brought about by farming, such as soil degradation and deforestation, reduced the land's capacity to support agriculture and livestock and thus its ability to support local populations (Colledge et al., 2019). Others argue that the reason behind the decline likely falls within the societal realm, including for example the strengthening of the idea of "households" (Furholt, 2021). Still others suggest that the close contact between humans and animals in the Neolithic increased the risk of pathogen emergence, which, together with the increased population density, increased the risk of transmission (Barrie et al., 2024; Rascovan et al., 2019).

I don't think this introduction to the Neolithic decline and the decline in the construction of megalithic tombs is adequate. It is too general and simplistic. The authors start by discussing the 'end' of complex megalithic architectures in northwestern Europe, but wouldn't this definition include megalithic sites which continue into the 3rd Millennium BC (stone circles/alignments, Skara Brae)? It becomes clear only later that they are referring to megalithic tombs specifically, but 1) this needs to be made clear from the beginning, and 2) this still doesn't work for places like Ireland and Orkney where megalithic tombs are being built into 3rd Millennium BC. I appreciate the authors acknowledge that evidence for 'decline' depends on the area, but I feel as though this variability is significantly underplayed in favour of depicting a broad pan-European trend of decline. Perhaps the authors are focussing on continental northwestern Europe to the exclusion of places like Britain and Ireland, but if so they should state this directly and qualify it when they are referring to 'pan-European' or references which discuss the Neolithic decline in Britain and Ireland as well as in continental Europe.

Moreover my understanding of the Neolithic 'decline' is that it primarily derives from and is defined by evidence of demographic change, with other factors such as megalithic monuments falling out of use seen as possible effects of this demographic change. However the way the authors have chosen to phrase this in their manuscript seems to begin with is the drop off in construction of megalithic monuments as indicative of some 'decline' in of itself. I don't think this is a reasonable position to hold given there could be lots of reasons why people might switch away from constructing megalithic tombs which doesn't involve a 'decline' of any sort, if for instance there was simply a shift in cultural preference. The way the authors bundle together demographic decline and the move away from the construction of megaliths takes for granted that the two are somehow inextricably linked and causal. The argument that demographic decline may be responsible for less monument building is reasonable, but the authors need to outline the reasoning for this, it can't be simply taken for granted and the way that it's expressed here puts the cart (reduction in megalithic tomb-building) before the horse (demographic change).

Given the authors acknowledge that measures of Late Neolithic decline and monument building are regionally variable across Europe, it is surprising they don't provide an account of decline in the particular region (Paris basin) where their tomb was found. It would be much easier to assess the narrative they propose for the Bury site if we had an understanding how that fits within the local, regional sequence rather than how it might vaguely fit into a pan-European phenomenon. Again, it feels like the local/regional contextualisation of the (extremely interesting and important) results are neglected in favour of serving a grand narrative.

Page 2: 'The presence of *Yersinia pestis*, the etiological agent of plague, in the Bronze Age is well established from ancient DNA research and was, until recently, the earliest known form of the pathogen (Andrades Valtueña et al., 2022; Rasmussen et al., 2015). However, the subsequent identification of earlier lineages of *Y. pestis* has demonstrated that plague was afflicting Neolithic communities already before the expansions from the Eurasian steppe (Rascovan et al., 2019).'

I don't think the various factors of the debate around Late Neolithic-Bronze Age *Yersinia pestis* are elaborated well here, and it means that the introduction of human expansions from the Eurasian steppe comes across as a bit of a non sequitur without a certain amount of prior knowledge on the part of the reader. I think the authors should add some detail about the fact that initially the Bronze Age *Yersinia pestis* lineages seemed to coincide with influence human expansions out of the Eurasian steppe, which in turn was used to suggest that these expansions had something to do with the appearance of this disease,

but that since then older plague genomes have shown that it was present prior to these movements off the steppe.

Page 2: 'As the age of these genomes coincides with the end of the fourth millennium BCE, it has been suggested that a possible plague pandemic could have contributed to a population collapse around 3,100 BCE (Rascovan et al., 2019).'

Again, this is treating the Late Neolithic decline as a singular homogenous pan-European process when I don't think there is the evidence to take that for granted.

Page 3: 'However, it remains unknown whether this regional outbreak was part of a larger pandemic across Europe during the megalithic period.'

The authors later mention the detection of *Yersinia pestis* DNA in a skeleton from the Banks tomb in Orkney. Why is this not mentioned here as background information? Wouldn't the (geographically) peripheral location of Orkney mean that it is already established to some extent that *Yersinia pestis* must have been widespread in this period, even if this could be because of regular zoonotic spillovers rather than a pandemic?

Page 3: 'The semi-megalithic allée sépulcrale of Bury, located 50 km north of Paris, is from a region where many other collective graves have been recorded (Chambon & Salanova, 1996).'

Given the centrality of megalithic architecture to this paper, it would be useful for the authors to define what they mean by 'semi-megalithic' here, given it is not a common phrase.

Page 3: 'The first phase at the end of the fourth millennium (3,500 - 3,000 BCE) was interrupted shortly after its beginning (Salanova et al., 2018), while the phase covered several centuries over the third millennium (2,900 - 2,470 BCE).'

Is there a missing 'second' here? '...while the second phase covered...'

Page 3: 'Rather, the demographic profile is suggestive of excess mortality, particularly affecting juvenile individuals, perhaps indicating a catastrophic event, such as war, famine or a disease outbreak or, on the contrary, a rapid increase in the population. Hinz specifically points to decline from around 3350 cal. BC in northwestern Europe, which is well before the first phase at Bury.'

I think it would be useful if the authors could comment in slightly more detail on the nature of the deposits of human remains at Bury. Do they have a sense of whether whole bodies were left to decompose in the tomb before being disturbed and commingled, or is there any indication some disarticulated bones were brought in from elsewhere? I think this is important in understanding how much selection was involved in whose remains ended up in the tomb. The catastrophic mortality profile is interesting here and relevant to the discussion of plague, but it is only easily interpretable if the people buried in the tomb were representative of who was dying more generally. I think it would be useful for the authors to caveat that if there was some selection of bones/individuals then the meaning of the mortality profile is difficult to discern.

Page 6: 'These results suggest that the individuals of phases one and two form genetically distinct communities.'

As the authors state later that they couldn't rule out minor population continuity between phase 1 and 2, is genetically 'distinct; a bit too absolute? Genetically differentiated maybe?

Page 8: 'We use them to describe biological relatedness, and do not imply that these terms, nor the notion of family, was understood in the same way by the population using the grave at Bury during the Neolithic times.'

I feel as though there is something missing from this sentence in terms of anachronistic project of present-day understandings of family. Perhaps '...was understood by the population using the grave at Bury during the Neolithic times in the same way as it is understood in Western societies today.'

Page 8: 'This ties in with the observation that there are no exogenous individuals in the grave.'

I found this sentence and the passage before it a bit confusing. Earlier the authors state that 'we find a complete lack of any exogenous individuals (aside from the first generations), implying that, other than the totally unrelated individuals, only people genetically descended or linked to others are buried in the grave.' How are the authors defining 'exogenous individual' here? By 'exogenous' do they mean people from outside the genealogy that had children with people inside the genealogy? If these people do exist in the first generation then how does this support the statement that there are no exogenous individuals in the grave? Are totally unrelated individuals not 'exogenous'? This needs clearing up.

Page 8: 'The two sequenced sons of ID222, ID291 and ID275 both show long runs of homozygosity (Supplementary Figure 7.5), indicating some possible relatedness between their mother (unsampled) and father ID222. One of these sons, ID291, has a further son (unsampled), who conceived three children with one female, ID262, who is a third or fourth degree relative to ID291 - her reproductive partner's biological father. One of the sons of this union (ID316) also exhibits long ROH segments. Incidentally, ID262 is also the only female buried in the grave who has both ancestors and children also buried at Bury.'

These are some tantalizing findings especially as high parental relatedness seems to be relatively uncommon across Neolithic Europe (Ringbauer et al. 2021). It's a shame that this is not discussed further in the Discussion section, again it feels as though this was neglected to focus on the grander narrative. Also, doesn't the heightened frequency of parental relatedness in phase 1 compared to phase 2 run contrary to there having been a significant pan-European population collapse? Wouldn't increased frequency of related parents be more likely in a smaller population? Unless the authors think this was more to do with social organisation in phase 1 compared to phase 2.

On a similar point, did the population from phase 2 show higher frequencies of shorter runs of homozygosity than phase 1, indicative of a smaller effective population size? This is what we might expect to see if there had been a substantial population decline. I suppose the authors could argue that the population in phase two was 'new' and moved in from a different region, but if the authors are trying to support a pan-European model of population decline, this shouldn't make that much difference.

Page 10: 'The predominance of a large genetic group during the first phase, probably related to the other small groups, testifies to the control of a group on the collective grave, or to the links that could exist between the inhabitants of the same place.'

Being slightly picky – 'control' implies a particular interpretation which I don't think the authors have justified here. Other situations could easily be imagined where individuals from these genealogies were interred in the tomb by consent of a wider community or indeed chosen to be interred there by a different group who controlled access to the tomb.

Page 12: 'Furthermore, the grave only represents a subset of the population, and it is perfectly possible that a severe plague epidemic would leave very little evidence behind, if the entire population perished without being buried or if they were buried elsewhere.'

This is a reasonable comment to make but rather implies that the evidence the authors have found at Bury is not substantial enough to support the scenario of a widespread plague pandemic by itself and requires some extra justification. Moreover, I think there really needs to be more discussion of the uncertainties involved with the recovery of pathogen DNA. Granted, it's likely that there are a larger number of false negatives, but how much larger may be highly dependent on context/site/regionally-specific preservation dictated by factors that are not well-understood. So, I think it is much more difficult than the authors convey to interpret what rates of particular pathogens mean when comparing sites or even phases.

Page 14: 'These factors collectively point to what should be termed a Neolithic Decline or Dark Age. It is also evidenced in a decline in the production of fine pottery in Northern Europe, and other technological skills.'

'Dark Age' is an extremely contentious phrase even when applied to its original early medieval context. It is commonly regarded to relate to the lack of textual sources rather than grim circumstances of people who lived through it (although it has since come to be associated with that aspect too). Periods of prehistory can be described and discussed on their own terms without having to reach for phrases from totally different periods/contexts. Therefore, 'Dark Age' is an entirely inappropriate and anachronistic term to use here and I think the authors should strongly reconsider their using it. In addition 'should be' sounds very prescriptive ('could be' might be more appropriate) – it falls to the broader scholastic community to decide whether 'Neolithic decline' is a useful term. The reference to the production of fine pottery and 'other technological skills' is an interesting component but given this is the first mention of this in the context of the Neolithic decline, it is extremely vague as well as being unreferenced. I think it certainly should be discussed in more detail alongside the decline in tomb-building, properly referenced, and potentially in the Introduction, rather than brought up off-handedly in the Discussion.

Page 14: 'There is also evidence from megaliths in northwestern France and central Germany that burials ended around 3,100 BCE.'

This requires a citation.

Page 14: 'All the observations point to a coherent narrative. An important event occurred in Northwestern Europe at the end of the fourth millennium. The collective burials, essentially megalithic, correspond to a demographic increase and a high population density (Salanova & Chambon, 2025), which stopped around 3,100 BCE. After a couple of centuries of interruption, the use of the Bury grave during the whole third millennium is different, both in ritual and population structure. The population decline in the collective grave and the plague pandemic could have made it easier for steppe populations to colonize new lands westwards after 2,900 BCE.'

I think it's reasonable to make this argument, but I think most of the evidence the authors discuss here is largely already published and the novel results from Bury are too ambiguous to substantially support the strong and wide-ranging conclusions the authors come to here.

References

Ringbauer, H., Novembre, J. and Steinrücken, M., 2021. Parental relatedness through time revealed by runs of homozygosity in ancient DNA. *Nature communications*, 12(1), p.5425.

Reviewer #4 (Remarks to the Author):

The manuscript by Ramsøe et al. analyzes 133 genomes from the Bury megalith site in the Paris Basin. The archaeological and anthropological integration is very solid and effectively contextualizes the results, especially regarding the biological kinship. The laboratory methodology and sample quality are excellent. However, some methods require a more detailed description and the application of additional statistical analyses, although they are generally adequate to support the discussion of the results.

Nevertheless, it seems that the objective of this case study is to argue for a population decline at a European scale. While the case study is highly interesting and of sufficient quality for publication, suggesting that the phases of a single burial site define the scope of a population decline at the continental level is a major claim. This is a hypothesis that can certainly be discussed but not asserted with the degree of certainty presented here.

Below are specific observations by section. For resubmission, it would be advisable to include line numbering.

INTRODUCTION

In the sentence:

“Radiocarbon dates indicate a period of construction for these collective tombs around 4,300-3,100 BCE, followed by a pan-European decline of burial activities between 3,000 BCE to 2,900, depending on the area (Hinz et al., 2012; Shennan et al., 2013).”

- Specify whether the mentioned decline refers exclusively to “collective burial activities” or to funerary activity in general.

In the statement:

“These tombs are numerous throughout northwestern Europe, with very high concentrations in places such as the Paris Basin, Central Germany and southern Scandinavia.”

- Indicate the cultural affiliations associated with these tombs for greater precision. Without a clear specification of the type of megalithic tomb, this statement could also apply to regions such as southern and western France, as well as the Iberian Peninsula, where such structures are also widely documented.

RESULTS

The site shows significant similarities with Fleury-sur-Orne (Rivollat et al., 2022), both in the predominance of males and in the diversity observed in Y-chromosome haplogroups and certain levels of background relatedness, based on the accumulation of low ROH levels. Since Fleury predates both phases of Bury, it would be advisable to integrate these analyses with the new data obtained from Bury for a better comparison.

- Figure 2a: The Kernel density calibration curve for phase 1 and phase 2 shows some overlap. What specific results support the existence of the 100-200-year chronological gap mentioned in the abstract?
- Figure 1: The image is too large for the amount of information it provides. It is suggested to reduce its size and combine it with a geographical map that includes the location of Bury and other similar sites or those co-analyzed in the study.
- Phase 1 seems to exhibit different levels of HG ancestry compared to phase 2, which could indicate a more recent admixture date. The authors are recommended to use DATES or similar software to assess whether phase 1 shows a more recent admixture with HG compared to phase 2, where the HG ancestry proportions appear more homogeneous in the population. This analysis could help determine whether there was no additional HG contribution in the second phase, which in turn could reflect a decline in the final HG population.
- The article could benefit from a supplementary PCA that includes other Neolithic individuals from France, Germany, and southern Scandinavia. The distribution of these populations in the PCA would help identify which group is genetically closest to the individuals from Bury.

IBD RESULTS

- Clarify which individuals were added to the customized dataset on which the GLIMPSE imputation was applied.
- Discuss the criteria for the clustering levels: are they based on the length or the number of IBD segments? Additionally, in a broad-scale analysis, shouldn't close relatives be excluded to avoid biases?

In the statement:

“These results suggest that the individuals of phases one and two form genetically distinct communities.”

- It is recommended to rephrase this more cautiously. As mentioned in the study itself, in the first cluster there are individuals from both phases, and in the third clustering level, close relatives are grouped together. Even if they are different communities, this result cannot automatically be extrapolated to a population scale to claim continuity or discontinuity at population level. It is likely that if another Neolithic site from France were added, it would form its own cluster at the same resolution, without implying a population discontinuity.
- Instead of suggesting population discontinuity (here referred as “distinct communities” but later discussed as “population discontinuity”), it would be more appropriate to indicate that a different community made use of the megalith in a later phase. These are distinct concepts. Community discontinuity cannot be directly extrapolated to population discontinuity/decline. A stronger statistical justification for the clustering method is required. The following is recommended:

1. Include a matrix representing the total number of analyzed individuals from the sites considered showing the total amount of shared IBD.
2. Apply statistical tests to determine whether individuals within a cluster are significantly more related to each other than to individuals in other clusters.
3. Even if a significant pattern is observed, its interpretation should be nuanced, avoiding conclusions at the population level, as these groups may belong to the same source population.

Correction in supplementary figures: All figures in Supplementary 5 and Supplementary 6 are numbered in the same way

(Supplementary Figure 6.1 to 6.7), which causes confusion. Additionally, Supplementary Figure 7.4, mentioned in the text, does not exist.

ANCESTRY MODELLING

- Specify whether the ancestry modeling is based on qpAdm or supervised ADMIXTURE, as this information is missing from the methodology.
- Explain why only Iberian Peninsula HGs were chosen as a proxy. These individuals retain more Pleistocene-Magdalenian-associated ancestry than Central European HGs, but the study focuses on the Paris Basin, Central Germany, and Southern Scandinavia, where this ancestry barely persists.
- To better interpret the origin of the HG ancestry resurgence, it is suggested to perform models using local HGs from these regions and compare the fit of both models. This would help determine whether the HG ancestry comes from local groups or from Neolithic populations with a higher proportion of HG ancestry.
- A statistical test should be included to determine whether the increase in HG ancestry between group 1 and group 2 is statistically significant.

Genetic Relatedness Results

- It is necessary to cite/discuss that no multiple sexual partners were identified in the analysis, similar to Gurgy (Rivollat et al. 2024), because it is not always the norm (e.g in Hazelton).
- Clarification on exogenous individuals:
 - o The main text states that "we find a complete lack of any exogenous individuals (aside from the first generations)", but Supplementary Figure 6.3 includes individuals not connected to the pedigree. Are these individuals not considered exogenous? It should be clarified what exogenous individuals means.
- Strontium (Sr) data:
 - o Sr data are presented in the Supplementary, but they are not mentioned in the main text. If they are not used in the analysis, they should not appear in the Supplementary, or alternatively, they should be integrated into the discussion to assess the presence of non-local individuals.
- Errors in Supplementary Figure 6.3:
 - o Some mitochondrial haplogroups are not transmitted to the next generation, suggesting that certain pedigrees are incorrect (this occurs in pedigrees 2A, 2B, and 2G). It is recommended to review mitochondrial transmission consistency and correct erroneous structures.
 - o The legend for Figure 6.3 should be more descriptive, clearly specifying what circles and squares represent and how mitochondrial and Y-chromosome haplogroups are marked. The age at death is a crucial data point that should be added to the pedigree.

Interpretation of the Female Exogamy Model

- The main text states: "All females in the first phase except one have no offspring that are buried in the grave. This could imply a practice of female exogamy..."
 - o This statement is correct; however, if many women appear at the end of pedigrees, it could be due to pre-reproductive mortality rather than a female exogamy model. To exclude this possibility, it is recommended to include individuals' age at death and discuss how this affects the interpretation of the exogamy model.
 - Sex distribution in the grave:
 - o The predominance of males could be discussed in relation to other sites, such as Flury, where a burial bias toward males has also been documented.
 - Definition of exogenous individuals:
 - o The statement "This ties in with the observation that there are no exogenous individuals in the grave" needs more precise explanation.
 - o In female exogamy contexts, many women are absent from pedigrees because their descendants are not buried at the site. However, in this case, more men fall outside the pedigree, which does not fit a classic reciprocal female exogamy model.
- #### Runs of Homozygosity (ROH) and Kinship Analysis
- It is mentioned that the children of ID222 (ID291 and ID275) exhibit long ROH segments, suggesting a possible consanguineous relationship between their mother (not sampled) and their father (ID222).
 - o It would be useful to quantify the degree of relatedness required to generate this ROH percentage. It is recommended to use HapROH or similar to estimate the most probable kinship between the parents.
 - o Since the parents do not share the same mitochondrial haplogroup, it would be relevant to analyze whether their relationship is through the paternal line and determine the most plausible kin relationship based on ROH and haplogroup data.

Discussion

- o The following statement is too ambitious for an analysis based on a single site and should be moderated: "The data from the Bury grave is the support that was lacking until now to propose a credible hypothesis regarding the population dynamics at the end of the fourth and third millennia BCE, including the 'Neolithic Decline'."
 - o It is recommended to rephrase this statement to reflect that the study contributes to the debate but does not constitute definitive proof regarding population dynamics in this period.
 - o The sentence "Given that in Bury a temporal hiatus of 100-200 years separates the two phases of use, and supporting genetic evidence for the discontinuity of a single population, we can tie this into a wider European picture in order to trace potential signals of demographic decline in the late Neolithic"

Assumes a strong population discontinuity, whereas the results do not show a clear break between the two phases.

While it is valid to discuss this issue, the text should reflect greater caution and avoid overgeneralizing a case study to a large-scale phenomenon.

It is necessary to reformulate both claims to indicate that the Bury data provide valuable evidence on potential demographic processes in the Late Neolithic but without presenting conclusions as definitive proof of a pan-European phenomenon.

*****END*****

Version 1:

Decision Letter:

14th October 2025

Dear Dr Sikora,

First, I'd like to apologise for the delay in getting the second round of reviewer reports on your manuscript entitled "Population Discontinuity Across the Neolithic Decline in the Paris Basin" back to you. This delay was caused by the absence of reviewer 2. However, since their expertise was primarily in ancient plague genomics, and emphasis on this has now been downplayed in the interpretation, we have decided to proceed with the advice of the remaining three reviewers, whose comments are attached. As you'll see, while they are pleased with the progress the manuscript has made, they still have a number of concerns which will need to be addressed before we can offer publication in Nature Ecology & Evolution. We will therefore need to see your responses to the criticisms raised and to some editorial concerns, along with a revised manuscript, before we can reach a final decision regarding publication.

We therefore invite you to revise your manuscript taking into account all reviewer and editor comments. Please highlight all changes in the manuscript text file.

* If you have not done so already please begin to revise your manuscript so that it conforms to our Article format instructions at <http://www.nature.com/natecolevol/info/final-submission>. Refer also to any guidelines provided in this letter.

* Extended Data Figures - please ensure that any supplementary figures and tables that are crucial to the manuscript's conclusions are converted into Extended Data figures and tables to increase visibility of these data. Extended Data figures and tables are online-only (present in the online PDF and full-text HTML versions of the paper), peer-reviewed display items that provide essential background to the article but are not included in the main article due to space constraints. A maximum of ten Extended Data display items (figures and tables) is permitted.

Link Redacted

Nature Ecology & Evolution is committed to improving transparency in authorship. As part of our efforts in this direction, we are now requesting that all authors identified as 'corresponding author' on published papers create and link their Open Researcher and Contributor Identifier (ORCID) with their account on the Manuscript Tracking System (MTS), prior to acceptance. ORCID helps the scientific community achieve unambiguous attribution of all scholarly contributions. You can create and link your ORCID from the home page of the MTS by clicking on 'Modify my Springer Nature account'. For more information please visit www.springernature.com/orcid.

[redacted]

Reviewer expertise:

as before

Reviewers' comments:

Reviewer #1 (Remarks to the Author):

Thank you to the authors for addressing the queries and issues raised in my first review. I now find the manuscript to be very strong and to contribute novel insights into European Neolithic social organisation and biology.

I only have a few minor typographical and editorial suggestions rather than further queries:

l. 119–120: Add a reference or additional information regarding the project “Time of Their Lives.”

l. 306: Add a reference to support the mention of Gurgy.

Throughout the text: Provide a clear definition of terms such as “full brother” (l. 309) and “extended family.”

Reviewer #3 (Remarks to the Author):

The other reviewers and I set a significant task for the authors in terms of reorientating their paper and providing better justification for their conclusions, but the authors have risen to the challenge impressively. In my view, the new focus of their paper on site-specific population change which potentially fits into broader regional population changes, possibly linked to movement of people out of Iberia is, to my mind a more significant finding than the focus on plague and demographic decline in the previous version of the manuscript. Their conclusions are much more robustly supported by their improved and extended analyses than in the original version. However, while the manuscript is significantly improved I still think there are a few areas of interpretation that are overreaching and should be caveated or which require more clarification.

Firstly, given the original manuscript set about establishing the evidence for a ‘Neolithic decline’ in Central/Northwestern Europe, it is odd to me that the title and the introduction to this new paper rather seem to take the ‘Neolithic decline’ for granted. Particularly the sentence on Line 61 ‘At the end of the fourth millennium BCE a demographic decline decimated populations across northwestern Europe³’. I don’t think there is evidence yet to support this assertion so unambiguously. The decline in megalithic tomb building could be down to population decline, but it could also be down to change in preferences. The pollen and summed radiocarbon dating data is regionally variable and could also be related in changes in subsistence which may or may not be related to demographic decline. All these things taken together provide good evidence that there was a decline, but I think here in several places the paper takes this as something that is known rather than something there is evidence for but is under discussion. This attitude towards the Neolithic decline is also at odds with their own argument in their ‘Environmental data on the Neolithic decline’ section: - if the fact of a decline is already as established for northwest Europe as the authors make out, why is further analysis required? I think the authors should be more cautious and instead build up the three factors (cessation of megalithic tomb building, pollen data, summed probability distributions of radiocarbon date) as evidence that they interpret in terms of a Neolithic decline at the beginning of 3rd Millennium BC and which they add to with their pollen data (and in particular with regards to the Bury tomb). I understand that the authors are limited in terms of words for the title but as the ‘Neolithic decline’ is something that does not yet have a universally accepted definition and is not yet generally accepted I think the authors should try and avoid it in their title. Maybe something like ‘Population Discontinuity across the late 4th/early 3rd Millennium BC in the Paris Basin.’

Part of my criticism about the last version of the manuscript was that the authors often spoke of some of the observations they see as part of the Neolithic decline on a Europe-wide scale, but they can be regionally variable, particularly in places like Britain and Ireland. The authors have responded to these criticisms appropriately, but I feel they still haven’t gone far enough at times – there are still suggestions in places (discussed below) that they are discussing a Europe-wide phenomenon rather than one which might be more restricted (by their terms) to Central and Northwestern Europe. Despite some edits, I still don’t think it is so clear throughout that they are excluding places like Britain and Ireland or Iberia, particularly when they often talk about ‘northwestern Europe’, which by most definitions would include Britain and Ireland. Perhaps northwestern/Central continental Europe would be a better term to use.

In several places the authors discuss the shift in genetic ancestry of the people interred in the tombs in absolute terms, i.e. that the people from the second phase were entirely distinct from those in the first phase. While this could be true, the authors themselves admit they cannot rule out a small amount of continuity from earlier populations, therefore I think they need mediate their language accordingly in places. In addition, the authors present convincing evidence for a substantial shift in ancestry in northwestern Europe at the beginning of 5th Millennium BC towards Iberia, which they interpret as indicating substantial northward movements of people. However, given during the first phase of activity at Bury there were

already people in the Paris Basin with a similar Iberia-related ancestry profile, and alternative explanation could be that local people carrying these Iberian-associated ancestries come to predominate (for whatever reason) rather than substantial movements of population from Iberia. This is my understanding particularly from looking at Figure 3, but if I'm missing some complexity articulated elsewhere, I could be wrong!

This issue is somewhat compounded by the authors switching between discussions of changes in population and changes in ancestry. I appreciate the two are intimately connected, but to a naïve reader a shift in population (particularly when discussing population replacement) and a shift in genetic ancestry are quite different things. I'd recommend the authors discuss changes/shifts in ancestry rather than population as this is closer to what they are measuring and will align more with what people without a background in population genetics take from those terms.

The authors' focus on local and regional population shifts is very interesting and novel, however, as much as the authors convincingly outline the differences between the two groups interred in the tomb, to me it begs the question of why then they decided to inter their dead in a tomb alongside people who they were not directly genetically descended from and who had different funerary practices. This is especially true given the authors themselves discuss people interred within the tomb in both phases as having been regarded as kin to some degree. Does this not also infer that the people who interred remains in the second phase in some way regarded them to be kin in some more abstract way of the people interred in the first phase? I'm presuming here that at least some of the human remains from the first phase would have been visible to the people who interred remains in the second phase. I think this is a salient point to explore given how much is often made of Neolithic 'ancestors' and their veneration within megalithic tombs. In the case of Bury it would appear in the second phase that the idea of ancestral connections were either unimportant or potentially invented. It also complicates the common archaeological assumption that continuity or revival of a practice is evidence for regional continuity of people. It may be I've missed something about the archaeology of Bury which suggests that the human remains from either phase were regarded separately by the people who interred bodies in the second phase, but if so I think the authors need to make this clearer.

Some specific comments related to the above:

Line 39 'At the transition between the third and the fourth millennium BCE, a significant population decline and halt in megalith building occurred across Northwestern Europe.'

There is evidence for population decline (halt in megalithic tomb building, pollen evidence for changes in forest cover) but the extent to which particularly the megalithic tomb building reflects a decline in population rather than a shift in cultural preference is debated so I don't think it can be said straightforwardly that there was a significant population decline. – '... there is evidence for a significant population decline...'

Line 40 'In Scandinavia this "Neolithic decline" instigated a massive population turnover, as farming communities disappeared and were replaced by people with steppe ancestry. In Western Europe, however, farming ancestry persisted beyond the Neolithic decline, and it remains unclear whether it was accompanied by a similar demographic replacement.'

While plausible, I think it's still unclear whether the 'Neolithic decline' instigated massive population turnover so I think the authors should be more circumspect here. Steppe-related ancestry comes to predominate in places where there is more and less evidence for a 'Neolithic decline' therefore it's unlikely shifts towards steppe-related ancestry were only down to this. In addition I understand what the authors are getting at with the term 'Farming ancestry', but there is a lot of assumed knowledge wrapped up in it and it could come across as a little bizarre to any archaeologists reading the paper, after all Neolithic farming groups often carry ancestry from early hunter-gatherer groups and people carrying the steppe ancestries were probably farmers to some extent too. I think the authors could be more precise here, perhaps discussing 'ancestry from preceding farming communities'. I also think the authors could be more careful in talking about 'population turnover' and 'replacement'. I appreciate these are technical terms which reflect something specific related to ancestry change, but again going back to that point of precision, we see shifts and replacement in genetic ancestry not necessarily literally populations of people. Talking of local populations 'disappearing' also implies something rapid and immediate, when again I don't think the authors can be confident it was so total and rapid.

Line 50: 'Our analysis revealed that the two burial phases at Bury represented distinct demographic patterns, correlated with discontinuous genetic groups.'

It is unclear to me what 'demographic patterns' means here. Do the authors mean the genetic relationships between individuals in the tomb or the different age-at-death profiles? I think the authors should consider rephrasing. Also again I think 'discontinuous' here could be misleading given there might be some small, continuity from the people in the first phase. Line 51: 'Furthermore, we show that the difference between the two burial phases can be linked to a northwards movement of Neolithic ancestry from the south which only spread into the Paris Basin after the Neolithic decline'

Again, I don't think the 'Neolithic decline' – its extent, temporal and geographical distribution is well resolved so I think it would be better for the authors to stick to dates and associations with other events (decline in tomb building, pollen record) rather than relying on taking the 'Neolithic decline' for granted.

Line 61: 'At the end of the fourth millennium BCE a demographic decline decimated populations across northwestern Europe'

I think this is too unequivocal given it is an interpretation which not everyone subscribes to. In order to make clear the

exclusion of the Britain and Ireland from their discussion, it might be worth the authors referring to 'continental northwestern Europe' as some definitions of 'northwestern Europe' would include the North Atlantic archipelago.

Line 78: 'Among the most prominent theories is that environmental exploitation brought about by farming, such...'

There seems to me to be a step of reasoning missing here – the authors take the decline in megalithic tomb building as evidence for demographic decline directly, when this could simply be a shift in cultural behaviour. I think it's reasonable for the authors to discuss the decline in tombs in terms of demographic decline but they need to make that link and acknowledge other possible interpretations. For instance, wouldn't demographic decline with no change in cultural preference see just steadily fewer tombs being built rather than an abrupt halt?

Page 86: 'The presence of *Yersinia pestis*, the etiological agent of plague, in the Bronze Age is well established and the disease was, until recently, assumed to have been spread by the migration from the Pontic steppe into Europe^{18,19}'

'...in Bronze Age Eurasia' maybe

Page 88: 'However, the subsequent identification of earlier lineages of *Y. pestis* has demonstrated that plague was afflicting Neolithic communities already before the Yamnaya expansions¹⁶'

This is the authors' first mention of 'Yamnaya' whereas previously they had only discussed 'steppe ancestry' which could be confusing as well as potentially imprecise (i.e. Corded Ware is not Yamnaya). For consistency I think it would be better for the authors to stick to 'steppe ancestries', as not all groups carrying steppe ancestries are culturally 'Yamnaya'.

Line 127: 'This imbalance concerns both adult and non-adult individuals, and is incompatible with a natural population, thus suggesting differential burial treatment between males and females at Bury: for some reason more than half of the females in the community were excluded from being buried in the grave.'

This is a very interesting results. It would be useful to get the authors views on whether they believe this has implications for interpretations of similar patterns in terms of female exogamy at other Neolithic tombs – is general differential treatment of female burials a competing hypothesis in these other cases?

Line 145: ') . From our principal component analysis (PCA), we found that all individuals from Bury fell within the broader diversity of 'Neolithic farmers' (Figure 2c).

Neolithic farmers from where (broad region)?

Line 173: 'This finding provides further support for considering the individuals from the two phases as distinct genetic groups.'

Is 'distinct groups' entirely supportable here given the authors later suggest they can't rule out some level of input from the first phase?

Line 175: 'Having established that the two phases at Bury form distinct populations, we decided to explore whether these findings are compatible with either genetic continuity or discontinuity between the two phases (see Methods)'

Again, can the authors say confidently that these two groups were fully distinct from one another, i.e. no connection at all? If not I think the authors need to qualify this statement.

Line 182: 'However, the simulation does not exclude the possibility of some amount of gene flow between the two burial phases.'

I think therefore this means the authors cannot talk about discontinuity distinct populations so robustly. Even 'largely or possibly completely discontinuous' would be more appropriate because this expresses the uncertainty.

Line 200: 'At some point after 2,900 BC, a final northwards push of the Iberian ancestry partially replaces the existing population in the Paris basin resulting in the homogenous population we observe in Phase 2.'

The authors switch between talking about replacement of ancestry and replacement of populations. I appreciate that in some ways these two concepts are synonymous in population genetics, but I think 'population replacement' can be misleading and it would be more accurate for the authors to discuss 'ancestry' specifically, as that is what they are principally measuring.

Line 202: 'After the end of Phase 2, around 2,500 BC, steppe ancestry first appears in the Paris Basin²⁹ (Figure 3). After a couple of hundred years this steppe ancestry will reach the Atlantic coast and mix with local populations carrying the Iberian ancestry.'

Another example of the authors shifting between 'ancestry' and population. I think for the sake of clarity it would be better if they stuck to one term. Also, the way this is written, without mentioning that these ancestries were carried by people, makes it sound like ancestry was moving independently of populations! Discussing ancestries moving also doesn't make sense

when discussing mixing with local populations. I think it's fine for the authors to discuss these movements of ancestry in terms of movements of individuals/groups of people, even if the mechanism, nature, dynamics and scale of these movements are unknown.

Figure 3: An excellent plain, illustrative figure but doesn't it suggest there were already individuals/populations in the Paris basin with high Iberian Neolithic-derived ancestry during Bury phase 1? In this case rather than movement of population couldn't simply local groups with this ancestry somehow came to predominate either generally or in terms of deposition in tombs? Or are the individuals with predominantly Iberian ancestry from phase 1 different in some way from those in phase 2? If so, I think this needs to be made clear.

Line 236: 'On the other hand, all females except one are related exclusively to their offspring, suggesting a high level of female exogamy.'

Given there is clearly evidence for sex-biased selection of whose remains ended up in the tomb, couldn't it have been that simply females that were part of this family were often buried elsewhere (which could be because of exogamy, but could just be differential burial practice).

Line 238: 'Interestingly, the sex bias observed within families is not as pronounced for unrelated individuals.'

As this is referencing a point made a fair way back it might improve clarity to add '...the male sex bias...'

Line 239: 'This observation aligns well with reports from sites with high levels of female exogamy, where unrelated individuals tend to be dominated by females, perhaps representing females that never produced offspring, or whose offspring were buried outside of the tomb.'

Again, should this be caveated by the authors' observation that females of all ages were underrepresented, and that it might not have been simply about exogamy but where females from these families were buried?

Line 302: This is the only example of half-siblings in the dataset, which implies that having two reproductive partners was not the norm, or that the progeny from these unions were buried elsewhere, and thus also perhaps not socially accepted.'

It may be also worth mentioning that if this reflected misattributed paternity, they might have been assumed to have been full biological siblings.

Line 311: 'However, the higher fraction of totally unrelated individuals could speak to the monument following a different kind of organisation not restricted to a few extended families.'

My understanding is that there were unrelated individuals in the first phase too and so I'm not sure even in this phase deposition could be described as restricted to a few extended families. This would instead suggest that this is a shift in degree to which non-biological relatives were included.

Line 378: 'However, the grave only represents a subset of the population, and it is perfectly possible that a severe plague epidemic would leave very little evidence behind, if the entire population perished without being buried or if they were buried differently, presumably elsewhere, as were the case for many plague victims during the second pandemic^{37,38}'

I think the authors should consider whether this passage is necessary – it seems to me to be an argument from an absence of evidence.

Line 386: 'We found evidence of forest regeneration in the Paris Basin from 2,900 to around 2,600 BCE, which is typically linked to a decrease in human activity (Supplementary Information 2).'

Is this a little late to be associated with Phase 1 at Bury? Also, does this imply that the population in Phase 2 also were perhaps not doing so well if forests continued to regenerate? If the authors theory of the movement in of a completely new population are correct than this would imply the demographics of all populations are relatively volatile.

Line 402: 'Since the beginning of the twentieth century, many authors have argued for an influx from Iberia to Northwestern Europe according to the Bell Beaker pottery diffusion during the third millennium BCE (see Salanova 200043, fig. 3). It now clearly appears that this influx precedes the Bell Beaker phenomenon⁴⁴: our finding demonstrates that the builders and first users of the tomb were largely replaced by a Neolithic population coming from southern France and Iberia as early as the beginning of the third millennium BCE.'

As discussed above, can the authors be sure that this definitely involved people moving out of Iberia, rather than being derived from standing variation in the Paris basin or movement from somewhere more proximate (e.g. southern France)? The mention of the Bell Beaker phenomenon is confusing here – are the authors arguing that the subsequent spread of the Bell Beaker phenomenon from Iberia may have been because of contacts established by these early population movements? Do we see any cultural changes in the early 3rd Millennium BC which might signal reorientation of connections to southern France/Iberia? This feels underexplored archaeologically.

Line 409: 'population dynamics in the Paris Basin is characterised by the replacement with another group of Neolithic

people, who persisted for around 500 years until the arrival of steppe-related populations'

Again, 'replacement' here seems to imply that one group was totally replaced by another, which may have been true but I don't think the authors can say with total certainty.

Line 416: 'Tied in with our findings on the replacement of the population between Phase 1 and Phase 2, this suggests that newcomers did not immediately reuse the tomb'

Same as above.

Line 427: 'Phase 2, on the other hand, is distinguished by its longer duration, dynasty-like family structure and high number of unrelated individuals which is not found at any other Neolithic sites from the area'

Earlier the authors refer to a hereditary family structure rather than 'dynastic'. I think hereditary is more appropriate here as dynastic reads as 'rulers' or 'elites' which I don't think the authors are arguing for (or at least haven't presented the evidence for) here. Also, what number of Neolithic monuments of the area have had human remains extensively sequenced? Is it all that meaningful that this is different than the couple of other examples that have been done? There could be local variation.

Line 430: 'Given the temporal hiatus that separates the two phases at Bury, the supporting genetic evidence suggesting discontinuity between the phases, and the evidence of reforestation (Supplementary information 2), we can tie this into a wider European picture in order to trace potential signals of demographic decline in the late Neolithic.'

'Evidence suggesting 'substantial' discontinuity...'

Line 434: 'There is also evidence from megaliths in Northern Germany that ended around 3,100-3,000 BCE, followed by decreasing open land^{13,21,49,50}'

'...megalithic tombs...'

Line 437: 'When we add the anomalies in the demographic structure of the buried population in Phase 1 of Bury as well as other megaliths from continental Europe, there are further indications of demographic change (Supplementary Information 1). It supports a more widespread phenomenon covering most of central and Northwest Europe.'

'...as well as other megaliths from continental Europe...' - Does this include megalithic tombs outside of North-Central continental Europe. If not I think the authors should be more specific and precise about where they are referring to given the regional evidence for 'decline' is patchy.

Line 443: 'After a couple of centuries of interruption, the use of the Bury grave during the whole third millennium is different, both in ritual and population structure, as in most of Northwest Europe.'

Do the authors discuss the differences in ritual beyond the relationships between the burials and the demographic features. For instance, it looks from Figure 1 that there were differences in burial position between the phases (flexed versus extended) but I don't think this is discussed in the main text (apologies if I've missed it).

Line 452: 'We may thus consider the possibility that both the Iberian northward migration, and the expansion from the steppe were related responses to the Neolithic decline in temperate Europe.'

This could do with a bit more elaboration – what is it exactly about the demographic declines in these regions which may have provoked the expansion of people out of Iberia and the steppe? I think the authors need to lay out their reasoning here.

Supplementary Figure 6.17.

This is a fascinating figure. Firstly, did the authors include/could this be at all affected by the individuals from the first phase who show elevated levels of consanguinity?

I appreciate the authors use the figure effectively to make a specific point about the people in the two phases of Bury being derived from divergent populations but I'm curious whether the authors could make more of the potential dynamics on show here. For instance, I'm surprised the authors do not discuss the potential support it provides for demographic decline in the population represented in Phase 1, given addressing a comment about evidence for demographic decline from one of the other reviewers was the reason for generating the figure in the first place. Or do the authors think it's too easy to overinterpret here? There is some evidence from the trend (albeit with large confidence intervals) that the population in Phase 2 was also somewhat declining by the time they were buried at Bury after a period of relative growth coinciding with the decline in the phase 1 population. Alternatively, could this apparent 'growth' be down to admixture with the population derived from phase 1? Does this suggest a volatility in populations of northern/central continental Europe perhaps partly related to 'boom and bust' agriculture? Does the decline (or plausible lack thereof depending on how you interpret the confidence intervals) of the Phase 2 population have implications for understanding the spread of people with steppe-related ancestry (i.e. if population decline was not as acute, does this suggest a different specific process of ancestry change with steppe-related ancestry)? What might be the nature of the population decline in Phase 2 given the people from this phase do not show a catastrophic mortality profile? It may be I'm simply reading too much into this figure, but if I am, other people will to so it might be worth

anticipating this with a bit of text explaining why it can't be taken too literally beyond gauging divergent population histories. Also it would also be useful if the authors could briefly state in the caption how they extrapolated census population size from effective population size here.

Reviewer #4 (Remarks to the Author):

I would like to thank the authors for having redefined the scope of the manuscript, providing a valuable discussion on population decline at the end of the Neolithic without making major claims based on a single case study. It is also evident that a considerable effort has been made to improve the manuscript and to address the previous suggestions. I sincerely believe that the manuscript has improved in both quality and scientific discussion.

The mixture modelling results strongly support the arguments about this demographic decline and contribute significantly to the broader archaeological debate. In particular, they demonstrate a northwards spread of Neolithic Iberians, first moving into the Paris Basin after/during the Neolithic decline. However, I consider that some additional aspects deserve further discussion in relation to this topic:

- This Neolithic Iberian source might not have originated directly from Iberia but rather from the British Isles and Ireland, where this type of ancestry has already been traced in the Middle Neolithic (Olalde 2018 Nature, Brace 2019 Nat Ecol Evol, Cassidy 2020 Nature). Please integrate the chronological evidence suggesting the presence of Iberian ancestry in the British Isles and Ireland into your discussion of the ~2900 BCE arrival. I also suggest that, if your analyses corroborate similar ancestries, these regions should be included in your Figure 3.
- Please check the subclade distributions of Y-chromosome haplogroup I2a1 in Ireland, Britain and Europe as presented in Cassidy's 2020. This could also provide clues about the origins/contact populations of phase 1. A similar exercise could be carried out for Y-chromosome haplogroup H2a1 as described in Rohlach et al. 2021 Scientific Reports.
- The discussion of the Neolithic decline and the Bell Beaker influx could be strengthened. Currently, only the date of Iberian influence in the region (~2900 BCE) is clearly stated. In order to better contextualize this debate, I strongly recommend presenting together the chronological ranges of the three phenomena: the Neolithic decline, the Bell Beaker influx (including its possible source regions), and the Iberian influence in the region. Depending on which dates are considered valid for the Bell Beaker expansion, the time ranges might not be so different. This does not necessarily imply that your results support an Iberian origin of the Bell Beaker, but it could highlight the possibility of two independent yet contemporaneous/almost contemporaneous phenomena. I believe this part of the discussion would benefit from a clearer chronological framework.

Minor comments

- Lines 119–120: the "The Time of Your Life(s)" project is mentioned but not properly referenced. Please clarify what type of project this refers to.
- Figure 2 caption: please specify what the grey symbols in the PCA represent.
- Line 336: remember to include the final reference for "marçais unpublished".
- Line 369: provide the archaeological context and geographical location of the genome RV2039.
- Supplementary Figure 6.14 (mixture modelling barplot): the colours used for Iran Neolithic and Early Neolithic France are too similar and difficult to distinguish.

*****END*****

Version 2:

Decision Letter:

6th January 2026

Dear Dr. Sikora,

Thank you for submitting your revised manuscript "Population Discontinuity Across the Neolithic Decline in the Paris Basin" (NATECOLEVOL-25020492B). It has now been seen again by the original reviewers and their comments are below. The reviewers find that the paper has improved in revision, and therefore we'll be happy in principle to publish it in Nature Ecology & Evolution, pending minor revisions to satisfy the reviewers' final requests and to comply with our editorial and formatting guidelines.

If you have not done so already, please ensure that you also email us a completed copy of the Reporting summary :

Reporting summary: https://www.nature.com/documents/nr-reporting-summary.pdf

We are now performing detailed checks on your paper and will send you a checklist detailing our editorial and formatting

requirements in about a week. Please do not upload the final materials and make any revisions until you receive this additional information from us.

[redacted]

Reviewer #3 (Remarks to the Author):

I'm very grateful to the authors for preparing another comprehensive and detailed response to my comments and they've again put an impressive amount of work into their response. Generally I'm satisfied with this updated version of the manuscript.

'For the title we prefer to keep the term 'the Neolithic Decline', as we believe that the coupling of our data to this hypothesis is central for the paper and essential for evaluating the current evidence supporting this idea.'

I'm happy to leave this up to the Editor to decide - I don't have an issue with the use of 'Neolithic decline' in the title, only that I think it would be better if it was articulated in a way which makes it clear it is not something around which there is total consensus. e.g. 'Population Discontinuity associated with evidence for Neolithic Decline in the Paris Basin.'

'Line 40 'In Scandinavia this "Neolithic decline" instigated a massive population turnover, as farming communities disappeared and were replaced by people with steppe ancestry. In Western Europe, however, farming ancestry persisted beyond the Neolithic decline, and it remains unclear whether it was accompanied by a similar demographic replacement.'

While plausible, I think it's still unclear whether the 'Neolithic decline' instigated massive population turnover so I think the authors should be more circumspect here. Steppe-related ancestry comes to predominate in places where there is more and less evidence for a 'Neolithic decline' therefore it's unlikely shifts towards steppe-related ancestry were only down to this. In addition I understand what the authors are getting at with the term 'Farming ancestry', but there is a lot of assumed knowledge wrapped up in it and it could come across as a little bizarre to any archaeologists reading the paper, after all Neolithic farming groups often carry ancestry from early hunter-gatherer groups and people carrying the steppe ancestries were probably farmers to some extent too. I think the authors could be more precise here, perhaps discussing 'ancestry from preceding farming communities'. I also think the authors could be more careful in talking about 'population turnover' and 'replacement'. I appreciate these are technical terms which reflect something specific related to ancestry change, but again going back to that point of precision, we see shifts and replacement in genetic ancestry not necessarily literally populations of people. Talking of local populations 'disappearing' also implies something rapid and immediate, when again I don't think the authors can be confident it was so total and rapid.'

'In Scandinavia, there is substantial evidence for an actual replacement of the local population as the first people with steppe ancestry in the area (the Battle Axe culture) carry no local ancestry of the preceding farming communities. Instead the people of the Battle Axe culture carry Neolithic ancestry from eastern Europe (mainly Poland)''

Apologies, I wasn't clear here - I wasn't questioning the size of the population turnover, only the authors' unequivocal statement that the 'Neolithic decline' instigated this population turnover. While plausible I don't think the authors can state this with such certainty at this stage and I think they could be more cautious with their language e.g. 'In Scandinavia this 'Neolithic decline' is followed by a massive population turnover...'

Again, these are only minor suggestions and I don't expect the manuscript would need to come to me again. I'd just like to congratulate the authors and thank them again for an excellent and constructive review process.

Reviewer #4 (Remarks to the Author):

I consider that the authors have done an excellent job, and that after implementing the suggested changes, both the analyses and the discussion of the data are now complete and presented with a very high level of quality.

I only ask the authors to carefully review the manuscript to ensure that all points where supplementary notes, figures, or tables are relevant are clearly indicated. For example, in the caption of Figure 4C, it would be important to explicitly explain what is meant by the A, B, and C family groups and to include a reference to the corresponding supplementary material. Similarly, in lines 457–459, the authors should clarify whether the Sr data derive from a published reference (and cite it

accordingly) or from supplementary material and make a call to the relevant section should be added.

I reiterate my congratulations to the authors and I do not suggest any further changes.

General remarks to reviewers

First we would like to thank the reviewers for their time, their comments have helped improve the paper significantly. We believe that the current version of the manuscript is considerably strengthened and represents a clear improvement on the initial submission. As suggested by all four reviewers (and the editor) we have changed the focus on the paper away from the plague, and have toned down the interpretation of the plague results significantly. Instead of focusing on what instigated the Neolithic Decline, we now focus on the effects that the Neolithic Decline had on the people. These changes have resulted in a new title, an entirely rewritten abstract and a largely rewritten discussion. Furthermore, we have added what we believe are very compelling new mixture modelling results (Figure 3) indicating a northwards spread of Neolithic Iberians that first moved into the Paris Basin after the Neolithic Decline. This result better contextualises the findings from Bury with other genetic datasets from Western Europe and clearly illustrates the population turnover that happened across the Neolithic Decline in the Paris Basin. Lastly, you will notice that we have changed the author order so that Frederik Seersholm is the first listed of the three first-authors.

We have addressed all of the comments below and hope that the revised version of the manuscript is suitable for publication in Nature Ecology and Evolution.

For easier reading, we have copied all reviewer comments into this letter (blue text color). The response to each comment is in black text color (non-italicised), while excerpts from the main text illustrating the changes are shown in italicised black text.

Response to reviewer comments

Reviewer #1 (Neolithic bioarchaeology)

Nature review

The paper titled “Plague and the European Neolithic Decline: Population Discontinuity and Shifts in Societal Structure” presents compelling paleogenomic results from the collective burial site of Bury (Paris Basin) in the Late Neolithic. The study identifies two distinct phases of body deposition, each exhibiting different genetic pedigree patterns. Additionally, four individuals were found to carry the *Yersinia pestis* pathogen. This finding is discussed in relation to population decline at the end of the Neolithic and the cessation of megalithic construction.

While the paper presents robust and novel data, it cannot be published in its current form and requires major revisions. My primary concern is the focus on the plague pathogen as the key argument for positioning Bury as the missing link in explaining Neolithic population decline. This claim is not sufficiently supported, given that only four individuals tested positive for the pathogen.

This overemphasis on plague diminishes the significance of the truly groundbreaking findings from Bury, which deserve greater attention. These include:

- Rare evidence of half-sibling relationships
- Clear indications of a shift in social organisation over 100–200 years

Rather than prioritising the plague narrative, more space should be dedicated to properly presenting the site and incorporating archaeoanthatological analysis to contextualise these important discoveries.

General comments

The dates used throughout the paper are inconsistent, alternating between BCE and BP. Additionally, some individuals are presented with a date range, while others are listed with a standard deviation (\pm). A uniform approach should be applied.

Corrected. We have changed all dates to BCE.

The authors use "pedigree" and "family" interchangeably, despite acknowledging that kinship and family structures in the Neolithic period remain unknown. It would be more appropriate to consistently use "pedigree", as it is a more neutral term.

We agree, and have changed the wording to pedigree or pedigree group throughout. However, we keep the terms mother, daughter etc for clarity of wording .

Terminology Clarifications

- PCA (Principal Component Analysis): This should be defined in the first paragraph of the section Genetic discontinuity between two phases.

Corrected

- "Founding act" and "founder": The reasoning behind these terms is not clearly explained. The authors should clarify whether this designation is based on C14 dating, stratigraphic position, or other criteria to justify their status as the source of the burial. If Pedigree 1 represents the earliest burial phase, this should be explicitly stated. A reference to Rivollat et al. (2023) could help strengthen the argument.

The individual 3 (BUR174, adult) is at the bottom of the sequence, in the corner of the rear end of the grave, whose access was difficult after the first deposits (Salanova et al. 2018). The 14C dates are moreover among the earliest ones. The best comparisons come from Germany, as for example the individual 44 found in the corner of the Schönstedt *totenhütte* (Feustel 1972). We have clarified this in the main text which now reads:

This arrangement was previously hypothesised to be part of a “founding act” of the monument because of the position and characteristics of BUR174 in the corner of the rear end of the grave² (Figure 1a). Given our genetic results and BUR174’s position in the first generation of pedigree group 1.A, this hypothesis seems likely.

- "Avuncular" (Figure 3): This term should be explicitly defined.

Corrected. We have added a sentence on this in the legend of Figure 3, which reads:

The stippled line represents an unknown avuncular relationship (relationship between uncle/aunt and nephew/niece)

- "Dark Age": This term should be avoided, as it is heavily associated with the Middle Ages. Since the paper consistently uses "Neolithic decline", this term should be maintained for consistency.

Corrected. We have avoided the use of the term 'Dark Age'.

Additional References Needed

More references should be included to support key discussions in the paper:

- Demographic profile: Additional references are needed to contextualise the demographic data and their interpretation, in the text and in supplementary information 1.

To keep the text short, and as the article is mainly focused on genetics, SI 1 only summarizes the results on the demographic data. However, since our initial submission, the full demographic analyses have been published separately: Chambon and Salanova 2025, *Prähistorische Zeitschrift*. To clarify this, we have cited this paper both in the main text and in Supplementary Information 1.

- Reference panel of ancient genomes: The discussion on IBD (identity by descent) and farming ancestry across Europe should be supported by recent studies.

In response to one of reviewer 2's comments, we have added a paragraph detailing a continent-wide IBD-sharing analysis to put our findings into broader context entitled 'Neolithic ancestry from the south could explain the shift at Bury'. In this paragraph we have also cited most of the relevant recent studies (see reply to reviewer 2 for details).

- Comparative studies on social structure: It is surprising that the authors do not reference recent publications addressing similar topics in either the same geographic area in earlier periods (e.g., Rivollat et al., 2023) or the same period and burial structure in a different region (Cassidy et al., 2020). Additionally, the social aspect of the population dynamic should be also backed up by social anthropology references and archaeoethnography studies (e.g. the work of Jeunesse and coll. or Gallay).

We agree. To correct this, we have added citations on relevant studies on social structure as follows:

-We cite Rivollat et al. 2023, Rivollat et al. 2022 and Immel et al. 2021 in the results section:

Genetic sexing confirmed the high predominance of males in both phases²² (Figure 2b; Supplementary Information 1), similar to reports from other Neolithic sites from present-day France and Germany²³⁻²⁶.

Considering the social aspect of the population dynamics we have cited Chambon and Salanova 2025 which lays out the theoretical framework, and details societal dynamics in comparable structures:

The demographic profile of the 180 individuals buried during the first phase does not correspond with the normal age-mortality pattern expected in such a population²¹ (Supplementary Information 1).

- Simulation of two scenarios (discontinuity vs. continuity): These should be explicitly detailed with appropriate references or a reference to the methods section.

Corrected. We have added a reference to the Methods in the relevant paragraph.

- Social and cultural ties in Phase 2: The argument that longer occupation suggests a greater role for social or cultural continuity should be supported with literature.

Good point. We have rephrased the sentence to focus on the higher number of unrelated individuals instead:

Phase 2, on the other hand, is characterised by a single paternal family line, and a handful of small pedigrees scattered throughout the different spaces of the tomb. As this phase is

also represented by a significantly higher proportion of unrelated individuals, this could imply that social or cultural ties were more important during this time.

- **Plague outbreaks and site abandonment:** The authors suggest that plague outbreaks may have led to site abandonment without proper burial of the last deceased. However, this claim requires references to comparable archaeological evidence of epidemic-driven abandonment.

Corrected. We have rephrased the text:

However, the grave only represents a subset of the population, and it is perfectly possible that a severe plague epidemic would leave very little evidence behind, if the entire population perished without being buried or if they were buried differently, presumably elsewhere, as were the case for many plague victims during the second pandemic^{38,39}.

- **The Scandinavian Neolithic decline:** This parallel is mentioned but lacks proper referencing and context.

Corrected. We have elaborated on differences and similarities to the Scandinavian Neolithic decline in a new paragraph of the discussion:

As opposed to Scandinavia where steppe-related groups replaced the local farmers 4,700 years ago⁴⁶, the population dynamics in the Paris Basin is characterised by the replacement with another group of Neolithic people, who persisted for around 500 years until the arrival of steppe-related populations.

- **Strontium analysis and broader mobility context (Supplementary Information 5):** More references should be included to strengthen the interpretation of these findings.

To address this we have added a couple of sentences on the Strontium analyses in the discussion:

Furthermore, our strontium isotope data depicts a more stable and sedentary population in Bury than earlier Neolithic groups from the area (Supplementary Information 5). Tied in with our findings on the replacement of the population between Phase 1 and Phase 2, this suggests that newcomers did not immediately reuse the tomb. Still, we do observe some mobility with a total of 14 Strontium outliers identified (6 from Phase 1, and 8 from Phase 2) most of which are either unrelated individuals or exogamous females.

Furthermore, we added the following references in to Supplementary Information 5 to strengthen our arguments and better contextualise our findings:

-Price, T. D., C. Johnson, J. Ezzo, J. Burton, and J. Ericson. 1994. Residential mobility in the American Southwest: A preliminary study using strontium isotope analysis. *Journal of Archaeological Science* 21: 315–330.

-Price, T. Douglas (ed.) 2023. *Isotopic Proveniencing and Mobility. The Current State of Research*. Cham: Springer.

-Bentley, R.A. 2006. Strontium isotopes from the Earth to the archaeological skeleton: A review. *Journal of Archaeological Method and Theory* 13: 135–187.

-Montgomery, J. 2010. Passports from the past: Investigating human dispersals using strontium isotope analysis of tooth enamel. *Annals of Human Biology* 37: 325–346.

Summary

Some editing is required. One sentence remains incomplete: “Recent research that is ‘Neolithic decline’ ...” This should be rephrased for clarity.

Thank you for noticing this. As part of a revision of the summary, this sentence was removed.

Additionally, *allée sépulcrale* should be written in italics.

Corrected

Introduction

The authors should provide a more detailed definition of what constitutes a collective burial site, along with an overview of the main theories regarding this type of site. This would strengthen the subsequent arguments about population decline and provide a clearer framework for discussion. What are the prevailing interpretations of collective burial sites in archaeological research?

Good point. We have expanded on the section about megalithic graves in the introduction to address this:

*These tombs are numerous throughout northwestern Europe, with very high concentrations in places such as the Paris Basin, Central Germany and southern Scandinavia. They are collective burials that accommodated deceased as they died, and thus collected tens of thousands of dead in the second half of the fourth millennium BCE⁷⁻⁹. However, the burial practices, in detail (position, orientation, intervention on skeletons after decay, grave goods...), all fluctuated according to chronology and cultural contexts. Constructions vary from so-called passage graves made from large boulders mainly found in northwestern Europe and southern Scandinavia, to long cist graves (gallery graves) made of stone slabs more common in the Paris Basin (*allées sépulcrales*) and central Germany (*Galeriegräber*) which were used by different archaeological groups: Seine-Oise for the *allées sépulcrales*¹⁰, Wartberg for the *Galeriegräber*¹¹, Bernburg for the *Totenhütten*¹² and the Funnelbeaker/Trichterbecher cultural complex for the *Scandinavian passage graves*¹³.*

Regarding the evidence for population decline, the authors should reference relevant literature, including archaeological and demographic sources, to support their claims. What types of evidence are typically used to identify such declines, and how do they apply to this study?

In the interest of keeping the length of the introduction down, these references are indicated in Chambon Salanova 2025 and summarized in Supplementary Information 1

Demographers have long proposed models for mortality. For pre-industrial populations, extrapolations were carried out from historical sources, leading to model life tables. French-speaking anthropologists regularly refer to those published by Ledermann (1969), synthesized by P. Sellier (1996). Seguy and Buchet have more recently updated and completed these tables (2010). Concerning the compliance of the mortality table with an attritional mortality table and the growth rate, we also commonly use the estimators proposed by Bocquet-Appel and Masset (1977)

Bocquet, J.-P., and C. Masset. 1977. Estimateurs en paléodémographie. *L'Homme* 17(4):65–90.

Ledermann, S. 1969. Nouvelles tables-types de mortalité. *Travaux et Documents* 53. Paris: Ined.

Séguy, I., and L. Buchet. 2011. *Manuel de paléodémographie*. Paris: Ined 2011.

Sellier, P. 1996. La mise en évidence d'anomalies démographiques et leur interprétation: population, recrutement et pratiques funéraires du tumulus de Courtesoult. In: J.-F. Piningre (ed.), *Nécropoles et société au premier âge du Fer: le tumulus de Courtesoult*. Paris: Maison des Sciences de l'Homme, pp 188–202.

The description of the semi-megalithic *allée sépulcrale* of Bury requires additional context and a more detailed stratigraphic account to demonstrate the presence of two distinct occupations. This could be included as supplementary information, such as field documentation like section drawings. It would allow for a clearer evaluation and validation of the authors' interpretations. Additionally, the concept of a semi-megalithic structure should be explicitly defined for clarity.

We agree. Semi-megalithic is a French denomination, which refers to building techniques, although the inner plans of graves are homogenous for both *allées sépulcrales* and gallery graves. In other parts of Europe, this kind of distinction is not made. To avoid confusion, we have avoided the use of this term. Moreover, to clarify what *allées sépulcrales* refers to, we have elaborated on this type of megalithic tomb in the introduction which now reads:

As a semi-underground monument of rectangular shape, Bury is a classic example of the allées sépulcrales found north-west of Paris, built with a combination of megalithic slabs and other techniques, like dry-stone walls.

Lastly, to better outline the difference in the use of the grave in the two burial phases, we have included schematics with burial positions for the different reconstructed pedigrees in Supplementary Information 7 and have updated the grave overview in Figure 1.

The demographic profile for the first phase is discussed, but there is no mention of the second phase. A similar analysis should be provided to ensure a comprehensive understanding of the demographic changes over time.

We have clarified this in Supplementary Information 1, that now constitutes a summary of the demographic results which have been published on its own in Chambon and Salanova 2025. Moreover, in the main text we have highlighted the fact that the Phase 2 population follows the mortality profile of a normal population:

The demographic profile of the 180 individuals buried during the first phase does not correspond with the normal age-mortality pattern expected in such a population²¹ (Supplementary Information 1). Rather, the demographic profile is suggestive of excess mortality, particularly affecting juvenile individuals, perhaps indicating a catastrophic event, such as war, famine or a disease outbreak or, on the contrary, a rapid increase in the population. Phase 2, on the other hand, shows no indication of elevated mortality.

Results

The authors present a sample of 182 teeth from 179 individuals. How was the selection made to ensure that the same individual was not tested twice? In commingled contexts, we usually sample the bone that give the best MNI (Minimum Number of Individuals), but this does not seem to be the case here. The sampling strategy should be detailed.

Corrected. We have added a section on the sampling strategy under Methods:

Sampling strategy

Sampling was carried out with approval from the relevant scientific authority at that time. In total, we collected 182 ancient human teeth samples, representing 179 individuals (57% of the total of 316 individuals estimated to have been buried at the site). We decided to sample mandibular teeth, since the skulls could not be linked to the identified archaeological individuals (Supplementary table 1). In total, we sampled more than 71% of the testable mandibles, including all but one of the mandibles from identified archaeological individuals. Lastly, we also sampled 21 loose teeth to ensure the broadest representation possible. Along with all other samples, these loose teeth were subsequently screened genetically to identify any duplicates (see below).

Lastly, it should also be added that the teeth offer the possibility to search for DNA of pathogens, which would not have been possible if we had taken femurs (this bone gives the best MNI in Bury).

In the Excel sheet presenting the sample, some individual numbers are listed alongside loose teeth. However, this aspect has not been explained in the context of the burial, highlighting the need for a clearer presentation of the site and a clear definition of the sampling strategy.

Corrected. See above.

The statement, “genetic sexing [confirmed] the high predominance of males in both phases,” is made without a full biological profile for the entire site or for each phase. A comprehensive biological profile is crucial for a proper presentation of the site.

Thank you for your suggestion. To address this we have added a short paragraph on the morphological sex determinations across the two phases to Supplementary Information 1, and cited this in the main text:

Morphological sex determination

On the basis of the coxal bone, the sex estimate counted 24 females and 52 males for the entire grave. Due to the lack of well-preserved coxal bones, this estimate is only one third of the minimum number of individuals over 15 (228). Indicative if not significant, the imbalance affects both phases, 12/29 in phase 1 and 22/24 in phase 2.

Figure 1: The publication date is missing in the caption.

Corrected.

Genetic discontinuity between two phases

The genetic ancestry determination should be contextualised more thoroughly. Given that this study focuses on the end of the Neolithic (even during its decline), why should we still be looking for Anatolian farmer origins and Mesolithic populations? Providing more context on recent research in this area would help guide the reader through the authors' line of reasoning and strengthen their argument.

We agree that the previous iteration of the manuscript did not include a sufficiently detailed comparison with other genomes from the literature. To address this, we have added a paragraph on how the genetic ancestry at the two phases in Bury reflects a large scale transition across Iberia and present day France:

Neolithic ancestry from the south could explain the shift at Bury

In order to put our findings from Bury into context with other Neolithic people from Western Europe, we modelled the proportion of various ancestries in all individuals from Bronze age and Neolithic Europe (Methods). In agreement with our PCA analysis, we found high diversity in the Phase 1 individuals with varying proportions of modelled ancestries from Early Neolithic France and from a group of Neolithic Iberians dated to the fourth

millennium BCE (Supplementary Note 6.1, Supplementary Figure 6.14 and 6.15). This pattern is reflected in other contemporaneous individuals from the Paris Basin from the sites Mont-Aime hypogee (I+II)²⁷, Wettolsheim²⁸ and Pont-sur-Seine²⁸. For Phase 2, on the other hand, the modelling revealed a very homogenous population modelled with over 80% (mean: 83.8%, $\pm 0.1\%$ SD) ancestry from Iberia.

When visualising major ancestry groups on a map (Figure 3), the mixture modelling reveals a stepwise northwards spread of this Neolithic Iberian ancestry (Supplementary Note 6.1). By 2,900 BC, populations across southern France and Iberia all constituted a large fraction of Iberian ancestry, while people in the Paris Basin still comprised mixed ancestry proportion as represented by the Phase 1 individuals. At some point after 2,900 BC, a final northwards push of the Iberian ancestry partially replaces the existing population in the Paris basin resulting in the homogenous population we observe in Phase 2. After the end of Phase 2, around 2,500 BC, steppe ancestry first appears in the Paris Basin²⁹ (Figure 3). After a couple of hundred years this steppe ancestry will gain the Atlantic coast and mix with local populations carrying the Iberian ancestry. As such, these results readily explain the difference between the populations of Phase 1 and 2, and could suggest an event that facilitated the northwards expansion of Neolithic Iberian ancestry at around 2,900 BC.

A shift in societal structure inferred by genetic links

“we found that three-quarters of individuals...” Including precise percentages or actual numbers would enhance clarity.

Corrected

A more detailed explanation of the sampling strategy would clarify why some individuals were not sampled or did not yield results. What were the selection criteria? For example, in Pedigree 1.A, the authors mention 29 sequenced individuals, while 21 were not sampled or were buried outside the grave.

See above re. the sampling strategy. For the ‘unsampled’ individuals in the pedigrees, our wording was unclear. The individuals of grey color represent inferred individuals that must have existed. I.e. to draw a maternal grand-parent/grand-child relationship, the mother has to be included, even though she was not identified in the grave. We have clarified this in the text (and updated counts), which now reads:

Pedigree 1.A is the largest of the Phase 1 pedigrees (Figure 4, Supplementary Figure 6.2a), containing 29 sequenced individuals and a further 19 inferred individuals who were either not sampled or buried outside the grave.

This highlights two key concerns:

1. The lack of contextual information regarding burial practices and funerary customs of the time.

Good point. To address this, we have expanded the introduction, and cited relevant work where a detailed record of funerary practices for the cultures discussed can be found:

These tombs are numerous throughout northwestern Europe, with very high concentrations in places such as the Paris Basin, Central Germany and southern Scandinavia. They are collective burials that accommodated deceased as they died, and thus collected tens of thousands dead in the second half of the fourth millennium BC⁷⁻⁹. However, the burial practices, in detail (position, orientation, intervention on skeletons after decay, grave goods...), all fluctuated according to chronology and cultural contexts. Constructions vary from so-called passage graves made from large boulders mainly found in northwestern Europe and southern Scandinavia, to long cist graves (gallery graves) made of stone slabs more common in the Paris Basin (allées sépulcrales) and central Germany (Galeriegräber) which were used by different archaeological groups: Seine-Oise for the allées sépulcrales¹⁰, Wartberg for the Galeriegräber¹¹, Bernburg for the Totenhütten¹² and the Funnelbeaker/Trichterbecher cultural complex for the Scandinavian passage graves¹³.

2. The absence of a clear explanation of the sampling strategy.

See above.

Regarding Pedigree 1.A, a summary would be helpful—such as the number of generations represented and a general biological profile of the individuals within this pedigree. This should be done for each pedigree (in supplementary information).

Corrected. We have added this information as *Supplementary Note 6.2 - Pedigree descriptions in Supplementary Information 6*.

For ID174, the reference to body position is noted, but the paper does not provide an overview of funerary practices, the bone assemblage at the site, or how these vary across different phases. This broader context is necessary for proper interpretation.

Thank you for the suggestion. To address this, we have added an overview of body positions as *Supplementary Information 7*.

The results for ID185 are particularly intriguing and should definitely be developed further. However, the only hypothesis proposed by the authors is that the genetic link belongs to an

unsampled individual. Could alternative explanations, such as adoption or fostering, be considered? While rarely addressed in archaeological samples, insights from social anthropology could support this possibility. If proven accurate, this would actually be a major discovery.

This is an interesting point. We know that ID185's grandparents are both buried in the grave, and they are part of pedigree group 1.B, but that both parents are missing. We added a short sentence on the possibility of adoption or fostering, which is a really intriguing point that sadly we cannot prove (or disprove):

One of the children, the only offspring of the unsampled daughter, BUR185 is a juvenile male who shares the same chromosome Y haplogroup as pedigree group 1.A - H2a1. This is remarkable, as the rest of pedigree group 1.B males have the I2a1 haplogroup. It could imply that BUR185's father has some genetic relationship to pedigree group 1.A, however this link has not been detected genetically - it could be that the relatives linking this individual with pedigree group 1.A either were not sampled, or were not buried in the grave. However, another possibility is that BUR185 has no relation to pedigree group 1.A, and that he was buried in the grave either due to having grandparents in group 1.B, or a societal relationship such as adoption or fostering, which is untraceable by DNA alone.

For ID307 and ID276, the identification of half-siblings is noteworthy—not only within this burial site but also in the archaeological record more broadly. This rare finding warrants a more detailed discussion and comparison with other known case studies, as it represents a significant outcome.

Good point. We have been through the literature in search for other examples of half-siblings. We found that, while it is not the norm, half-siblings are relatively often documented. We have expanded on this in the text:

Furthermore, the first generation of pedigree group 2.A is represented by two half-brothers (BUR307 and BUR276) who share the same mother, and whose fathers were related to each other. This is the only example of half-siblings in the dataset, which implies that having two reproductive partners was not the norm, or that the progeny from these unions were buried elsewhere, and thus also perhaps not socially sanctified. This finding is relatively rare among comparable genetic studies, but not unheard of. While no half brothers were found in the Neolithic site Gurgy from present day France, half brothers have been identified in Neolithic sites from the British Isles³⁴ and Scandinavia²⁰, and from later sites^{35,36}.

Finally, the study does not present body positions for each identified individual, making it difficult to evaluate the claim that there is no correlation between genetic relationships and burial placement. Providing this data would strengthen the argument.

Good point. We have added this data as Supplementary Information 7, and cited it in the main text:

There is no correlation between burial position within the grave and genetic relationship, except in the founding deposit (Supplementary Information 7).

Evidence for the earliest diverging lineage of *Y. pestis* at Bury

The paper does not provide any data on the general health status of the population(s). Including such information would offer important context and potentially strengthen the authors' claims.

While no published record on the general health of the populations in Bury exist, an unpublished analysis of the oral-dental area did not find evidence of poor conditions in the population (Marçais unpublished). We have highlighted this information in the paragraph on the pathogen screening:

While no published record on the general health of the populations in Bury exist, an analysis of the oral-dental area did not find evidence of poor conditions (Marçais unpublished). To investigate if this pattern could be corroborated by genetic data, we screened all samples for pathogen DNA (see Methods). In total, we [...]

Additionally, I was unable to locate Figure 7.7.

Corrected. Please see our response to reviewer 2, re. mislabeled supplementary figures.

Regarding the number of cases reported across the two phases, several points require clarification:

- The reported numbers (85 + 56 = 141) do not match the total population (n = 316), the sampled population (n = 179), or the number of teeth sampled (n = 182). Further explanation is needed to clarify these discrepancies.

Good point. The numbers refer only to the part of the samples with genetic data (coverage $\geq 0.1x$) that could be assigned to a phase. We have clarified this in the text (and updated the numbers to reflect changes made during the review process):

Given that we find three plague cases in Phase 1 (3/74, 4% of Phase 1 samples with coverage $\geq 0.01x$), and only a single case in Phase 2 (1/51, 2% of Phase 2 samples with coverage $\geq 0.01x$) [...]

- The authors effectively argue against their own hypothesis—that a plague outbreak led to the end of Phase 1—by presenting strong counterarguments. However, the alternative explanations they propose—stratigraphic position and a significant population increase—are not well documented and require further evidence.

We agree. We have rephrased this paragraph to focus on a single alternative explanation: That a plague outbreak could leave little evidence behind if infected individuals were buried elsewhere. The rephrased paragraph now reads:

However, the grave only represents a subset of the population, and it is perfectly possible that a severe plague epidemic would leave very little evidence behind, if the entire population perished without being buried or if they were buried differently, presumably elsewhere, as were the case for many plague victims during the second pandemic^{38,39}.

Environmental data on the Neolithic decline

This paragraph is poorly documented and not well connected to the paper's results, as it does not include any direct data from Bury. Additionally, this section, along with Supplementary Information 3 and 4, does not appear to be relevant to the paper.

To address this, we have written a new Supplementary section summarising pollen literature from the Paris Basin (Supplementary Information 2). We have rewritten the entire paragraph to reflect this:

Environmental Data on the Neolithic Decline

Lastly, we investigated pollen data from the Paris Basin to assess if the population collapse observed between Phase 1 and Phase 2 could be linked to any vegetational changes. We found evidence of forest regeneration in the Paris Basin from 2,900 to around 2,600 BCE, which is typically linked to a decrease in human activity (Supplementary Information 2). A similar pattern was observed, both in Scania, Sweden (Supplementary Information 2), where forest regrowth reached a climax around 3100 BCE, and in Zealand, Denmark, showing a climax between 3,000-2,800 BCE (Supplementary Information 3). Similar results have previously been documented in northern Germany,⁴⁰ and in central Europe where the decline period could be dated to between 3,300-2,950 BCE⁴¹ based on a combination of Summed Probability Distributions (SPDs) of radiocarbon dates and palaeoecological proxies. As such, these observations from a number of well documented regions can be interpreted as resulting from abandonment of grazing lands and fields, implying settlements were given up. Accordingly, they describe a significant decline in human activity, and are in agreement with similar observations after the Justinian plague⁴² and the Black Death⁴³.

Discussion

I disagree with the statement that Bury is the missing link in demonstrating the "Neolithic decline", as no convincing evidence has been presented to establish a plague outbreak, its environmental impact, or other contributing factors. While I agree that these elements could be explored further using the new data from Bury, the current evidence is not compelling enough to support this claim.

Corrected. We have rephrased the sentence to:

The data from the Bury grave provides important new evidence regarding both the 'Neolithic Decline' and the general population dynamics in the fourth and third millennium BCE.

The parallel drawn with Scandinavian data is weak, lacking sufficient references and clear examples that directly relate to what happened at Bury.

Corrected. See above

Overall, the discussion comes across as a tentative and unsubstantiated attempt to link Bury to broader patterns of Neolithic decline in Europe, rather than a well-supported argument grounded in robust comparative evidence.

We agree. We have rewritten large parts of the discussion to address this, adding our interpretation of the new mixture modelling data and referencing the Strontium data. We believe that this revised version offers a more well-founded and cautious discussion.

Methods

Paleogenetics is not my area of expertise, so I have no specific comments on this section. However, the sample selection process should be clearly outlined here.

Corrected. See above.

Supplementary Information 1

While I agree that the integration of juveniles into archaeological studies is not yet standard in biological anthropology, some relevant work has been published in the last 10 years that would be worth mentioning.

Good point. Since large parts of Supplementary Information 1 was published on its own after our initial submission (Chambon and Salanova 2025), this section has now been compressed to a summary. In this new shortened text, we have removed the phrase stating that the inclusion of infant mortality is uncommon biological anthropology.

Supplementary Information 4

Although this section presents an interesting topic, its connection to Bury is not clearly explained, making it seem irrelevant to the paper. Additionally, the references are not cited in the main text.

Thanks for pointing this out. We agree that we did not clearly justify the SI on TRB megalithic sites in the original manuscript. In the rewritten version of the manuscript, we have made sure to cite the section and to outline its relevance for our study. The rewritten paragraph reads:

The same hiatus in burial sequence is known in similar graves from Germany^{48,49}. There is also evidence from megaliths in Northern Germany that ended around 3,100-3,000 BCE, followed by decreasing open land^{13,21,50,51}. Additionally, we present evidence of precise

construction dates for Danish passage graves ending around 3,000 BCE (Supplementary Information 4). When we add the anomalies in the demographic structure of the buried population in Phase 1 of Bury as well as other megaliths from continental Europe, there are further indications of demographic change (Supplementary Information 1).

Supplementary Information 5

This section is confusing, as it does not clarify why Le Tumulus des Sables is relevant for comparison with the Bury site. Further explanation is needed to establish its significance.

We agree that it was unclear why data from Le Tumulus des Sables was included. We have clarified this in the beginning of the paragraph about that site:

To better contextualise the findings at Bury, we decided to include data from the site Le Tumulus des Sables here. The burial mound of Le Tumulus des Sables, near Bordeaux in southwest France, dates from the Neolithic to the Iron Age, and is one of the few Neolithic sites from present-day France where extensive Strontium work has been conducted. Isotopic analyses of human teeth [...]

Reviewer #2 (Palaeopathogen genomics and Yersinia pestis)

The paper Plague and the European Neolithic Decline presents a highly speculative and poorly substantiated claim: that a genetic discontinuity at the Bury burial site reflects a widespread population collapse, potentially linked to Yersinia pestis. However, this interpretation suffers from major methodological flaws, weak genetic evidence, incomplete pathogen identification, and an overall lack of transparency in analytical procedures. The conclusions drawn are not supported by the data or methods described.

Major Issues

1. Population Genetics: Incomplete Comparisons and Lack of Continental Context

The paper argues that a break in identity-by-descent (IBD) sharing between burial phases at Bury reflects a broader demographic collapse. This argument is seriously undermined by several critical omissions:

No Pan-European IBD Framework:

The authors fail to provide a continent-wide IBD-sharing analysis to place the Bury discontinuity in context. Without comparison to other contemporaneous Neolithic populations, the claim of a continent-wide collapse remains speculative and unsupported.

We agree that the last version of the paper lacked a proper IBD-sharing analysis to put our findings into context. To address this we have added a paragraph on the matter in the main text, along with one main figure (Figure 3) and two supplementary figures (Supplementary Figures 6.14 and 15). The newly added paragraph reads:

Neolithic ancestry from the south could explain the shift at Bury

In order to put our findings from Bury into context with other Neolithic people from Western Europe, we modelled the proportion of various ancestries in all individuals from Bronze age and Neolithic Europe (Methods). In agreement with our PCA analysis, we found high diversity in the Phase 1 individuals with varying proportions of modelled ancestries from Early Neolithic France and from a group of Neolithic Iberians dated to the fourth millennium BCE (Supplementary Note 6.1, Supplementary Figure 6.14 and 6.15). This pattern is reflected in other contemporaneous individuals from the Paris Basin from the sites Mont-Aime hypogee (I+II)²⁷, Wettolsheim²⁸ and Pont-sur-Seine²⁸. For Phase 2, on the other hand, the modelling revealed a very homogenous population modelled with over 80% (mean: 83.8%, $\pm 0.1\%$ SD) ancestry from Iberia.

When visualising major ancestry groups on a map (Figure 3), the mixture modelling reveals a stepwise northwards spread of this Neolithic Iberian ancestry (Supplementary

Note 6.1). By 2,900 BC, populations across southern France and Iberia all constituted a large fraction of Iberian ancestry, while people in the Paris Basin still comprised mixed ancestry proportion as represented by the Phase 1 individuals. At some point after 2,900 BC, a final northwards push of the Iberian ancestry partially replaces the existing population in the Paris basin resulting in the homogenous population we observe in Phase 2. After the end of Phase 2, around 2,500 BC, steppe ancestry first appears in the Paris Basin²⁹ (Figure 3). After a couple of hundred years this steppe ancestry will gain the Atlantic coast and mix with local populations carrying the Iberian ancestry. As such, these results readily explain the difference between the populations of Phase 1 and 2, and could suggest an event that facilitated the northwards expansion of Neolithic Iberian ancestry at around 2,900 BC.

Unclear IBD Comparisons with Other Sites:

The methodology for comparing IBD-sharing between Bury and other Neolithic sites is not transparently described. It remains unclear whether the genetic differences at Bury are unique or fall within expected regional variation. This lack of comparative clarity casts doubt on the robustness of the conclusions.

We agree. As outlined above, we have added a new paragraph detailing how the patterns at Bury are reflected in other genomes from the region. In this paragraph we have added a short statement on some of the sites that reflects the patterns observed at Bury:

This pattern is reflected in other contemporaneous individuals from the Paris Basin from the sites Mont-Aime hypogee (I+II)²⁷, Wettolsheim²⁸ and Pont-sur-Seine²⁸.

Missing or Misreferenced Figures:

Several figures mentioned in the main and supplementary texts are either missing or incorrectly referenced. For example, Supplementary Figure 7.4 (referenced for hierarchical clustering of IBD) is not present; Suppl. Information 6 only includes figures 6.1–6.7. Such inconsistencies undermine confidence in the visual evidence supporting key claims.

This was a mistake on our part. In the last minute before submission, we removed a supplementary section, which inadvertently caused all figures from Supplementary Information 6 to be mislabelled as Supplementary Figure 7.X (instead of 6.X). To fix this, we have gone through the main text and all associated manuscript files to ensure that cross references have been updated.

Lack of Effective Population Size (Ne) Estimates:

The authors do not include demographic modeling such as Ne estimates, genome-wide heterozygosity analyses, or coalescent simulations. These are essential to support claims of

population decline. Instead, they rely on burial pattern interpretations, which could reflect social, ritual, or taphonomic changes rather than demographic collapse.

Thank you for the suggestion. To address this we have included IBDne estimates of population sizes for the two phases. As depicted in the figure, these results suggests that the populations from each phase follow separate population size trajectories, strengthening our hypothesis of two distinct populations at Bury:

Supplementary Figure 6.17. Population size estimated. IBDne population size estimates for phase 1 and 2, respectively plotted against absolute age (ca. BCE).

Cultural Transitions Misinterpreted as Demographic Events:

The genetic shifts around 4500 BP coincide with significant cultural transformations and changing burial customs. The persistence of distinct genetic profiles at Bury (e.g., minimal Steppe ancestry) argues against a collapse scenario and instead points to cultural continuity or interaction.

We agree that it was unclear whether the shift at Bury represented only cultural or also genetic transformations in the old version of the manuscript. With the updated data correlating the findings at Bury with a large expansion of Iberian Neolithic ancestry from 3,800 to 2,300 BC, we believe that we have shown that the two populations studied were in fact two different ancestry groups.

2. Kinship Analysis: Overreach from Limited and Ambiguous Data

The kinship analysis identifies three extended families within the burial site, yet the authors overgeneralize these findings to the entire population.

Inadequate Justification of Methodological Choices:

The authors used five different programs to estimate kinship—potentially increasing robustness—but failed to explain how conflicting outputs were reconciled. These tools often

yield differing results, so clarity about which results were prioritized, and why, is essential for transparency and reproducibility.

We agree. To clarify exactly how each tool performed, and how the prediction from each tool fit with our final pedigrees, we have added a new table (Supplementary table 7). In this table we have included the output of each tool as well as our final ‘call’ based on all available data. We have also included two figures to summarise the performance of the tools used (Supplementary figures 6.10 and 6.11) and a short text discussing our approach in the methods section:

We found a very high level of agreement between all approaches for high coverage samples, when comparing the performance from each of the five kinship tools (Supplementary Figures 6.10 and 6.11). However, for lower coverage samples, we observe greater variability in the results. We generally found that KIN, ngsRelate2 and READ2 produced more consistent results than the two other approaches, and accordingly we have put highest confidence in predictions from these tools. In cases where these three tools did not agree, we generally followed the prediction from KIN unless contradicted by other data, such as mitochondrial haplogroups, sex or relatedness to other individuals (Supplementary Table 7).

Arbitrary Population-Level Extrapolations:

Most individuals are not closely related, contradicting the notion of a kinship-based burial pattern. Nevertheless, the authors generalize from a few family groups to the whole population without considering alternative explanations (e.g., social organization, mortuary variability).

Good point. For Phase 1, there is good evidence of a kinship-based burial pattern as 81% of buried individuals are part of a family. For Phase 2, on the other hand, this ratio is 60%. We have clarified that the burial practices in Phase 2 might follow other kinds of social organisation:

However, the higher fraction of totally unrelated individuals could speak to the monument following a different kind of organisation not restricted to a few extended families.

Speculative Interpretations of Sex Ratios and Kinship Structures:

The underrepresentation of females is presented as evidence of patrilineal burial, but other plausible factors—such as preservation bias, differential burial treatment, or sampling artifacts—are not addressed.

While preservation bias and sampling artifacts are less likely, it is certainly possible that the observed patterns could be explained by differential burial treatment. We have clarified this in the beginning of the results section:

This imbalance concerns both adult and non-adult individuals, and is incompatible with a natural population, thus suggesting differential burial treatment between males and females at Bury: for some reason more than half of the females in the community were excluded from being buried in the grave.

3. Yersinia pestis Detection: Insufficient Evidence for Epidemic or Mortality Impact

The study's most dramatic claim—that *Y. pestis* contributed to population collapse—is based on questionable evidence.

Inadequate Pathogen Identification:

The study does not provide rigorous validation of *Y. pestis* reads. No mapping statistics, depth coverage data, or comparison with related environmental bacteria (e.g., *Y. pseudotuberculosis*) are presented. Contamination or misidentification cannot be ruled out.

Thank you for your comment. To address this, we have added a new table with mapping statistics for the four plague cases (Supplementary Table 8). Furthermore, we have clarified that comparisons between all species within the genus *Yersinia* were carried out as part of the pathogen screening pipeline in the Methods sections:

This approach ensures that all closely related species within each genus are compared with competitive mapping after the initial krakenUniq step. A microbial hit is only considered a true positive detection when a given microbial species is detected with higher similarity than all of its close relatives.

Lastly, as another sanity check, we have included DNA damage plots for the plague positive samples in the study:

Supplementary Figure 6.8. Ancient DNA damage for *Yersinia pestis* reads. C>T and G>A misincorporations in the four plague positive individuals detected in this study. Only double stranded libraries that were not USER treated are shown.

Lack of Evidence for Virulence:

There is no reported coverage of key virulence factors (pla, caf, ymt), making it impossible to assess whether the strains could cause disease.

Good point. As suggested we have included a plot with coverage for key virulence genes for the plague case with highest coverage (BUR218). For the lower coverage plague detections, the lack of data produces too much noise in the plot to provide any meaningful insights:

Supplementary Figure 6.9. Virulence gene coverage in BUR318. Color indicates per gene coverage normalised to the average plague coverage in the sample.

No Osteological or Burial Context Support:

There is no skeletal evidence of rapid death, nor signs of epidemic burial (e.g., mass graves, irregular positioning). Furthermore, the low prevalence of *Y. pestis*—4% in Phase 1 and 2% in Phase 2—is far too low to suggest a significant outbreak.

We agree. Since one of the main revisions of the paper involved toning down the plague narrative, this new version does not argue for a significant plague outbreak any longer. Yet, we note that it is possible to observe similar patterns in epidemic events, if only the first few disease cases were buried in the tomb, while mass graves or other ways of dealing with dead bodies were created elsewhere when the outbreak accelerated.

Absence of Epidemiological Modeling:

The study lacks any modeling to assess the demographic impact of *Y. pestis*, leaving its role speculative and unconvincing.

We agree. However, epidemiological modelling is very challenging given our current dataset as the population that were supposedly hit by this outbreak didn't survive. We hope that the revised and refocused narrative addresses the reviewer's concerns.

4. Missing Methodological Details and Reproducibility Failures

The paper fails to provide critical methodological information necessary for independent verification and replication:

Mapping and Genotype Calling:

The reference genome used for mapping (e.g., hg19) is not stated. It is also unclear which positions were used for pseudohaploid calling—was the 1240K panel used?

Thank you for your comment. We have clarified these points in the methods section which now reads:

[...] by randomly selecting alleles at all positions in the reference genome (hg38) and merging this dataset with a panel of 4,748 imputed ancient human genomes (Supplementary Table 9). We characterised genetic diversity using the principal component analysis implemented in PLINK⁶³ and the smartpca⁶⁴ approach from the eigensoft tool⁶⁵, focusing on a subset of 2,011,119 transversion-only SNPs with a minor allele frequency over 0.1%.

Dataset Merging:

The authors state that they merged their data with ~6,000 samples, but do not specify which samples, from which release, or on what criteria (e.g., geography, time period). The current AADR release includes many more samples.

Good point. We use an in-house database that is regularly updated following our needs. During this review process we have updated the database and have removed genomes <0.1X coverage, so it now contains 4,748 genomes. We included the full list of all samples in our panel as Supplementary Table 9.

Principal Component Analysis (PCA):

The strategy for PCA is ambiguous. Was it performed using modern West Eurasian populations with projection of ancient individuals, or using a combined dataset?

Only ancient data was used for the plot in Figure 2c. We have clarified this in the methods section which now reads:

For the smartpca plot in Figure 2c a sub-panel of 3,779 ancient west eurasians was used (panel: 'Eur' in Supplementary Table 9), with samples from this study projected onto the reference panel diversity.

IBD Analysis Parameters:

The IBD analysis was conducted using IBDseq, but no parameter details are provided. More suitable tools for ancient, low-coverage DNA (e.g., ancIBD) are not considered, nor is a rationale given for choosing IBDseq.

Thank you for your comment. In fact, our group has recently carried out an extensive comparison between IBDseq and ancIBD as part of another paper: McColl et al. 2025 bioRxiv, Supplementary Information section S5.3.5 *Comparisons with ancIBD* (link). From this analysis we found that IBDseq better fit our needs because of its efficient resolution of short IBD segments which are important for our downstream analysis. We have clarified this in our methods section which now reads:

*To investigate fine scale genetic structures within each population, we merged these imputed genomes with our imputed reference panel (Supplementary Table 9), and filtered genotypes retaining only sites with a MAF over 1% and imputation quality (INFO score) over 0.5. We used IBDseq (r1206)⁶⁸ to characterise regions shared identity-by-descent (IBD) between all pairs of individuals with default settings. We chose IBDseq for calling IBD segments because of its advantage in detecting shorter segments, which are integral for our downstream clustering and mixture modelling analysis (See McColl et al. 2024 bioRxiv⁴², Supplementary Information section S5.3.5 *Comparisons with ancIBD*).*

Haplogroup Calling:

mtDNA and Y-chromosome haplogroups are reported without any details on the calling procedure, QC measures, or tools used. If performed manually, the criteria and process should be described.

For mtDNA we used haplogrep, while we used an in-house script based on the ISOGG snps for Y chromosome haplogroups. We have clarified this in the methods section under ‘Basic bioinformatics’:

Mitochondrial haplogroups were called using Haplogrep 2 (v.2.1.25) on variants called by mutserve (v.1.3.0). Chromosome Y haplogroups on the other hand, were called using an in-house script detailed in Supplementary Note 2 of Seersholm et al. 2024²⁰.

Ancestry Estimation:

The study reports ancestry proportions (e.g., 90% farmer, 10% hunter-gatherer) but does not state which method was used (e.g., supervised ADMIXTURE, qpAdm). Modeling details, parameter choices, and full results (e.g., in Supplementary Tables) are missing.

We agree, this information should have been included in the original submission. For the ancestry proportions we used the mixture modelling approach described in Allentoft et al 2024 and available on github here: mixmodel_ibd. We have added this information as a new paragraph in the methods section, with information on the mixture modelling source groups in Supplementary Note 6.1:

Main text:

Mixture modelling

Raw IBD coordinates from IBDseq were first converted into centimorgan using the genetic map for GRCh38: genetic map hg38 withX.txt.gz . Next, we removed IBD tracts with excess sharing across all individuals, and filtered out segments shorter than 2cM, as described in Allentoft et al. 2024⁶⁹. We modelled all individuals from this study as mixtures of 8 source groups (see Supplementary Note 6.1) using the mixmodel_ibd.R script (mixmodel_ibd) and plotted the resulting ancestry proportions in an admixture style plot using the script plot_mixmodel.R.

Supplementary Information 6:

Supplementary Note 6.1 - Mixture modelling source groups

AnatolianFarmers: Bon004, I0708, I1098, Tep002, Tep003, Tep004

BellBeaker: I13025, I13028, I5748

CHG: KK1, NEO281, SATP

Early Neolithic France: BUCH2, CB13, GLN246, GLN275, GLN284, GLN285A, GLN308, GLN309, GLN320, I0410, I0412, I0413, I4304, NEO812, R6, mur

IranNeolithic: AH4, GD13a, I1954, NEO816, WC1

Middle Neolithic Iberia: I8134, TOR6, LugarCanto41, I0406

Yamnaya: I0231, I0370, I0438, I0443, RISE547, RISE550

balticCentralEurHG: Donkalis6, KO1, Latvia_HG2, NEO307, PL_N22, Spiginas1

earlyCentralEurNeolithicLBK: I0025, I0026, I15818, I1904, I2739, Sch72-15

earlySpainHG: Chan, NEO694

russianHG: Latvia_MN2, NEO166, NEO167, NEO170, NEO171, NEO178, NEO179, NEO180, NEO184, NEO186, NEO189, NEO192, NEO193, NEO194, NEO195, NEO197

scandinavianNeolithic: CGG106494, NEO38, NEO43, NEO46, NEO744, NEO757, NEO896, ans003

ukraineHG: NEO270, NEO501, NEO521, NEO524, NEO552, Ukraine_N1

westernHG: Loschbour, PER1150503, R7, SRA62, Villabruna

Furthermore, we have included the full results from the mixture modelling analyses as Supplementary Tables 10 & 11.

5. Supplementary Materials: Missing Figures and Unclear Tables

Figure Inconsistencies:

As mentioned, several referenced figures are missing or misnumbered, including Supplementary Figure 7.4. This raises concerns about data accessibility and manuscript coherence.

Corrected. Please see our response above re. mislabeled supplementary figures.

Unclear Table Headers and Acronyms:

Supplementary Tables lack clear introductory descriptions. Column headers such as “nDer_woDam” are cryptic. If this refers to the number of derived alleles excluding those with deamination (i.e., only transitions), it should be explicitly stated. All acronyms should be written out to ensure accessibility for readers beyond the immediate research community.

We have fixed this by adding table legends for all Supplementary Tables, detailing ambiguous acronyms and clarifying unclear terminology.

This study does not provide the data quality, analytical transparency, or methodological rigor required to support its dramatic claims of a *Yersinia pestis*-driven demographic collapse in Neolithic Europe.

The genetic discontinuity observed at Bury is not convincingly linked to broader European trends.

The kinship analysis is based on limited and ambiguously interpreted data.

The pathogen evidence lacks rigor and plausibility.

The methods are insufficiently described to allow for replication or proper scrutiny. Supplementary materials contain missing and confusing elements that further weaken the study's credibility.

In sum, what is interpreted here as population collapse is more plausibly explained by cultural and mortuary transitions around 4500 BP. Future studies must apply far more robust analytical frameworks and uphold higher standards of scientific transparency before advancing such consequential conclusions

We hope that the edits listed above address all of the concerns raised by the reviewer. If not, we're happy to do further edits in a possible second review round.

Reviewer #3 (Neolithic/Bronze Age bioarchaeology)

Ramsøe et al. present archaeogenetic data from 133 human burials recovered from a the Bury megalithic tomb in northern France where there were two distinct phases of construction and burial separated by one to two centuries: at the end of the fourth Millennium BC and at the beginning of the 3rd millennium BC. They find that while the people buried in both phases showed a bias towards males, patterns of relatedness differed, suggesting different social rules dictating whose remains were placed in the tombs, with the first phase including selected members of wider genealogies, and the second including more infrequent groups of close genetic relatives. They also find that ancestries carried by groups in either phase varied, and patterns of IBD DNA segment sharing suggests that the people from the second phase were largely not descendants of those in the first phase and represented a new community of people. Previous osteological analysis of the assemblage had found that the demographic profile of the people in the first phase was indicative of a catastrophic mortality profile or one where the population was growing rapidly. Finally, they found that four individuals, three of the final burials from the first phase and one from the second phase showed evidence for having been infected with *Yersinia pestis* bacteria (plague) when they died.

The authors' interpretation of this sequence as the community represented in the first phase of activity having been largely killed off by plague before a new, mostly unrelated community with different patterns of social organisation moved in two centuries later and remodelled the tomb to bury their own dead. The authors argue this supports a model where disease and specifically a plague pandemic substantially contributed to the 'Late Neolithic decline' in Europe a period which sees evidence for population decline which they specifically link to the decline in the construction of 'complex megalithic architecture'. Similarly-dated individuals infected with *Yersinia pestis* from Orkney and Scandinavia have been argued to be the result of localised outbreaks related to zoonotic spillovers rather than indicative of a widespread pandemic, but the authors argue their evidence from Bury suggests that plague was widespread and deadly enough to be responsible for significant demographic decline across Europe.

Clearly an incredible amount of effort has gone into the data generation and computational analysis for this paper and the authors should be commended for this. In this aspect alone, this paper would be an impressive addition to the published literature. Archaeogenetics papers presenting results or even (as in this case) hundreds of ancient genomes from single sites have rapidly become the norm more recently, but the number of individuals included here, the age of the samples and the variability in patterns of organisation they find mean that the authors' paper is a cut above, providing some truly novel insights into communities who used the Bury tomb for burying some of their dead. It is the first time, to my knowledge that the differences between

two different phases of deposition at a tomb like this have been explored in this much detail, and it really provides fascinating evidence of reuse by a second community largely unrelated to the first. The authors use of IBD segment-sharing to look at continuity between the phase 1 and phase 2 population in the tomb is ingenious and provides a way forward for assessing these sorts of questions at other sites/monuments used by communities with similar broad ancestries. The paper draws upon standard established methods of data generation and analysis, is well written and the figures are all clear and appropriate.

Thank you for this endorsement.

While the standard of data and analysis is exceptional, I'm far from convinced by the interpretation, particularly in terms of broader arguments about a Late Neolithic decline and the association with disease and the decline of megalithic structures. The authors assert in no uncertain terms that the results from Bury when considered alongside recent related results indicate an 'event' at the end of 4th Millennium BC whereby a pan-European plague pandemic caused the collapse of megalithic tomb-building societies. In my opinion the authors' (still impressive) study of a human remains from a single megalithic tomb in the Paris Basin cannot sustain such an unequivocal and sweeping conclusion. The continental-scale framing of the significance of the results from Bury is, I think, unjustified and unnecessary. There is absolutely a wonderful paper in here which sets out quite a compelling narrative of the use of monument through time and the differing natures of the societies that used the tomb to bury their dead. Certainly given how much ink has been spilt on the reverence of ancestors in the European Neolithic, the finding that a tomb was remodelled and reused by a community that was largely not descended from the original builders is a fascinating and novel counterpoint. Similarly, the authors' study adds to the evidence for regional deviations in funerary behaviour in different tombs/regions, potentially suggesting that each community of tomb users organised themselves slightly differently, a tantalizing insight, However these aspects are underexplored in favour of a bigger narrative of disease-driven decline in Neolithic Europe.

These are of course very good points that resonate with comments from both the editor and all other reviewers. To address this we have rewritten large parts of the text, moving the focus away from the discussion of whether or not plague caused the Neolithic decline. In this new iteration of the manuscript, we focus on the effects of the Neolithic decline instead of its causes. Most importantly, we have added new mixture modelling results suggesting that Neolithic ancestry from Iberia moved northwards during the fourth millennium BCE, but only moved into the Paris Basin after the decline, during the third millennium. This change in ancestry coincides with the two use phases at Bury. In addition, we have focused on the differences in social organisation observed between the two phases. We believe that this new, rewritten manuscript offers a much better and much more nuanced representation of our data set.

I would certainly agree with the authors that their results from Bury are consistent with the 'Neolithic decline' scenario the authors describe, but given the low detection rate of *Yersinia pestis* at Bury and other possible reasons for communities stopping burying their dead in specific tombs, I don't think the new data the authors present here in of itself provides strong evidence in favour of a pan-European 'Late Neolithic decline fuelled by plague. The catastrophic mortality profile identified by osteological analysis at Bury is interesting and provides important context for the disease results, but these patterns are only straightforwardly interpretable in cases where the community are well-represented in the burial assemblage. Given at Bury there are numerous reasons (male bias, occurrence of close genetic relatives) to think this is not a representative sample of a community, the catastrophic mortality profile is less easy to interpret and doesn't translate straightforwardly into rapid population rise or mass death.

We agree. We have addressed this by toning down our plague interpretation considerably. We hope that this new version of the manuscript which presents the pathogens we found, without making any sweeping conclusions, better meets the expectations of the reviewer.

Given the number of samples the authors have analysed from Bury, the rate of *Yersinia pestis* infection is very low and unless the authors can provide analysis to the contrary it looks to me like the varied occurrence in different phases could have occurred by chance. In addition, the factors affecting the within-site, between-phase and between-site (and sometimes intra-skeletal) preservation of pathogen DNA are poorly understood and it is difficult to know to what extent within-site and between site comparisons of rates of disease represent genuine variation in prevalence or just preservation. I agree that the authors findings of *Yersinia pestis* at Bury adds to the impression that this disease was widespread, but given, as the authors reference, it has already been found in Orkney, a location that is exceptionally geographically peripheral, Therefore, I don't think the authors' results from Bury necessarily adds so much to the discussion of how widespread plague was in Late Neolithic Europe.

Good point. See our reply above re. the rephrased plague discussion.

I found the authors' discussion of the broader archaeological context of the 'Neolithic decline' and disuse of megalithic structures inadequate and simplistic. In the initial discussion it is unclear whether the authors are discussing megalithic structures generally (which would include things like stone circles and menhirs) or megalithic tombs specifically. Even if the latter, the situation is much more complex than the authors make out, with regional differences in use of megaliths and measure of demographic change. The obvious example is Britain and Ireland. In Britain, stone circles and Stonehenge specifically are being built during this supposed decline. Megalithic tombs are still being built in Orkney and Ireland into 3rd Millennium BC. I had wondered whether a more detailed discussion was included in the supplementary information. There was a well-written and interesting discussion of the development of Funnelbeaker-associated megalithic monuments,

but it is unclear how this related to the Bury tomb specifically. The authors make no effort to understand their results in their local or regional contexts. Given the regionality in markers of a Neolithic demographic decline, it would be interesting to interrogate how the results from Bury match regional trends, but this aspect is neglected in favour of grand narratives of disease and decline across Europe. The authors do allude to this regional variability, but this is very cursory.

Thank you for the comment. To address this, we have clarified that we refer to megalithic tombs specifically in the introduction. Moreover, as part of this revision, we have rewritten the discussion entirely. During this process, the discussion on megalith or collective graves was removed. The main problem with comparisons to nearby sites is that well documented comparative material from the Paris area is rare and possible comparisons are limited to few examples from Germany (see explanation in Chambon and Salanova 2025, pp 10-11).

Lastly, we have clarified, in the discussion, that the text focuses on continental Europe, and do not take Britain and Ireland into account:

When we add the anomalies in the demographic structure of the buried population in Phase 1 of Bury as well as other megaliths from continental Europe, there are further indications of demographic change (Supplementary Information 1). It supports a more widespread phenomenon covering most of central and Northwest Europe.

The authors' discussion also seems to conflate, the 'Late Neolithic decline' which is usually derived from measures of demographic decline with changes in the disuse/abandonment of megalithic tombs. I think it is reasonable to argue that the disuse of megalithic tombs is related to population decline, but given there could be lots of reasons why communities stopped using these monuments to bury their dead I don't think the authors can take this connection for granted. While their data and analysis are excellent, I don't think the authors have presented any definitive evidence for population collapse at Bury, only that one groups ceased using the tomb to bury their dead.

Thank you for your comment. We have addressed this in our newly added mixture modelling analysis, where we show that the populations in the Paris Basin were markedly different before and after the Neolithic decline. Additionally, this analysis demonstrated that the Phase 2-like population, and not the population from Phase 1, where the ones who mixed with steppe related people to create the 'Bell-beaker-like' ancestry at the end of the third millenium BCE:

When visualising major ancestry groups on a map (Figure 3), the mixture modelling reveals a stepwise northwards spread of this Neolithic Iberian ancestry (Supplementary Note 6.1). By 2,900 BC, populations across southern France and Iberia all constituted a large fraction of Iberian ancestry, while people in the Paris Basin still comprised mixed ancestry proportion as represented by the Phase 1 individuals. At some point after 2,900

BC, a final northwards push of the Iberian ancestry partially replaces the existing population in the Paris basin resulting in the homogenous population we observe in Phase 2. After the end of Phase 2, around 2,500 BC, steppe ancestry first appears in the Paris Basin²⁹ (Figure 3). After a couple of hundred years this steppe ancestry will gain the Atlantic coast and mix with local populations carrying the Iberian ancestry. As such, these results readily explain the difference between the populations of Phase 1 and 2, and could suggest an event that facilitated the northwards expansion of Neolithic Iberian ancestry at around 2,900 BC.

Furthermore, to address the comment on whether or not a demographic decline can be directly linked to the abandonment of burial sites, we have changed the wording in the introduction. In the new version of the text, we introduce the demographic collapse and the cessation of megalith building as two distinct events, that are not necessarily linked:

At the end of the fourth millennium BCE a demographic decline decimated populations across northwestern Europe³. At the same time construction of complex megalithic tombs ended for unknown reasons. Radiocarbon dates indicate a period of construction for these collective tombs around 4,300-3,100 BCE [...]

In sum, I don't think the results from Bury presented by the authors are enough to substantially tip the balance in discussions of the role of plague in the Neolithic decline and the abandonment of megalithic tombs in the way the authors argue. Normally I would say that the paper needs a major revision and reframing, but without the grander applications to disease and demographic decline I'm not sure a revised version of this paper would be well-suited to Nature Ecology and Evolution and perhaps would be better placed elsewhere.

We hope the revised text addresses this point. Please refer to the detailed answers above.

Some more specific comments related to the above:

Page 2, Line 1: Around the Late 4th Millennium '..., a significant population decline in Europe and halt in megalith building occurred...'

I appreciate this is only part of a summary, but this is too simplistic and should be toned down significantly. Megalithic building in Britain, for instance, continues well into the 3rd Millennium BC, for instance. If the authors are only referring to tombs (which they don't specify clearly here or later), than this is also not accurate – e.g. developed passage tombs in Ireland and Orkney.

This statement is also somewhat contradicted by their own paper given that Bury is a megalithic monument used and remodelled in 3rd Millennium BC.

Correct, we have changed the text to clarify that we are talking about NW Europe from Paris Basin to south Scandinavia. Furthermore, we are strictly referring to end of megalithic constructions, not their later re-use. The rephrased text now reads:

At the transition between the third and the fourth millennium BCE, a significant population decline and halt in megalith building occurred across Northwestern Europe.

Page 2, Line 2: 'Recent research that this 'Neolithic decline' linked to a plague outbreak and shifts in societal structure and genetic ancestry.'

This sentence is incomplete. Also 'associated with' rather than 'linked to' might be a more appropriate phrase here as 'linked to' implies some level of causality, which is debated.

Good point, we have removed this sentence entirely during the revision of the summary.

Page 2, line 4: 'To investigate, we sequenced 133 ancient genomes from the French semi megalithic allée sépulcrale at Bury, which spans two burial phases—...'

I think it would be useful for the authors to include which specific region of France. Also the authors could consider whether 'French' works when discussing a Neolithic monument rather than 'in present-day France.'

Corrected. We have rephrased the sentence to:

To investigate the population dynamics around the Neolithic decline in present-day France, we sequenced 133 ancient genomes from the allée sépulcrale at Bury. Located in the Paris area, Bury spans two [...]

Page 2, Line 7: 'Our analysis revealed societal changes and two distinct genetic groups between phases.'

I think it's always important in these sorts of studies to acknowledge that these are investigations of organisation amongst the dead, not the living. While certainly one is likely to have implications for the other, there is not necessarily a straightforward link. The authors find evidence for changes in the relationships between people buried in the tomb in different phases. I think there is a reasonable case to be made that this difference reflects differences in social organisation but I don't think the authors can say that their study directly reveals societal changes, rather evidence for differences in deposition and organisation which might reflect changes in society. I think it is necessary for the authors to lay out this reasoning, otherwise the assumptions they are making are not so transparent.

Good point. We have added a sentence to address this in the paragraph on pedigrees:

We acknowledge that these analyses only reflect organisation amongst the dead, not the living. However, we also note that the two are tightly linked and one is very likely to have implications for the other.

Page 2: 'The end of complex megalithic architectures in northwestern Europe has never been properly explained. Radiocarbon dates indicate a period of construction for these collective tombs around 4,300-3,100 BCE, followed by a pan-European decline of burial activities between 3,000 BCE to 2,900, depending on the area (Hinz et al., 2012; Shennan et al., 2013). These tombs are numerous throughout northwestern Europe, with very high concentrations in places such as the Paris Basin, Central Germany and southern Scandinavia. Many possible reasons have been put forward as contributing to this large-scale 'Neolithic decline'. Among the most prominent theories is that environmental exploitation brought about by farming, such as soil degradation and deforestation, reduced the land's capacity to support agriculture and livestock and thus its ability to support local populations (Colledge et al., 2019). Others argue that the reason behind the decline likely falls within the societal realm, including for example the strengthening of the idea of "households" (Furholt, 2021). Still others suggest that the close contact between humans and animals in the Neolithic increased the risk of pathogen emergence, which, together with the increased population density, increased the risk of transmission (Barrie et al., 2024; Rascovan et al., 2019).

I don't think this introduction to the Neolithic decline and the decline in the construction of megalithic tombs is adequate. It is too general and simplistic. The authors start by discussing the 'end' of complex megalithic architectures in northwestern Europe, but wouldn't this definition include megalithic sites which continue into the 3rd Millennium BC (stone circles/alignments, Skara Brae)? It becomes clear only later that they are referring to megalithic tombs specifically, but 1) this needs to be made clear from the beginning, and 2) this still doesn't work for places like Ireland and Orkney where megalithic tombs are being built into 3rd Millennium BC. I appreciate the authors acknowledge that evidence for 'decline' depends on the area, but I feel as though this variability is significantly underplayed in favour of depicting a broad pan-European trend of decline. Perhaps the authors are focussing on continental northwestern Europe to the exclusion of places like Britain and Ireland, but if so they should state this directly and qualify it when they are referring to 'pan-European' or references which discuss the Neolithic decline in Britain and Ireland as well as in continental Europe.

We have expanded the text to clarify this. The text now reads:

At the end of the fourth millennium BCE a demographic decline decimated populations across northwestern Europe³. At the same time construction of complex megalithic tombs ended for unknown reasons. Radiocarbon dates indicate a period of construction for these

collective tombs around 4,300-3,100 BCE, followed by a decline of burial activities in general between 3,000 to 2,600 BCE, depending on the area⁴⁻⁶. These tombs are numerous throughout northwestern Europe, with very high concentrations in places such as the Paris Basin, Central Germany and southern Scandinavia. They are collective burials that accommodated deceased as they died, and thus collected tens of thousands of dead in the second half of the fourth millennium BC⁷⁻⁹. However, the burial practices, in detail (position, orientation, intervention on skeletons after decay, grave goods...), all fluctuated according to chronology and cultural contexts. Constructions vary from so-called passage graves made from large boulders mainly found in northwestern Europe and southern Scandinavia, to long cist graves (gallery graves) made of stone slabs more common in the Paris Basin (allées sépulcrales) and central Germany (Galeriegräber) which were used by different archaeological groups: Seine-Oise for the allées sépulcrales¹⁰, Wartberg for the Galeriegräber¹¹, Bernburg for the Totenhütten¹² and the Funnelbeaker/Trichterbecher cultural complex for the Scandinavian passage graves¹³. Many possible reasons have been put forward to explain the large-scale decline in the construction of megaliths and their use towards the end of the fourth millennium BCE. Among the most prominent theories is that environmental exploitation brought about by farming, such as soil degradation and deforestation, reduced the land's capacity to support agriculture and livestock and thus its ability to support local populations¹⁴. Others argue that the reason behind the decline likely falls within the societal realm, including for example the strengthening of the idea of "households"¹⁵. Still others suggest that the close contact between humans and animals in the Neolithic increased the risk of pathogen emergence, which, together with the increased population density, increased the risk of transmission^{16,17}.

Moreover my understanding of the Neolithic 'decline' is that it primarily derives from and is defined by evidence of demographic change, with other factors such as megalithic monuments falling out of use seen as possible effects of this demographic change. However the way the authors have chosen to phrase this in their manuscript seems to begin with is the drop off in construction of megalithic monuments as indicative of some 'decline' in of itself. I don't think this is a reasonable position to hold given there could be lots of reasons why people might switch away from constructing megalithic tombs which doesn't involve a 'decline' of any sort, if for instance there was simply a shift in cultural preference. The way the authors bundle together demographic decline and the move away from the construction of megaliths takes for granted that the two are somehow inextricably linked and causal. The argument that demographic decline may be responsible for less monument building is reasonable, but the authors need to outline the reasoning for this, it can't be simply taken for granted and the way that it's expressed

here puts the cart (reduction in megalithic tomb-building) before the horse (demographic change)

Thanks for pointing this out, we have rewritten the beginning of the text to clarify. In the updated version we first introduce the demographic decline, and then the end of megalith building:

At the end of the fourth millennium BCE a demographic decline decimated populations across northwestern Europe³. At the same time construction of complex megalithic tombs ended for unknown reasons. Radiocarbon dates indicate a period of construction for these collective tombs around 4,300-3,100 BCE [..]

Given the authors acknowledge that measures of Late Neolithic decline and monument building are regionally variable across Europe, it is surprising they don't provide an account of decline in the particular region (Paris basin) where their tomb was found. It would be much easier to assess the narrative they propose for the Bury site if we had an understanding how that fits within the local, regional sequence rather than how it might vaguely fit into a pan-European phenomenon. Again, it feels like the local/regional contextualisation of the (extremely interesting and important) results are neglected in favour of serving a grand narrative.

Good point. Since the initial submission of this paper, a thorough comparison of the Bury tomb with other sites from the Paris Basin have been published (Chambon and Salanova 2025). We have cited this paper in the introduction, and highlighted all genetic studies of contemporaneous sites from the Paris Basin in the results section. Furthermore, we added a new pollen botanical summary focusing on the Paris Basin which shows forest regrowth in the period after the end of the first phase (Supplementary Information 2).

Salanova, L. & Chambon, P. Demography from Late Neolithic graves NW of Paris. *Praehist. Z.* (2025) doi:10.1515/pz-2024-2056.

Page 2: 'The presence of *Yersinia pestis*, the etiological agent of plague, in the Bronze Age is well established from ancient DNA research and was, until recently, the earliest known form of the pathogen (Andrades Valtueña et al., 2022; Rasmussen et al., 2015). However, the subsequent identification of earlier lineages of *Y. pestis* has demonstrated that plague was afflicting Neolithic communities already before the expansions from the Eurasian steppe (Rascovan et al., 2019).'

I don't think the various factors of the debate around Late Neolithic-Bronze Age *Yersinia pestis* are elaborated well here, and it means that the introduction of human expansions from the Eurasian steppe comes across as a bit of a non sequitur without a certain amount of prior knowledge on the part of the reader. I think the authors should add some detail about the fact that initially the Bronze Age *Yersinia pestis* lineages seemed to coincide with influence human

expansions out of the Eurasian steppe, which in turn was used to suggest that these expansions had something to do with the appearance of this disease, but that since then older plague genomes have shown that it was present prior to these movements off the steppe.

Corrected. The rephrased sentence now reads:

The presence of Yersinia pestis, the etiological agent of plague, in the Bronze Age is well established and the disease was, until recently, assumed to have been spread by the migration from the Pontiac steppe into Europe^{18,19}. However, the subsequent identification of earlier lineages [...]

Page 2: 'As the age of these genomes coincides with the end of the fourth millennium BCE, it has been suggested that a possible plague pandemic could have contributed to a population collapse around 3,100 BCE (Rascovan et al., 2019).'

Again, this is treating the Late Neolithic decline as a singular homogenous pan-European process when I don't think there is the evidence to take that for granted.

Good point. To clarify we have moved the last sentence of the paragraph up which highlights that it is not known whether this was just a regional outbreak:

As the age of these genomes coincides with the end of the fourth millennium BCE, it has been suggested that a possible plague pandemic could have contributed to a population collapse around 3,100-3,000 BCE^{16,20}. However, it remains unknown whether this regional outbreak was part of a larger pandemic across Europe during the Neolithic period.

Page 3: 'However, it remains unknown whether this regional outbreak was part of a larger pandemic across Europe during the megalithic period.'

The authors later mention the detection of Yersinia pestis DNA in a skeleton from the Banks tomb in Orkney. Why is this not mentioned here as background information? Wouldn't the (geographically) peripheral location of Orkney mean that it is already established to some extent that Yersinia pestis must have been widespread in this period, even if this could be because of regular zoonotic spillovers rather than a pandemic?

Good point. The detection of plague from Orkney have now been published (<https://www.nature.com/articles/s41586-025-09192-8>), but since the date is a little uncertain we have removed the discussion of these plague cases from the text.

Page 3: 'The semi-megalithic allée sépulcrale of Bury, located 50 km north of Paris, is from a region where many other collective graves have been recorded (Chambon & Salanova, 1996).'

Given the centrality of megalithic architecture to this paper, it would be useful for the authors to define what they mean by 'semi-megalithic' here, given it is not a common phrase.

Corrected, we have clarified what ‘allée sépulcrale’ means in the introduction, and avoided the use of ‘semi-megalithic’ as it might cause unnecessary confusion. The paragraph now reads:

Located 50 km north of Paris, the Neolithic burial site of Bury is from a region where many other collective graves have been recorded⁶. As a semi-underground monument of rectangular shape, Bury is a classic example of the allées sépulcrales found north-west of Paris, built with a combination of megalithic slabs and other techniques, like dry-stone walls. The Bury grave held primary burials of 316 individuals [...]

Page 3: ‘The first phase at the end of the fourth millennium (3,500 - 3,000 BCE) was interrupted shortly after its beginning (Salanova et al., 2018), while the phase covered several centuries over the third millennium (2,900 - 2,470 BCE).’

Is there a missing ‘second’ here? ‘...while the second phase covered...’

Corrected

Page 3: ‘Rather, the demographic profile is suggestive of excess mortality, particularly affecting juvenile individuals, perhaps indicating a catastrophic event, such as war, famine or a disease outbreak or, on the contrary, a rapid increase in the population. Hinz specifically points to decline from around 3350 cal. BC in northwestern Europe, which is well before the first phase at Bury.’

I think it would be useful if the authors could comment in slightly more detail on the nature of the deposits of human remains at Bury. Do they have a sense of whether whole bodies were left to decompose decompose in the tomb before being disturbed and commingled, or is there any indication some disarticulated bones were brought in from elsewhere? I think this is important in understanding how much selection was involved in whose remains ended up in the tomb. The catastrophic mortality profile is interesting here and relevant to the discussion of plague, but it is only easily interpretable if the people buried in the tomb were representative of who was dying more generally. I think it would be useful for the authors to caveat that if there was some selection of bones/individuals then the meaning of the mortality profile is difficult to discern.

Yes, good point. Field observations, reconstruction of skeletons in the laboratory and osteological profiles clearly indicate that we dealt with primary burials (for further details, see Salanova et al. 2017 and 2018, Chambon and Salanova 2025). We have clarified this in the introduction, which now reads:

The Bury grave held primary burials of 316 individuals, divided in two main burial phases¹. The first phase was used over a relatively short period at the end of the fourth millennium (around 3,200-3,100 BCE)², while the second phase covered several centuries over the third millennium until 2,470 BCE.

Page 6: 'These results suggest that the individuals of phases one and two form genetically distinct communities.'

As the authors state later that they couldn't rule out minor population continuity between phase 1 and 2, is genetically 'distinct; a bit too absolute? Genetically differentiated maybe?

We agree, as part of the revision, we have removed this sentence.

Page 8: 'We use them to describe biological relatedness, and do not imply that these terms, nor the notion of family, was understood in the same way by the population using the grave at Bury during the Neolithic times.'

I feel as though there is something missing from this sentence in terms of anachronistic project of present-day understandings of family. Perhaps '...was understood by the population using the grave at Bury during the Neolithic times in the same way as it is understood in Western societies today.'

Thank you for the suggestion, we have incorporated this edit into the text which now reads:

Furthermore, we use the term pedigree, along with kinship terms such as "mother", "daughter" to describe biological relatedness, and do not imply that these terms, nor the notion of family, was understood by the population using the grave at Bury during the Neolithic times in the same way as it is understood in Western societies today.

Page 8: 'This ties in with the observation that there are no exogenous individuals in the grave.'

I found this sentence and the passage before it a bit confusing. Earlier the authors state that 'we find a complete lack of any exogenous individuals (aside from the first generations), implying that, other than the totally unrelated individuals, only people genetically descended or linked to others are buried in the grave.' How are the authors defining 'exogenous individual' here? By 'exogenous' do they mean people from outside the genealogy that had children with people inside the genealogy? If these people do exist in the first generation then how does this support the statement that there are no exogenous individuals in the grave? Are totally unrelated individuals not 'exogenous'? This needs clearing up.

We agree, it was worded very confusingly. We do indeed mean people from outside the genealogy that had children with people inside the genealogy. We have reworded this:

We observe no cases where males from outside the family line entered it by having children with its members. Aside from the founding generation of each pedigree, all subsequent males are genetically descended from earlier members.

Page 8: 'The two sequenced sons of ID222, ID291 and ID275 both show long runs of homozygosity (Supplementary Figure 7.5), indicating some possible relatedness between their mother

(unsampled) and father ID222. One of these sons, ID291, has a further son (unsampled), who conceived three children with one female, ID262, who is a third or fourth degree relative to ID291 - her reproductive partner's biological father. One of the sons of this union (ID316) also exhibits long ROH segments. Incidentally, ID262 is also the only female buried in the grave who has both ancestors and children also buried at Bury.'

These are some tantalizing findings especially as high parental relatedness seems to be relatively uncommon across Neolithic Europe (Ringbauer et al. 2021). It's a shame that this is not discussed further in the Discussion section, again it feels as though this was neglected to focus on the grander narrative. Also, doesn't the heightened frequency of parental relatedness in phase 1 compared to phase 2 run contrary to there having been a significant pan-European population collapse? Wouldn't increased frequency of related parents be more likely in a smaller population? Unless the authors think this was more to do with social organisation in phase 1 compared to phase 2.

Good point, our main hypothesis is that the difference between the two phases at Bury reflects a population replacement rather than a population decline in the same population. The observations listed above align well with this interpretation (see answer below for more details).

On a similar point, did the population from phase 2 show higher frequencies of shorter runs of homozygosity than phase 1, indicative of a smaller effective population size? This is what we might expect to see if there had been a substantial population decline. I suppose the authors could argue that the population in phase two was 'new' and moved in from a different region, but if the authors are trying to support a pan-European model of population decline, this shouldn't make that much difference.

No, in fact we see the opposite: Phase 1 generally have longer ROHs indicative of a small population size (see the updated Supplementary Figures 6.5a and 6.5b). A smaller population size in Phase 1 is also confirmed by IBDne results (Supplementary Figure 6.17), suggesting that the effective population size was significantly larger for Phase 2 than Phase 1. These results agrees with our hypothesis of two distinct populations at Bury.

Page 10: 'The predominance of a large genetic group during the first phase, probably related to the other small groups, testifies to the control of a group on the collective grave, or to the links that could exist between the inhabitants of the same place.'

Being slightly picky – 'control' implies a particular interpretation which I don't think the authors have justified here. Other situations could easily be imagined where individuals from these genealogies were interred in the tomb by consent of a wider community or indeed chosen to be interred there by a different group who controlled access to the tomb.

Corrected. We have rephrased the sentence which now reads:

The predominance of a large genetic group during the first phase, probably related to the other small groups, suggests a strong association between this group and the collective grave, or to the links that could exist between the inhabitants of the same place.

Page 12: 'Furthermore, the grave only represents a subset of the population, and it is perfectly possible that a severe plague epidemic would leave very little evidence behind, if the entire population perished without being buried or if they were buried elsewhere.'

This is a reasonable comment to make but rather implies that the evidence the authors have found at Bury is not substantial enough to support the scenario of a widespread plague pandemic by itself and requires some extra justification. Moreover, I think there really needs to be more discussion of the uncertainties involved with the recovery of pathogen DNA. Granted, it's likely that there are a larger number of false negatives, but how much larger may be highly dependent on context/site/regionally-specific preservation dictated by factors that are not well-understood. So, I think it is much more difficult than the authors convey to interpret what rates of particular pathogens mean when comparing sites or even phases.

We agree. The most significant modification we have made to the manuscript during the review process was to dramatically tone down the focus on the plague narrative. As such, the current version of the paper does not attempt to draw any grand conclusions from the plague data.

Page 14: 'These factors collectively point to what should be termed a Neolithic Decline or Dark Age. It is also evidenced in a decline in the production of fine pottery in Northern Europe, and other technological skills.'

'Dark Age' is an extremely contentious phrase even when applied to its original early medieval context. It is commonly regarded to relate to the lack of textual sources rather than grim circumstances of people who lived through it (although it has since come to be associated with that aspect too). Periods of prehistory can be described and discussed on their own terms without having to reach for phrases from totally different periods/contexts. Therefore, 'Dark Age' is an entirely inappropriate and anachronistic term to use here and I think the authors should strongly reconsider their using it. In addition 'should be' sounds very prescriptive ('could be' might be more appropriate) – it falls to the broader scholastic community to decide whether 'Neolithic decline' is a useful term. The reference to the production of fine pottery and 'other technological skills' is an interesting component but given this is the first mention of this in the context of the Neolithic decline, it is extremely vague as well as being unreferenced. I think it certainly should be discussed in more detail alongside the decline in tomb-building, properly referenced, and potentially in the Introduction, rather than brought up off-handedly in the Discussion.

We take the point, and avoid Dark Age. We have also removed the sentence on pottery.

Page 14: 'There is also evidence from megaliths in northwestern France and central Germany that burials ended around 3,100 BCE.'

This requires a citation.

Yes, agree. We have cited four papers to support this statement:

Salanova, L. & Chambon, P. Demography from Late Neolithic graves NW of Paris. *Præhist. Z.* (2025) doi:10.1515/pz-2024-2056.

Brozio, J. P., Müller, J., Furholt, M., Kirleis, W., Dreibrodt, S., Feeser, I., Dörfler, W., Weinelt, M., Raese, H., & Bock, A. What drove their variability in the middle-Holocene Neolithic? *The Holocene* 29, (2019).

Ricardo Fernandes, Christoph Rinne, Pieter M. Grootes, Marie-Josée Nadeau. Revisiting the chronology of northern German monumentality sites: preliminary results. in *Frühe Monumentalität und soziale Differenzierung Band 2* (ed. M. Hinz, J. M.) (2012).

Pape, E. A Shared Ideology of Death? The Architectural Elements and the Uses of the Late Neolithic Gallery Graves of Western Germany and the Paris Basin. (2019).

Page 14: 'All the observations point to a coherent narrative. An important event occurred in Northwestern Europe at the end of the fourth millennium. The collective burials, essentially megalithic, correspond to a demographic increase and a high population density (Salanova & Chambon, 2025), which stopped around 3,100 BCE. After a couple of centuries of interruption, the use of the Bury grave during the whole third millennium is different, both in ritual and population structure. The population decline in the collective grave and the plague pandemic could have made it easier for steppe populations to colonize new lands westwards after 2,900 BCE.'

I think it's reasonable to make this argument, but I think most of the evidence the authors discuss here is largely already published and the novel results from Bury are too ambiguous to substantially support the strong and wide-ranging conclusions the authors come to here.

We agree. To address this, and to reflect all the changes made to the manuscript we have completely rewritten the end of the discussion:

Given the temporal hiatus that separates the two phases at Bury, the supporting genetic evidence suggesting discontinuity between the phases, and the evidence of reforestation (Supplementary information 2), we can tie this into a wider European picture in order to

trace potential signals of demographic decline in the late Neolithic⁴⁷. The same hiatus in burial sequence is known in similar graves from Germany^{48,49}. There is also evidence from megaliths in Northern Germany that ended around 3,100-3,000 BCE, followed by decreasing open land^{13,21,50,51}. Additionally, we present evidence of precise construction dates for Danish passage graves ending around 3,000 BCE (Supplementary Information 4). When we add the anomalies in the demographic structure of the buried population in Phase 1 of Bury as well as other megaliths from continental Europe, there are further indications of demographic change (Supplementary Information 1). It supports a more widespread phenomenon covering most of central and Northwest Europe. The construction of collective burials correspond to a demographic increase and a high population density among Neolithic societies in Northwest Europe²¹, which stopped around 3,100 BCE, as exemplified in Bury. After a couple of centuries of interruption, the use of the Bury grave during the whole third millennium is different, both in ritual and population structure, as in most of Northwest Europe. It should be stressed, however, that the Neolithic Decline is not synchronous across Europe, even if it falls within a timespan from roughly the late fourth into the beginning of the third millennium BCE. Within this time span there are temporal and geographical variations⁵², and we should expect that while a gradual reduction of populations due to both pathogens, and other environmental factors, may have contributed by making Neolithic populations more vulnerable^{47,52}.

We may thus consider the possibility that both the Iberian northward migration, and the expansion from the steppe were related responses to the Neolithic decline in temperate Europe.

References

Ringbauer, H., Novembre, J. and Steinrücken, M., 2021. Parental relatedness through time revealed by runs of homozygosity in ancient DNA. *Nature communications*, 12(1), p.5425.

Reviewer #4 (ancient genomics):

The manuscript by Ramsøe et al. analyzes 133 genomes from the Bury megalith site in the Paris Basin. The archaeological and anthropological integration is very solid and effectively contextualizes the results, especially regarding the biological kinship. The laboratory methodology and sample quality are excellent. However, some methods require a more detailed description and the application of additional statistical analyses, although they are generally adequate to support the discussion of the results.

Nevertheless, it seems that the objective of this case study is to argue for a population decline at a European scale. While the case study is highly interesting and of sufficient quality for publication, suggesting that the phases of a single burial site define the scope of a population decline at the continental level is a major claim. This is a hypothesis that can certainly be discussed but not asserted with the degree of certainty presented here.

Below are specific observations by section. For resubmission, it would be advisable to include line numbering.

INTRODUCTION

In the sentence:

“Radiocarbon dates indicate a period of construction for these collective tombs around 4,300-3,100 BCE, followed by a pan-European decline of burial activities between 3,000 BCE to 2,900, depending on the area (Hinz et al., 2012; Shennan et al., 2013).”

- Specify whether the mentioned decline refers exclusively to “collective burial activities” or to funerary activity in general.

In the Paris Basin, there is no funerary activity dated from the beginning of the third millennium BCE, despite that some settlements are known, and as such we refer to burial activities in general. We have clarified this in the sentence in question:

Radiocarbon dates indicate a period of construction for these collective tombs around 4,300-3,100 BCE, followed by a decline of burial activities in general between 3,000 to 2,600 BCE, depending on the area⁴⁻⁶.

In the statement:

“These tombs are numerous throughout northwestern Europe, with very high concentrations in places such as the Paris Basin, Central Germany and southern Scandinavia.”

- Indicate the cultural affiliations associated with these tombs for greater precision. Without a clear specification of the type of megalithic tomb, this statement could also apply to regions such as southern and western France, as well as the Iberian Peninsula, where such structures are also widely documented.

Thank you for the suggestion. To address this we have expanded on the different types of megalithic graves and outlined the cultural complex associated with each:

These tombs are numerous throughout northwestern Europe, with very high concentrations in places such as the Paris Basin, Central Germany and southern Scandinavia. They are collective burials that accommodated deceased as they died, and thus collected tens of thousands dead in the second half of the fourth millennium BC⁷⁻⁹. However, the burial practices, in detail (position, orientation, intervention on skeletons after decay, grave goods...), all fluctuated according to chronology and cultural contexts. Constructions vary from so-called passage graves made from large boulders mainly found in northwestern Europe and southern Scandinavia, to long cist graves (gallery graves) made of stone slabs more common in the Paris Basin (allées sépulcrales) and central Germany (Galeriegräber) which were used by different archaeological groups: Seine-Oise for the allées sépulcrales¹⁰, Wartberg for the Galeriegräber¹¹, Bernburg for the Totenhütten¹² and the Funnelbeaker/Trichterbecher cultural complex for the Scandinavian passage graves¹³.

RESULTS

The site shows significant similarities with Fleury-sur-Orne (Rivollat et al., 2022), both in the predominance of males and in the diversity observed in Y-chromosome haplogroups and certain levels of background relatedness, based on the accumulation of low ROH levels. Since Fleury predates both phases of Bury, it would be advisable to integrate these analyses with the new data obtained from Bury for a better comparison.

Thank you for your suggestion. In order to integrate this data with our panel, we downloaded all available data from Fleury-sur-Orne (Rivollat 2020 SciAdv, and Rivollat 2022 PNAS). Unfortunately, all sequenced individuals from these two studies have very limited sequencing data available owing to the combination of relatively shallow sequencing and the focus on few capture sites. We have set our pipeline up to exclude all samples with less than 0.1X coverage across all sites, which means that captured samples need significantly higher coverage at captured sites to make this cut. To be consistent, we decided to maintain our 0.1X cutoff, and instead cite these papers in the main text.

- Figure 2a: The Kernel density calibration curve for phase 1 and phase 2 shows some overlap. What specific results support the existence of the 100-200-year chronological gap mentioned in the abstract?

The modelling of radiocarbon dates indicates a short use for phase one, ending before 3,100 BCE (low hypothesis, Salanova et al. 2018). The oldest interval that corresponds to the second phase

of burials does not begin before 2,900 BCE (2σ) (Salanova et al. 2017 and Supplementary table 4). Moreover, there is also strong archaeological evidence of a hiatus in the grave between phases 1 and 2, according to the organisation of the bodies, stone features and the category of grave goods. To clarify, we have added these references to the abstract.

- Figure 1: The image is too large for the amount of information it provides. It is suggested to reduce its size and combine it with a geographical map that includes the location of Bury and other similar sites or those co-analyzed in the study.

Good point. As suggested, we have recreated the figure ensuring that the grave schematics take up less space, and included a new map:

Figure 1. Overview of the Bury grave. a) Location of Bury and similar sites with genetic data available with the geographical extent of the Paris Basin highlighted. **b)** Schematic overview of the Bury grave during Phase 1. **c)** Schematic overview of the Bury grave during Phase 2 (from Chambon and Salanova 2025²¹).

- Phase 1 seems to exhibit different levels of HG ancestry compared to phase 2, which could indicate a more recent admixture date. The authors are recommended to use DATES or similar software to assess whether phase 1 shows a more recent admixture with HG compared to phase 2, where the HG ancestry proportions appear more homogeneous in the population. This analysis could help determine whether there was no additional HG contribution in the second phase, which in turn could reflect a decline in the final HG population.

Thank you for the suggestion. We ran DATES using the two source populations ‘earlySpainHg’ and ‘earlySpainFranceNeolithic’ which were identified in our mixture modelling analysis as the best sources for HG and Neolithic ancestry, respectively, in the Bury individuals. As depicted below, this analysis demonstrated that a single pulse of HG ancestry at around 4,250 BCE (6,200 BP) is the most likely scenario for both phases:

Supplementary Figure 6.13. Dating of Hunter-gatherer admixture. Estimates of the timing of hunter-gatherer and Neolithic admixture in individuals from Bury using DATES⁴. As source groups we used the two populations ‘earlySpainHg’ and ‘earlySpainFranceNeolithic’ which were identified in our mixture modelling analysis as the best sources for HG and Neolithic ancestry, respectively, in the Bury individuals (see ‘Mixture modelling’ above). Most likely sample ages are depicted as grey circles (phase1: 3,250 BCE, phase2: 2,650 BCE). While no time point exists where all confidence intervals overlap, we have highlighted the most likely admixture time in the grey rectangle (4,248-4,262 BCE).

- The article could benefit from a supplementary PCA that includes other Neolithic individuals from France, Germany, and southern Scandinavia. The distribution of these populations in the PCA would help identify which group is genetically closest to the individuals from Bury.

As requested, we included a PCA of all Neolithic individuals in our reference data:

Supplementary Figure 6.12. PCA of Neolithic individuals. Individuals with more than 50% Neolithic ancestry dated to before 2,550 BCE were plotted. Reference data from France, Spain and Portugal are represented by squares, whereas all other individuals are represented by circles. Phase 1 and Phase 2 individuals from this study were highlighted in green/brown color, respectively. **a)** Full plot with groups of individuals from the British Isles, Northern Europe and Western Asia highlighted. **b)** Insert focusing on individuals from Bury, with reference data from Seguin-Orlando et al. 2021 highlighted³.

As depicted, this PCA clearly differentiates the major groups within Neolithic Europe. PC1 mainly separates individuals based on age, with early Anatolian farmers towards the left, while PC2 differentiates between the various ancestries of Western Europe. As expected all individuals from Bury fall within the diversity of contemporaneous genomes from France, Spain and Portugal (highlighted with squares). Furthermore, the higher diversity in phase 1 becomes clear, as Phase 1 individuals are scattered over a larger area, while Phase 2 individuals cluster in the center of the plot.

IBD RESULTS

- Clarify which individuals were added to the customized dataset on which the GLIMPSE imputation was applied.

Corrected. The rephrased text now reads:

Imputation was carried out on all genomes generated for this study with GLIMPSE⁶⁶ using the phased data from the 1,000 genomes⁶⁷ dataset as reference. To investigate fine scale genetic structures within each population, we merged these imputed genomes with our imputed reference panel (Supplementary Table 9), and filtered genotypes retaining [...]

- Discuss the criteria for the clustering levels: are they based on the length or the number of IBD segments? Additionally, in a broad-scale analysis, shouldn't close relatives be excluded to avoid biases?

As part of this revision we have updated our clustering approach to use the base R function `hclust` (with the "ward.D2" method). In the updated clustering workflow we have also implemented a user-specified tree height cut-off for cutting the final hierarchical tree. We tested various cutoff heights on the Bury dataset (0, 2000, 4000, 5000, 10,000, and 50,000) and found that a cutoff of 4,000 best balances our need for capturing fine-scale population structure, while still retaining sufficient sample size within each cluster for downstream mixture modelling. We have clarified this in the methods section on clustering which now reads:

Lastly, using this dataset of unrelated individuals, we removed short IBD segments (<2cM) and clustered genomes of similar ancestry using hierarchical clustering on a euclidean distance matrix of total IBD sharing vectors between all pairs of individuals. Final cluster memberships were extracted using dynamic branch cutting implemented in the 'dynamicTreeCut'⁶⁹ package in R, with a cluster height cut off of 4,000.

Concerning closely related individuals: Yes, they were removed prior to clustering. We have clarified the exact approach to remove close relatives in the methods:

To filter out closely related individuals, we converted all pairs of individuals sharing more than 800 cM IBD into a graph object, and calculated the maximal set of independent vertices using the `max_ivs` function from the R package `igraph`. For each graph, we calculated maximal sets of unrelated individuals and retained the set with the highest number of individuals. In cases where multiple sets were of equal size, we retained the set with the highest mean coverage.

In the statement:

"These results suggest that the individuals of phases one and two form genetically distinct communities."

- It is recommended to rephrase this more cautiously. As mentioned in the study itself, in the first cluster there are individuals from both phases, and in the third clustering level, close relatives are grouped together. Even if they are different communities, this result cannot automatically be extrapolated to a population scale to claim continuity or discontinuity at population level. It is likely that if another Neolithic site from France were added, it would form its own cluster at the same resolution, without implying a population discontinuity.
- Instead of suggesting population discontinuity (here referred as "distinct communities" but later discussed as "population discontinuity"), it would be more appropriate to indicate that a different community made use of the megalith in a later phase. These are distinct concepts. Community discontinuity cannot be directly extrapolated to population discontinuity/decline.

We agree, as part of the revision, we have removed this sentence.

Furthermore, we believe that our new paragraph on the regional ancestry patterns that reflect the two populations at Bury further support our interpretation of two distinct communities at the site.

A stronger statistical justification for the clustering method is required. The following is recommended:

1. Include a matrix representing the total number of analyzed individuals from the sites considered showing the total amount of shared IBD.
2. Apply statistical tests to determine whether individuals within a cluster are significantly more related to each other than to individuals in other clusters.
3. Even if a significant pattern is observed, its interpretation should be nuanced, avoiding conclusions at the population level, as these groups may belong to the same source population.

Thank you for the suggestion. As requested, we generated a heatmap of total IBD sharing among all genomes generated as part of this study, and stratified the plot by genetic cluster (Supplementary Figure 6.16a). To test whether individuals within a cluster are significantly more related to each other than to individuals in other clusters, we also plotted total IBD sharing within and between clusters in a boxplot (Supplementary Figure 6.16b). We applied a Wilcoxon rank-sum test on the data points within and between groups and found a statistically significant difference between the two (p-value < 2.2e-16):

Supplementary Figure 6.16. Validation of the hierarchical clustering. *a)* Heatmap of shared IBD between all samples generated for this study stratified by genetic clusters. *b)* Violin plot comparing the amount of shared IBD within and between clusters (p -value < $2.2e-16$, Wilcoxon rank-sum test).

Correction in supplementary figures: All figures in Supplementary 5 and Supplementary 6 are numbered in the same way (Supplementary Figure 6.1 to 6.7), which causes confusion. Additionally, Supplementary Figure 7.4, mentioned in the text, does not exist.

Thank you for noticing this error, it has now been corrected so that figures from Supplementary 5 are numbered 5.1 to 5.8 while figures from Supplementary 6 are numbered 6.1, 6.2, 6.3, and so on. We have also fixed the instances where “Supplementary Figure 7.4” was used instead of “Supplementary Figure 6.4” in the main text.

ANCESTRY MODELLING

- Specify whether the ancestry modeling is based on qpAdm or supervised ADMIXTURE, as this information is missing from the methodology.

Good point. Following a comment from reviewer 2, we have added a new Methods section describing our mixture modelling approach (see above for details).

- Explain why only Iberian Peninsula HGs were chosen as a proxy. These individuals retain more Pleistocene-Magdalenian-associated ancestry than Central European HGs, but the study focuses on the Paris Basin, Central Germany, and Southern Scandinavia, where this ancestry barely persists.

For the mixture modelling plotted in Supplementary Figure 6.1, all distal sources groups (AnatolianFarmers, CHG, IranNeolithic, balticCentralEurHG, earlySpainHG, russianHG, ukraineHG, westernHG) were included in the modelling as potential sources. Yet, the Bury individuals were modelled exclusively with varying proportions of ancestry from the populations ‘earlySpainHG’ and ‘AnatolianFarmers’, suggesting that ‘earlySpainHG’ is the best source for the hunter-gatherer ancestry in the Bury individuals. We have clarified this in the figure legend which now reads:

Supplementary Figure 6.1. Modelled ancestry proportions from Bury. *a)* Mixture modelling results using the following groups as source populations: AnatolianFarmers, CHG, IranNeolithic, balticCentralEurHG, earlySpainHG, russianHG, ukraineHG, westernHG. All individuals were modelled exclusively with varying proportions of ancestry from ‘earlySpainHG’ and ‘AnatolianFarmers’, suggesting that these two populations are the best sources for the hunter-gatherer and Neolithic ancestries in the Bury individuals. See Allentoft et al. (2024) for details. *b)* Proportion of hunter-gatherer (source: earlySpainHG)

ancestry between the two phases at Bury. ***Wilcoxon rank-sum test: p-value = 2.281e-07

- To better interpret the origin of the HG ancestry resurgence, it is suggested to perform models using local HGs from these regions and compare the fit of both models. This would help determine whether the HG ancestry comes from local groups or from Neolithic populations with a higher proportion of HG ancestry.

See above.

- A statistical test should be included to determine whether the increase in HG ancestry between group 1 and group 2 is statistically significant.

Good point. We ran a Wilcoxon rank-sum test and found a significant difference between Phase 1 and 2 (p-value = 2.281e-07). We added this information in the figure legend, and added asterisks to the boxplot to indicate this.

Genetic Relatedness Results

- It is necessary to cite/discuss that no multiple sexual partners were identified in the analysis, similar to Gurgy (Rivollat et al. 2024), because it is not always the norm (e.g in Hazelton).

We agree, we have addressed this in the results section:

This is the only example of half-siblings in the dataset, which implies that having two reproductive partners was not the norm, or that the progeny from these unions were buried elsewhere, and thus also perhaps not socially sanctified. This finding is relatively rare among comparable genetic studies, but not unheard of. While no half brothers were found in the Neolithic site Gurgy from present day France, half brothers have been identified in Neolithic sites from the British Isles³⁴ and Scandinavia²⁰, and from later sites^{35,36}.

- Clarification on exogenous individuals:

o The main text states that "we find a complete lack of any exogenous individuals (aside from the first generations)", but Supplementary Figure 6.3 includes individuals not connected to the pedigree. Are these individuals not considered exogenous? It should be clarified what exogenous individuals means.

Thank you for noticing this error. What we meant was that we found a complete lack of any exogenous *males*. We have rewritten the sentence in question:

We observe no cases where males from outside the family line entered it by having children with its members. Aside from the founding generation of each pedigree, all subsequent males are genetically descended from earlier members. On the other hand, all

females except one are related exclusively to their offspring, suggesting a high level of female exogamy.

- **Strontium (Sr) data:**

- o Sr data are presented in the Supplementary, but they are not mentioned in the main text. If they are not used in the analysis, they should not appear in the Supplementary, or alternatively, they should be integrated into the discussion to assess the presence of non-local individuals.

Corrected. We have integrated the Strontium data in the discussion as suggested. This new paragraph reads:

Furthermore, our strontium isotope data depicts a more stable and sedentary population in Bury than earlier Neolithic groups from the area (Supplementary Information 5). Tied in with our findings on the replacement of the population between Phase 1 and Phase 2, this suggests that newcomers did not immediately reuse the tomb. Still, we do observe some mobility with a total of 14 Strontium outliers identified (6 from Phase 1, and 8 from Phase 2) most of which are either unrelated individuals or exogamous females.

- **Errors in Supplementary Figure 6.3:**

- o Some mitochondrial haplogroups are not transmitted to the next generation, suggesting that certain pedigrees are incorrect (this occurs in pedigrees 2A, 2B, and 2G). It is recommended to review mitochondrial transmission consistency and correct erroneous structures.

Thank you for noticing this. It was an error that occurred when copy pasting from an old iteration of the figure. We have corrected the errors mentioned, and we have gone over all pedigrees to double check for consistency in mitochondrial haplogroups.

- o The legend for Figure 6.3 should be more descriptive, clearly specifying what circles and squares represent and how mitochondrial and Y-chromosome haplogroups are marked. The age at death is a crucial data point that should be added to the pedigree.

Corrected. We have updated the figure legends for figure 6.2 and 6.2, which now reads:

Supplementary Figure 6.3. Pedigrees from phase 2. *Circles and squares represent females and males, respectively, while color specifies burial phase (green: phase 2, green/brown: unknown phase). Inside each shape, individuals are labelled with their mitochondrial haplogroup (first line), chromosome Y haplogroup (second line) and calibrated median age (last line). Furthermore, solid black lines between shapes indicate well defined first degree relationships, while stippled black and grey lines specify unknown or uncertain first and second degree relationships, respectively.*

Furthermore, we have added info on age of death for the three children identified among the individuals with sequencing data (BUR189, BUR182, and BUR185).

Interpretation of the Female Exogamy Model

- The main text states: "All females in the first phase except one have no offspring that are buried in the grave. This could imply a practice of female exogamy..."

- o This statement is correct; however, if many women appear at the end of pedigrees, it could be due to pre-reproductive mortality rather than a female exogamy model. To exclude this possibility, it is recommended to include individuals' age at death and discuss how this affects the interpretation of the exogamy model.

We agree. Unfortunately, we only have age of death estimates for very few individuals (see Supplementary Table 2). Accordingly, we have removed the statement in question.

- Sex distribution in the grave:

- o The predominance of males could be discussed in relation to other sites, such as Flury, where a burial bias toward males has also been documented.

Corrected. The text now reads:

Genetic sexing confirmed the high predominance of males in both phases²² (Figure 2b; Supplementary Information 1), similar to reports from other Neolithic sites from present-day France and Germany^{23–26}.

- Definition of exogenous individuals:

- o The statement "This ties in with the observation that there are no exogenous individuals in the grave" needs more precise explanation.

Thank you for noticing this. The sentence should have said that there are no exogenous *males*. We have fixed this, and clarified what we mean by exogenous in the main text:

We observe no cases where males from outside the family line entered it by having children with its members. Aside from the founding generation of each pedigree, all subsequent males are genetically descended from earlier members.

- o In female exogamy contexts, many women are absent from pedigrees because their descendants are not buried at the site. However, in this case, more men fall outside the pedigree, which does not fit a classic reciprocal female exogamy model.

Good point. We looked further into the sex bias within families and for unrelated individuals. From this analysis, we found that the sex bias observed within the families (high proportion of males) is actually less pronounced for the unrelated individuals where the sex ratio is closer to 50/50. We have addressed this in the main text:

On the other hand, all females except one are related exclusively to their offspring, suggesting a high level of female exogamy. Interestingly, the sex bias observed within families is not as pronounced for unrelated individuals. This observation aligns well with

reports from similar sites with high levels of female exogamy, where unrelated individuals tend to be dominated by females, perhaps representing females that never produced offspring, or whose offspring were buried outside of the tomb²⁰.

Runs of Homozygosity (ROH) and Kinship Analysis

- It is mentioned that the children of ID222 (ID291 and ID275) exhibit long ROH segments, suggesting a possible consanguineous relationship between their mother (not sampled) and their father (ID222).

- o It would be useful to quantify the degree of relatedness required to generate this ROH percentage. It is recommended to use HapROH or similar to estimate the most probable kinship between the parents.

- o Since the parents do not share the same mitochondrial haplogroup, it would be relevant to analyze whether their relationship is through the paternal line and determine the most plausible kin relationship based on ROH and haplogroup data.

Thank you for the suggestion. To address this we have included estimated ROH profiles of 1st, 2nd and 3rd cousins from hapROH in the plot. Additionally, we have used the same bin sizes as hapROH, excluding smaller ROH segments to focus on ROHs generated because of inbreeding:

Supplementary Figure 6.5a. Runs of homozygosity (RoH), Phase 1 . Sum of RoH lengths coloured by segment size per individual. Plot is stratified by burial phase and family line. 'Expected ROH' represents estimated ROH profiles of 1st, 2nd and 3rd cousins from hapROH².

From these updated plots it is evident that the parents of BUR291 and BUR275 were most likely first cousins. We have addressed this in the main text which now reads:

The two sequenced sons of BUR222, BUR291 and BUR275 both show long runs of homozygosity with similar length profiles as expected from parents that are first cousins (Supplementary Figure 6.5a), suggesting that BUR222 and the unsampled mother were 3rd degree relatives.

Discussion

o The following statement is too ambitious for an analysis based on a single site and should be moderated: "The data from the Bury grave is the support that was lacking until now to propose a credible hypothesis regarding the population dynamics at the end of the fourth and third millennia BCE, including the 'Neolithic Decline'."

o It is recommended to rephrase this statement to reflect that the study contributes to the debate but does not constitute definitive proof regarding population dynamics in this period.

Corrected. The rephrased sentence now reads:

The data from the Bury grave provides important new evidence regarding both the 'Neolithic Decline' and the general population dynamics in the fourth and third millennium BCE.

o The sentence "Given that in Bury a temporal hiatus of 100-200 years separates the two phases of use, and supporting genetic evidence for the discontinuity of a single population, we can tie this into a wider European picture in order to trace potential signals of demographic decline in the late Neolithic" Assumes a strong population discontinuity, whereas the results do not show a clear break between the two phases.

While it is valid to discuss this issue, the text should reflect greater caution and avoid overgeneralizing a case study to a large-scale phenomenon.

It is necessary to reformulate both claims to indicate that the Bury data provide valuable evidence on potential demographic processes in the Late Neolithic but without presenting conclusions as definitive proof of a pan-European phenomenon.

Corrected. The sentence has been rephrased to:

Given the temporal hiatus that separates the two phases at Bury, the supporting genetic evidence suggesting discontinuity between the phases, and the evidence of reforestation (Supplementary information 2), we can [...]

Response to reviewer comments

Reviewer #1 (Remarks to the Author):

Thank you to the authors for addressing the queries and issues raised in my first review. I now find the manuscript to be very strong and to contribute novel insights into European Neolithic social organisation and biology.

I only have a few minor typographical and editorial suggestions rather than further queries:

l. 119–120: Add a reference or additional information regarding the project “Time of Their Lives.”

Corrected. The sentence now reads:

It was then detailed and completed with the ERC funded advanced grant ‘The Times of their Lives’.

l. 306: Add a reference to support the mention of Gurgy.

Corrected.

Throughout the text: Provide a clear definition of terms such as “full brother” (l. 309) and “extended family.”

We have clarified what we mean by the term full brother/sister in the beginning of the paragraph on kinship:

Furthermore, we use the term pedigree, along with kinship terms such as “mother”, “daughter”, “full brother” and “half sister” to describe biological relatedness, and do not imply that these terms, nor the notion of family, was understood by the population using the grave at Bury during the Neolithic times in the same way as it is understood in Western societies today.

Furthermore, the only case of after the first use of the term in the text, we have added a parenthesis explaining the phrase:

*In the first phase of the burial, the sampled material is dominated by large biological groups spanning several generations, with several cases of three or four full siblings (**i.e. siblings sharing both parents, as opposed to half siblings**) or their offspring being buried in the grave (Figure 4, Supplementary Figure 6.2).*

Lastly, the only case of the phrase ‘extended family’ in the text was edited out to address a comment from reviewer #3 on the number of unrelated individuals in Phase 2.

Reviewer #3 (Remarks to the Author):

The other reviewers and I set a significant task for the authors in terms of reorientating their paper and providing better justification for their conclusions, but the authors have risen to the challenge impressively. In my view, the new focus of their paper on site-specific population change which potentially fits into broader regional population changes, possibly linked to movement of people out of Iberia is, to my mind a more significant finding than the focus on plague and demographic decline in the previous version of the manuscript. Their conclusions are much more robustly supported by their improved and extended analyses than in the original version. However, while the manuscript is significantly improved I still think there are a few areas of interpretation that are overreaching and should be caveated or which require more clarification.

Thank you for this positive assessment, we completely agree that this new version of the paper represents a very significant improvement over its last form!

Firstly, given the original manuscript set about establishing the evidence for a 'Neolithic decline' in Central/Northwestern Europe, it is odd to me that the title and the introduction to this new paper rather seem to take the 'Neolithic decline' for granted. Particularly the sentence on Line 61 'At the end of the fourth millennium BCE a demographic decline decimated populations across northwestern Europe³'. I don't think there is evidence yet to support this assertion so unambiguously. The decline in megalithic tomb building could be down to population decline, but it could also be down to change in preferences. The pollen and summed radiocarbon dating data is regionally variable and could also be related in changes in subsistence which may or may not be related to demographic decline. All these things taken together provide good evidence that there was a decline, but I think here in several places the paper takes this as something that is known rather than something there is evidence for but is under discussion. This attitude towards the Neolithic decline is also at odds with their own argument in their 'Environmental data on the Neolithic decline' section: - if the fact of a decline is already as established for northwest Europe as the authors make out, why is further analysis required? I think the authors should be more cautious and instead build up the three factors (cessation of megalithic tomb building, pollen data, summed probability distributions of radiocarbon date) as evidence that they interpret in terms of a Neolithic decline at the beginning of 3rd Millennium BC and which they add to with their pollen data (and in particular with regards to the Bury tomb). I understand that the authors are limited in terms of words for the title but as the 'Neolithic decline' is something that does not yet have a universally accepted definition and is not yet generally accepted I think the authors should try and avoid it in their title. Maybe something like 'Population Discontinuity across the late 4th/early 3rd Millennium BC in the Paris Basin.'

Thank you for your suggestion. To address this we rewrote the beginning of the introduction removing the reference to the Neolithic Decline:

The construction of complex megalithic tombs, a hallmark feature of the Neolithic time period, ceased across continental northwestern Europe at the end of the fourth millennium BCE for unknown reasons³. Radiocarbon dates indicate a period of construction for these collective [...]

Furthermore, when the Neolithic Decline is introduced later in the introduction, we have clarified that this is one of several possible scenarios that could explain the halt in megalith building (see also our reply to your comment re. Line 78):

The large-scale decline in the construction of these megaliths towards the end of the fourth millennium BCE, could, in principle, reflect either a shift in cultural behaviour or a demographic decline. However, recent genetic results^{14,15} in combination with data from summed distributions of radiocarbon dates^{5,16} have provided increasing support for the latter hypothesis. Among the most prominent theories put forward to explain this so-called Neolithic Decline, is that [...]

For the title we prefer to keep the term ‘the Neolithic Decline’, as we believe that the coupling of our data to this hypothesis is central for the paper and essential for evaluating the current evidence supporting this idea.

Part of my criticism about the last version of the manuscript was that the authors often spoke of some of the observations they see as part of the Neolithic decline on a Europe-wide scale, but they can be regionally variable, particularly in places like Britain and Ireland. The authors have responded to these criticisms appropriately, but I feel they still haven’t gone far enough at times – there are still suggestions in places (discussed below) that they are discussing a Europe-wide phenomenon rather than one which might be more restricted (by their terms) to Central and Northwestern Europe. Despite some edits, I still don’t think it is so clear throughout that they are excluding places like Britain and Ireland or Iberia, particularly when they often talk about ‘northwestern Europe’, which by most definitions would include Britain and Ireland. Perhaps northwestern/Central continental Europe would be a better term to use.

Corrected. Throughout the text we have replaced the phrase northwestern Europe with continental northwestern Europe when drawing conclusions from our data.

In several places the authors discuss the shift in genetic ancestry of the people interred in the tombs in absolute terms, i.e. that the people from the second phase were entirely distinct from those in the first phase. While this could be true, the authors themselves admit they cannot rule out a small amount of continuity from earlier populations, therefore I think they need mediate their language accordingly in places. In addition, the authors present convincing evidence for a

substantial shift in ancestry in northwestern Europe at the beginning of 5th Millennium BC towards Iberia, which they interpret as indicating substantial northward movements of people. However, given during the first phase of activity at Bury there were already people in the Paris Basin with a similar Iberia-related ancestry profile, and alternative explanation could be that local people carrying these Iberian-associated ancestries come to predominate (for whatever reason) rather than substantial movements of population from Iberia. This is my understanding particularly from looking at Figure 3, but if I'm missing some complexity articulated elsewhere, I could be wrong!

Thank you for your comment. We have addressed this by using 'partial replacement' instead of 'replacement' when referring to the shift between the phases, throughout the text. Furthermore, we have added a new paragraph detailing alternative interpretations of the data presented in Figure 3:

The scenario outlined above represents our interpretation of this mixture modelling data. An alternative explanation could be that the few individuals who already had high proportions of 'Middle Neolithic Iberia' DNA in Phase 1 proliferated and came to dominate the population in Phase 2. However, if the descendants of a few Phase 1 individuals had come to dominate in Phase 2, we would expect to see a strong bottleneck in Phase 2, more similarity in the population size trajectories for both phases (Supplementary figure 6.17), and simulation results indicative of higher levels of population continuity. As none of these patterns are present in our data, we find this explanation less likely. Instead, we view the individuals with Iberian ancestry in Phase 1 as early arrivals originating from outside the local region. This interpretation is supported by the observation that half of the unrelated Phase 1 individuals carried high proportions of 'Middle Neolithic Iberia' ancestry, while only three individuals within the pedigrees exhibited this genetic profile.

This issue is somewhat compounded by the authors switching between discussions of changes in population and changes in ancestry. I appreciate the two are intimately connected, but to a naïve reader a shift in population (particularly when discussing population replacement) and a shift in genetic ancestry are quite different things. I'd recommend the authors discuss changes/shifts in ancestry rather than population as this is closer to what they are measuring and will align more with what people without a background in population genetics take from those terms.

We agree. We have adopted the use of 'ancestry' instead of 'populations' throughout the text.

The authors' focus on local and regional population shifts is very interesting and novel, however, as much as the authors convincingly outline the differences between the two groups interred in the tomb, to me it begs the question of why then they decided to inter their dead in a tomb alongside people who they were not directly genetically descended from and who had different funerary practices. This is especially true given the authors themselves discuss people interred

within the tomb in both phases as having been regarded as kin to some degree. Does this not also infer that the people who interred remains in the second phase in some way regarded them to be kin in some more abstract way of the people interred in the first phase? I'm presuming here that at least some of the human remains from the first phase would have been visible to the people who interred remains in the second phase. I think this is a salient point to explore given how much is often made of Neolithic 'ancestors' and their veneration within megalithic tombs. In the case of Bury it would appear in the second phase that the idea of ancestral connections were either unimportant or potentially invented. It also complicates the common archaeological assumption that continuity or revival of a practice is evidence for regional continuity of people. It may be I've missed something about the archaeology of Bury which suggests that the human remains from either phase were regarded separately by the people who interred bodies in the second phase, but if so I think the authors need to make this clearer.

This is a great point. To begin with, we would like to emphasize that the Bury grave is not the first evidence of such a reuse of a tomb by a different population. It has been suggested during the nineteenth century by archaeologists that monumental graves could have been used by different populations, according to change in the grave goods deposited near the bodies buried in the megalithic graves. Similarly, at the site Fräsegården from Sweden (Seersholm et al. 2024, Nature) there is evidence of four individuals with steppe DNA interred in the tomb, in addition to the 47 Neolithic farmers. These steppe groups should be considered an entirely different population from the preceding farming communities (outlined below in response to comment re. line 40). In fact, at the Fräsegården site we found evidence from two separate steppe groups, one dated to 2400 cal BCE and another from 2000 BCE.

To address these points, we have added a paragraph to the discussion covering this:

The clear differences between the phases at Bury suggests that a different population reused a tomb built by the people that came before them. A similar reuse was observed at the site Fräsegården from Sweden¹⁴ where there is evidence for four individuals with steppe DNA interred in the tomb, in addition to the 47 Neolithic farmers who presumably built the grave. Such observations warrant questions about how the people of the second phase of Bury regarded the original builders of the tomb from Phase 1. Could a small amount of continuity between the phases exist, that justified the appropriation of the grave in Phase 2? How did the transmission of the burial place occur? These reflections highlight broader uncertainties about how ancestry and ownership were understood in these societies.

Some specific comments related to the above:

Line 39 'At the transition between the third and the fourth millennium BCE, a significant population decline and halt in megalith building occurred across Northwestern Europe.'

There is evidence for population decline (halt in megalithic tomb building, pollen evidence for changes in forest cover) but the extent to which particularly the megalithic tomb building reflects a decline in population rather than a shift in cultural preference is debated so I don't think it can be said straightforwardly that there was a significant population decline. – '...there is evidence for a significant population decline...'

Corrected. The rephrased sentence now reads:

At the transition between the third and the fourth millennium BCE, there is evidence for a population decline concurrent with the end of megalith building across continental northwestern Europe.

Line 40 'In Scandinavia this "Neolithic decline" instigated a massive population turnover, as farming communities disappeared and were replaced by people with steppe ancestry. In Western Europe, however, farming ancestry persisted beyond the Neolithic decline, and it remains unclear whether it was accompanied by a similar demographic replacement.'

While plausible, I think it's still unclear whether the 'Neolithic decline' instigated massive population turnover so I think the authors should be more circumspect here. Steppe-related ancestry comes to predominate in places where there is more and less evidence for a 'Neolithic decline' therefore it's unlikely shifts towards steppe-related ancestry were only down to this. In addition I understand what the authors are getting at with the term 'Farming ancestry', but there is a lot of assumed knowledge wrapped up in it and it could come across as a little bizarre to any archaeologists reading the paper, after all Neolithic farming groups often carry ancestry from early hunter-gatherer groups and people carrying the steppe ancestries were probably farmers to some extent too. I think the authors could be more precise here, perhaps discussing 'ancestry from preceding farming communities'. I also think the authors could be more careful in talking about 'population turnover' and 'replacement'. I appreciate these are technical terms which reflect something specific related to ancestry change, but again going back to that point of precision, we see shifts and replacement in genetic ancestry not necessarily literally populations of people. Talking of local populations 'disappearing' also implies something rapid and immediate, when again I don't think the authors can be confident it was so total and rapid.

In Scandinavia, there is substantial evidence for an actual replacement of the local population as the first people with steppe ancestry in the area (the Battle Axe culture) carry no local ancestry of the preceding farming communities. Instead the people of the Battle Axe culture carry Neolithic ancestry from eastern Europe (mainly Poland):

[Allentoft et al. 2025, Nature]: “Individuals associated with the CWC carry a mix of steppe-related and Neolithic farmer-related ancestry; we show that the latter can be modelled as deriving exclusively from a genetic cluster associated with the Late Neolithic Globular Amphora culture (GAC) (Poland_5000BP_4700BP), and that this ancestry co-occurred with steppe-related ancestry across all sampled European regions (Fig. 4a and Extended Data Fig. 6). This suggests that the spread of steppe-related ancestry was predominantly mediated through groups already admixed with GAC-related farmer groups of the eastern European plains—an observation that has major implications for understanding the emergence of the CWC.”

To avoid confusion re. the phrase ‘*farming ancestry*’ we decided to use “*ancestry associated with Neolithic farmers*” instead:

In Western Europe, however, ancestry associated with Neolithic farmers persisted beyond the Neolithic decline, and it remains unclear whether it was accompanied by a similar demographic replacement.

Line 50: ‘Our analysis revealed that the two burial phases at Bury represented distinct demographic patterns, correlated with discontinuous genetic groups.’

It is unclear to me what ‘demographic patterns’ means here. Do the authors mean the genetic relationships between individuals in the tomb or the different age-at-death profiles? I think the authors should consider rephrasing. Also again I think ‘discontinuous’ here could be misleading given there might be some small, continuity from the people in the first phase.

Good point. We have rephrased the sentence as follows:

Our analysis revealed that the two burial phases at Bury represented largely discontinuous genetic groups of a markedly different social organisation as inferred from four large pedigrees.

Line 51: ‘Furthermore, we show that the difference between the two burial phases can be linked to a northwards movement of Neolithic ancestry from the south which only spread into the Paris Basin after the Neolithic decline’

Again, I don’t think the ‘Neolithic decline’ – it’s extent, temporal and geographical distribution is well resolved so I think it would be better for the authors to stick to dates and associations with other events (decline in tomb building, pollen record) rather than relying on taking the ‘Neolithic decline’ for granted.

Thank you for the suggestion. To address this we have added the specific date, while retaining the reference to ‘the Neolithic Decline’ as we believe that this coupling adds important context for our discussion later in the paper:

We show that the difference between the two burial phases can be linked to a northwards movement of Neolithic ancestry from the south which only spread into the Paris Basin after the Neolithic decline, at around 2,900 BCE.

Line 61: 'At the end of the fourth millennium BCE a demographic decline decimated populations across northwestern Europe³'

I think this is too unequivocal given it is an interpretation which not everyone subscribes to. In order to make clear the exclusion of the Britain and Ireland from their discussion, it might be worth the authors referring to 'continental northwestern Europe' as some definitions of 'northwestern Europe' would include the North Atlantic archipelago.

Corrected. We opted for the phrase '*continental northwestern Europe*'.

Line 78: 'Among the most prominent theories is that environmental exploitation brought about by farming, such...'

There seems to me to be a step of reasoning missing here – the authors take the decline in megalithic tomb building as evidence for demographic decline directly, when this could simply be a shift in cultural behaviour. I think it's reasonable for the authors to discuss the decline in tombs in terms of demographic decline but they need to make that link and acknowledge other possible interpretations. For instance, wouldn't demographic decline with no change in cultural preference see just steadily fewer tombs being built rather than an abrupt halt?

Good point. We have rewritten this section to better reflect that the end of megalith building could be explained either by cultural shifts or a population collapse:

The large-scale decline in the construction of these megaliths towards the end of the fourth millennium BCE, could, in principle, reflect either a shift in cultural behaviour or a demographic decline. However, demographic analyses²¹, recent genetic results^{14,15} in combination with data from distributions of radiocarbon dates^{5,16} have provided increasing support for the latter hypothesis. Among the most prominent theories put forward to explain this so-called Neolithic Decline, is that [...]

Page 86: 'The presence of *Yersinia pestis*, the etiological agent of plague, in the Bronze Age is well established and the disease was, until recently, assumed to have been spread by the migration from the Pontic steppe into Europe^{18,19}'

'...in Bronze Age Eurasia' maybe

Corrected.

Page 88: 'However, the subsequent identification of earlier lineages of *Y. pestis* has demonstrated that plague was afflicting Neolithic communities already before the Yamnaya expansions¹⁶'

This is the authors' first mention of 'Yamnaya' whereas previously they had only discussed 'steppe ancestry' which could be confusing as well as potentially imprecise (i.e. Corded Ware is not Yamnaya). For consistency I think it would be better for the authors to stick to 'steppe ancestries', as not all groups carrying steppe ancestries are culturally 'Yamnaya'.

Corrected. The rephrased sentence now reads:

[...] *already before the expansion of steppe related ancestries*

Line 127: 'This imbalance concerns both adult and non-adult individuals, and is incompatible with a natural population, thus suggesting differential burial treatment between males and females at Bury: for some reason more than half of the females in the community were excluded from being buried in the grave.'

This is a very interesting results. It would be useful to get the authors views on whether they believe this has implications for interpretations of similar patterns in terms of female exogamy at other Neolithic tombs – is general differential treatment of female burials a competing hypothesis in these other cases?

In short, no we do not believe that the two hypothesis are competing. One could easily imagine a pedigree with a male sex bias without evidence of female exogamy. In such a scenario, relatively few women would be present in the pedigree, but their (male) parents and grandparents, would also be present in the grave. The inverse scenario of high female exogamy and no sex bias is also perfectly possible, here we would expect females to be common but they would only be related to their (male) offspring in the grave. It is, however, no coincidence that the two patterns are often observed concurrently, as both ultimately stem from patriarchal social structures.

Line 145:'). From our principal component analysis (PCA), we found that all individuals from Bury fell within the broader diversity of 'Neolithic farmers' (Figure 2c).

Neolithic farmers from where (broad region)?

Corrected:

[...] *of 'Neolithic farmers' from western Eurasia (Figure 2c).*

Line 173: 'This finding provides further support for considering the individuals from the two phases as distinct genetic groups.'

Is 'distinct groups' entirely supportable here given the authors later suggest they can't rule out some level of input from the first phase?

What we wanted to convey here was that the two groups are clearly different from one another. While it's perfectly possible that there was some amount of gene flow between the two, they clearly represent distinct/different populations. To clarify, we decided to use '*separate populations*' instead, but we are open to other suggestions.

Line 175: 'Having established that the two phases at Bury form distinct populations, we decided to explore whether these findings are compatible with either genetic continuity or discontinuity between the two phases (see Methods)'

Again, can the authors say confidently that these two groups were fully distinct from one another, i.e. no connection at all? If not I think the authors need to qualify this statement.

Corrected. The text now reads:

Having established that the two phases at Bury form two separate populations

Line 182: 'However, the simulation does not exclude the possibility of some amount of gene flow between the two burial phases.'

I think therefore this means the authors cannot talk about discontinuity distinct populations so robustly. Even 'largely or possibly completely discontinuous' would be more appropriate because this expresses the uncertainty.

Thank you for the suggestion. To clarify we have replaced '*discontinuous*' with '*largely discontinuous*' in the abstract, and '*discontinuity*' with '*a large amount of discontinuity*' in the discussion.

Line 200: 'At some point after 2,900 BC, a final northwards push of the Iberian ancestry partially replaces the existing population in the Paris basin resulting in the homogenous population we observe in Phase 2.'

The authors switch between talking about replacement of ancestry and replacement of populations. I appreciate that in some ways these two concepts are synonymous in population genetics, but I think 'population replacement' can be misleading and it would be more accurate for the authors to discuss 'ancestry' specifically, as that is what they are principally measuring.

Corrected:

At some point after 2,900 BC, a final northwards push of the Iberian ancestry partially replaces the existing local ancestry in the Paris basin resulting in the homogenous population we observe in Phase 2.

Line 202: 'After the end of Phase 2, around 2,500 BC, steppe ancestry first appears in the Paris Basin²⁹ (Figure 3). After a couple of hundred years this steppe ancestry will reach the Atlantic coast and mix with local populations carrying the Iberian ancestry.'

Another example of the authors shifting between 'ancestry' and population. I think for the sake of clarity it would be better if they stuck to one term. Also, the way this is written, without mentioning that these ancestries were carried by people, makes it sound like ancestry was moving independently of populations! Discussing ancestries moving also doesn't make sense when discussing mixing with local populations. I think it's fine for the authors to discuss these movements of ancestry in terms of movements of individuals/groups of people, even if the mechanism, nature, dynamics and scale of these movements are unknown.

Good point. To address this, we rewrote the sentence focusing on people, not ancestry:

After the end of Phase 2, around 2,500 BC, people with steppe ancestry first appear in the Paris Basin²⁹ where they mix with the local population to form the genetic profile typically associated with Bell Beakers (Figure 3).

Figure 3: An excellent plain, illustrative figure but doesn't it suggest there were already individuals/populations in the Paris basin with high Iberian Neolithic-derived ancestry during Bury phase 1? In this case rather than movement of population couldn't simply local groups with this ancestry somehow came to predominate either generally or in terms of deposition in tombs? Or are the individuals with predominantly Iberian ancestry from phase 1 different in some way from those in phase 2? If so, I think this needs to be made clear.

Thank you for your suggestion. To address this we have added a new paragraph to the section on the mixture modelling results in the main manuscript:

The scenario outlined above represents our interpretation of this mixture modelling data. An alternative explanation could be that the few individuals who already had high proportions of 'Middle Neolithic Iberia' DNA in Phase 1 proliferated and came to dominate the population in Phase 2. However, if the descendants of a few Phase 1 individuals had come to dominate in Phase 2, we would expect to see a strong bottleneck in Phase 2, more similarity in the population size trajectories for both phases (Supplementary figure 6.17), and simulation results indicative of higher levels of population continuity. As none of these patterns are present in our data, we find this explanation less likely. Instead, we view the individuals with Iberian ancestry in Phase 1 as early arrivals originating from outside the local region. This interpretation is supported by the observation that half of the unrelated Phase 1 individuals carried high proportions of 'Middle Neolithic Iberia' ancestry, while only three individuals within the pedigrees exhibited this genetic profile.

Line 236: 'On the other hand, all females except one are related exclusively to their offspring, suggesting a high level of female exogamy.'

Given there is clearly evidence for sex-biased selection of whose remains ended up in the tomb, couldn't it have been that simply females that were part of this family were often buried elsewhere (which could be because of exogamy, but could just be differential burial practice).

In principle yes, but that would imply that related females and new comers were treated differently. If they were treated the same, the pattern of female exogamy should be independent from the sex bias. To address this, we specified that this interpretation applies to the females buried in the grave only:

On the other hand, all females except one are related exclusively to their offspring, suggesting a high level of female exogamy, for the females buried in the tomb.

Line 238: 'Interestingly, the sex bias observed within families is not as pronounced for unrelated individuals.'

As this is referencing a point made a fair way back it might improve clarity to add '...the male sex bias...'

Good point, we used 'male sex bias' instead.

Line 239: 'This observation aligns well with reports from sites with high levels of female exogamy, where unrelated individuals tend to be dominated by females, perhaps representing females that never produced offspring, or whose offspring were buried outside of the tomb.'

Again, should this be caveated by the authors' observation that females of all ages were underrepresented, and that it might not have been simply about exogamy but where females from these families were buried?

Many collective graves were destroyed without knowledge about their content, and we cannot exclude that the rest of the "family" was buried there, or indeed with a different ritual. In some regions, other types of graves are known for this period, less monumental and consequently more difficult to detect in the field. They generally contained fewer individuals, buried or cremated (see Salanova 2019: *From graves to society. Monuments and forms of differentiation in death*).

As the preceding sentence clearly establishes that this sex bias exist, we do not believe that it is necessary to also caveat the sentence in question, but please let us know if you feel strongly about this:

Interestingly, the male sex bias observed within families is not as pronounced for unrelated individuals. This observation aligns well with reports from sites with high levels of female exogamy, where unrelated individuals tend to be dominated by females, perhaps representing females that never produced offspring, or whose offspring were buried outside of the tomb¹⁴.

Line 302: This is the only example of half-siblings in the dataset, which implies that having two reproductive partners was not the norm, or that the progeny from these unions were buried elsewhere, and thus also perhaps not socially accepted.'

It may be also worth mentioning that if this reflected misattributed paternity, they might have been assumed to have been full biological siblings.

Good point. We have addressed this in the text which now reads:

[...] or that the progeny from these unions were buried elsewhere, and thus also perhaps not socially accepted. Moreover, it is perfectly possible that this single case of half-siblings represented misattributed paternity, and thus was not known in the community. This finding is relatively rare among comparable genetic studies, but not unheard of.

Line 311: 'However, the higher fraction of totally unrelated individuals could speak to the monument following a different kind of organisation not restricted to a few extended families.'

My understanding is that there were unrelated individuals in the first phase too and so I'm not sure even in this phase deposition could be described as restricted to a few extended families. This would instead suggest that this is a shift in degree to which non-biological relatives were included.

Yes, there were some unrelated individuals in Phase 1, but only 16. Seven of these sixteen individuals were women, and as such, they could have been part of the family through offspring not buried in the grave. Furthermore, given the incomplete sampling of the grave in this study, it is very likely that many of the males could also be added to the pedigree if more 'missing link' individuals were sequenced.

Still, we acknowledge that the current data does not provide strong enough evidence to conclude that Phase 1 consisted exclusively of 'a few extended families'. Accordingly, we rephrased the sentence as follows:

However, the higher fraction of totally unrelated individuals could speak to the monument following a different kind of organisation with a much higher degree of non-biological relatives included in the grave.

Line 378: 'However, the grave only represents a subset of the population, and it is perfectly possible that a severe plague epidemic would leave very little evidence behind, if the entire population perished without being buried or if they were buried differently, presumably elsewhere, as were the case for many plague victims during the second pandemic^{37,38}'

I think the authors should consider whether this passage is necessary – it seems to me to be an argument from an absence of evidence.

We think that it is important to note that our data does not necessarily contradict the hypothesis of severe plague outbreak during the Neolithic decline. We have attempted to write a balanced discussion presenting both arguments for and against the presence of a potential outbreak after Phase 1, allowing the reader to draw their own conclusions. With the exclusion of the statement in question, only arguments for one side of the discussion are outlined. Accordingly, we have decided to keep the sentence in the main text for now, but we are of course open to suggestions on how to better represent this.

Line 386: 'We found evidence of forest regeneration in the Paris Basin from 2,900 to around 2,600 BCE, which is typically linked to a decrease in human activity (Supplementary Information 2).'

Is this a little late to be associated with Phase 1 at Bury?

Good point. This question prompted us to notice a mistake in the text, the correct time window is actually 2,900-2,500, which obviously makes your comment even more relevant.

The end point of 2,500 is clearly late, but it is important to note that this analysis was carried out in windows of relatively large sizes because of the low resolution and fragmentary pollen records from this region. In the paper by David et al. (2012) the Neolithic of the Paris Basin is divided into seven time windows: 5100-4600 cal BC (the early Neolithic), 4600-4200 cal BC (middle Neolithic 1), 4200-3750 cal BC (middle Neolithic 2), 3750-3400 cal BC (middle Neolithic 3), 3400-2900 cal BC (late Neolithic), 2900-2500 cal BC (final Neolithic 1) and 2500-2200 cal BC (final Neolithic 2). Of these windows the 'final Neolithic 1' completely encompasses the transition from Phase 1 to Phase 2.

To address this, we have clarified the pollen data from the Paris Basin is based on relatively large temporal windows:

Of the seven Neolithic temporal windows analysed by David et al, the interval from 2,900 to 2,500 BCE shows evidence of forest regeneration which is typically linked to a decrease in human activity (Supplementary Information 2)³⁹.

Also, does this imply that the population in Phase 2 also were perhaps not doing so well if forests continued to regenerate? If the authors theory of the movement in of a completely new population are correct than this would imply the demographics of all populations are relatively volatile.

Good point, but as outlined above the resolution of the pollen data is unfortunately not sufficiently high to assess such fine scale patterns.

Line 402: 'Since the beginning of the twentieth century, many authors have argued for an influx from Iberia to Northwestern Europe according to the Bell Beaker pottery diffusion during the

third millennium BCE (see Salanova 2000⁴³, fig. 3). It now clearly appears that this influx precedes the Bell Beaker phenomenon⁴⁴: our finding demonstrates that the builders and first users of the tomb were largely replaced by a Neolithic population coming from southern France and Iberia as early as the beginning of the third millennium BCE.'

As discussed above, can the authors be sure that this definitely involved people moving out of Iberia, rather than being derived from standing variation in the Paris basin or movement from somewhere more proximate (e.g. southern France)?

The genetic source group used for our mixture modelling analysis consists of four individuals (I8134, TOR6, LugarCanto41, I0406) from Spain and Portugal dated to between 5030 and 3750 cal BCE. Given these dates and locations, the genetic signature clearly originated in Iberia. However, the northwards movement of this ancestry was a gradual process, and it is most likely that the final push into the Paris Basin came from the immediate vicinity, probably just south of the Paris Basin. As the reviewer correctly points out, there is already some of this Iberian ancestry in Phase 1 of Bury, albeit in relatively few individuals. If the descendants of these individuals had come to dominate in Phase 2, we would expect to see a strong bottleneck in the IBDne plot for Phase 2 and more similarity in the population size trajectories for both phases, which is not the case.

The mention of the Bell Beaker phenomenon is confusing here – are the authors arguing that the subsequent spread of the Bell Beaker phenomenon from Iberia may have been because of contacts established by these early population movements? Do we see any cultural changes in the early 3rd Millennium BC which might signal reorientation of connections to southern France/Iberia? This feels underexplored archaeologically.

We suggest that a link to Iberia existed before the spread of the Bell Beaker complex. During the third millennium BCE, movements from the South to the Paris Basin are well-known by important circulations of products (flint blades, copper objects) and craft specialists. We have clarified this in the main text which now reads:

Our data demonstrate an influx of genetic ancestry from Iberia before the Bell Beaker phenomenon⁴⁴ [...]

Line 409: 'population dynamics in the Paris Basin is characterised by the replacement with another group of Neolithic people, who persisted for around 500 years until the arrival of steppe-related populations'

Again, 'replacement' here seems to imply that one group was totally replaced by another, which may have been true but I don't think the authors can say with total certainty.

We agree. To correct this we have added a parentheses with 'or partial replacement':

[...] *is characterised by the replacement (or partial replacement) with another group of Neolithic people [...]*

Line 416: 'Tied in with our findings on the replacement of the population between Phase 1 and Phase 2, this suggests that newcomers did not immediately reuse the tomb'

Same as above.

Corrected. We decided to use '*the full or partial replacement*' instead.

Line 427: 'Phase 2, on the other hand, is distinguished by its longer duration, dynasty-like family structure and high number of unrelated individuals which is not found at any other Neolithic sites from the area'

Earlier the authors refer to a hereditary family structure rather than 'dynastic'. I think hereditary is more appropriate here as dynastic reads as 'rulers' or 'elites' which I don't think the authors are arguing for (or at least haven't presented the evidence for) here. Also, what number of Neolithic monuments of the area have had human remains extensively sequenced? Is it all that meaningful that this is different than the couple of other examples that have been done? There could be local variation.

Good point. We replaced '*dynastic*' with '*hereditary*'.

Regarding the comparison of Bury Phase 2 with other sites from the area, we agree a comparison is not particularly meaningful here. Accordingly we have reworded the sentence which now reads:

Phase 2, on the other hand, stands out with its longer duration, hereditary family structure and high number of unrelated individuals.

Line 430: 'Given the temporal hiatus that separates the two phases at Bury, the supporting genetic evidence suggesting discontinuity between the phases, and the evidence of reforestation (Supplementary information 2), we can tie this into a wider European picture in order to trace potential signals of demographic decline in the late Neolithic.'

'Evidence suggesting 'substantial' discontinuity...'

Corrected. We opted for '*a large amount of discontinuity*'.

Line 434: 'There is also evidence from megaliths in Northern Germany that ended around 3,100-3,000 BCE, followed by decreasing open land^{13,21,49,50}

'...megalithic tombs...'

Corrected.

Line 437: 'When we add the anomalies in the demographic structure of the buried population in Phase 1 of Bury as well as other megaliths from continental Europe, there are further indications of demographic change (Supplementary Information 1). It supports a more widespread phenomenon covering most of central and Northwest Europe.'

'...as well as other megaliths from continental Europe...' - Does this include megalithic tombs outside of North-Central continental Europe. If not I think the authors should be more specific and precise about where they are referring to given the regional evidence for 'decline' is patchy.

Corrected:

*When we add the anomalies in the demographic structure of the buried population in Phase 1 of Bury as well as other megaliths from **northern France and Germany** (Supplementary Information 1) there are further indications of demographic change. It supports a more widespread phenomenon covering most of **continental northwestern Europe**.*

Line 443: 'After a couple of centuries of interruption, the use of the Bury grave during the whole third millennium is different, both in ritual and population structure, as in most of Northwest Europe.'

Do the authors discuss the differences in ritual beyond the relationships between the burials and the demographic features. For instance, it looks from Figure 1 that there were differences in burial position between the phases (flexed versus extended) but I don't think this is discussed in the main text (apologies if I've missed it).

No you are right, the previous version of the manuscript did not include any statements on the burial positions. We've rectified this by adding two short statements in the introductory paragraph on the Bury grave with references to the original work where details on body positions can be found:

The first phase was used over a relatively short period at the end of the fourth millennium (around 3,200-3,100 BCE)², and represents burials in extended body positions oriented by the main axis of the grave¹ (Figure 1b). Phase 2 on the other hand covered several centuries over the third millennium until 2,470 BCE, and is characterised by flexed body positions with no preferred orientation¹ (Figure 1c).

Line 452: 'We may thus consider the possibility that both the Iberian northward migration, and the expansion from the steppe were related responses to the Neolithic decline in temperate Europe.'

This could do with a bit more elaboration – what is it exactly about the demographic declines in these regions which may have provoked the expansion of people out of Iberia and the steppe? I think the authors need to lay out their reasoning here.

Good point. We have addressed this by expanding the statement:

We may thus consider the possibility that both the Iberian northward migration and the expansion from the steppe were related responses to the Neolithic decline, as widespread demographic contraction would have created a vacuum that neighbouring groups could expand into.

Supplementary Figure 6.17.

This is a fascinating figure. Firstly, did the authors include/could this be at all affected by the individuals from the first phase who show elevated levels of consanguinity?

Good point, to address this we did a couple of comparisons: The original version of Supplementary Figure 6.17 was generated using input data of unrelated individuals from Bury with relatively high coverage ($>0.1x$). We defined related individuals as pairs of individuals sharing more than 800cM DNA as estimated from IBDseq (equivalent to 2nd degree relatives). To test the sensitivity of IBDne on a data set with more relatives we carried out the same analysis using a relatedness cutoff of 2000cM which includes 1st degree relatives in the input data as well. Furthermore, to test the effect of including highly inbred individuals in the data set, such as BUR291 and BUR275, we also ran the tool on input data with individuals with high ROH values ($>50cM$) excluded:

As depicted in the plot above, the overall patterns are relatively robust to changes in the input data. The most significant difference between the four approaches is that the second local peak

in the effective population size estimate of Phase 2 disappears with the most conservative filtering strategy (800cM cutoff + remove high ROH). Still, there are trends that are consistent in all versions of the plot, namely (1) the larger population size in phase 2 than phase 1 at the point of sampling, (2) the very recent decrease in population size for Phase 1 and (3) the generally more stable population in Phase 2. Yet, these results are biased by the different social organisation between the phases. Since Phase 1 consists of a single large family with few unrelated individuals, the effective size of this population would naturally be very small. For Phase 2, the high number of unrelated individuals would drive the size estimate upwards.

I appreciate the authors use the figure effectively to make a specific point about the people in the two phases of Bury being derived from divergent populations but I'm curious whether the authors could make more of the potential dynamics on show here. For instance, I'm surprised the authors do not discuss the potential support it provides for demographic decline in the population represented in Phase 1, given addressing a comment about evidence for demographic decline from one of the other reviewers was the reason for generating the figure in the first place. Or do the authors think it's too easy to overinterpret here?

Thank you for your comment. As you suggest, we do think that it is very easy to overinterpret these figures. However, with the relatively robust results provided from different filtering strategies on the input data presented above, we feel a little more confident that some informative patterns can be extracted from this analysis (see below).

There is some evidence from the trend (albeit with large confidence intervals) that the population in Phase 2 was also somewhat declining by the time they were buried at Bury after a period of relative growth coinciding with the decline in the phase 1 population.

Given the large confidence intervals of the size of the Phase 2 population, the data might as well represent a relatively stable population. Hence, we are not confident drawing any major conclusions from the shape of the Phase 2 curve.

Alternatively, could this apparent 'growth' be down to admixture with the population derived from phase 1?

In principle yes, admixture with an unrelated group would increase the effective population size. However, as outlined above we don't believe the Phase 2 data is strong enough to suggest such a trend.

Does this suggest a volatility in populations of northern/central continental Europe perhaps partly related to 'boom and bust' agriculture? Does the decline (or plausible lack thereof depending on how you interpret the confidence intervals) of the Phase 2 population have implications for understanding the spread of people with steppe-related ancestry (i.e. if

population decline was not as acute, does this suggest a different specific process of ancestry change with steppe-related ancestry)? What might be the nature of the population decline in Phase 2 given the people from this phase do not show a catastrophic mortality profile?

We think that it is safe to say that Phase 1 has some patterns of recent population contraction (IBDne results and mortality profile), while Phase 2 does not. Since we don't have data from the very end of the Phase 2 population, or during the arrival of people with steppe DNA, it is difficult to assess exactly what happened. Based on our data the Phase 2 population seems to have been thriving. However, if severe and swift catastrophic events (violence or disease) led to the demise of the Phase 2 population, it is very likely that the last generations would not have been buried in the grave.

It may be I'm simply reading too much into this figure, but if I am, other people will do so it might be worth anticipating this with a bit of text explaining why it can't be taken too literally beyond gauging divergent population histories. Also it would also be useful if the authors could briefly state in the caption how they extrapolated census population size from effective population size here.

We agree. To address this we have summarised the main points outlined above in a new supplementary section:

Supplementary Note 6.4 - Effective population size estimates

To assess if the populations at the two phases followed similar population size trajectories, we estimated effective population sizes projected back in time using the program IBDNe (v. 23Apr20.ae9)¹. As input, we used IBD segments estimated with IBDseq⁶⁶ (see methods), from unrelated individuals stratified by phase. We ran IBDNe on this data using 12 threads and default parameters, and plotted the resulting population size estimates in R, converting generations ago (GEN) to absolute age using 28 years per generation and average ages of Phase 1 and 2 of 3250 and 2650 cal BCE, respectively.

As depicted in Supplementary Figure 6.17, the populations from each phase follow separate population size trajectories, strengthening our hypothesis of two largely distinct populations at Bury. Furthermore, while the uncertainties are large for the majority of the estimates, three main observations are well supported in the data: (1) the larger population size in phase 2 than phase 1 at the point of sampling, (2) the very recent decrease in population size for Phase 1 and (3) the generally more stable population in Phase 2. Yet, these results are biased by the different social organisation between the phases. Since Phase 1 consists of a single large family with few unrelated individuals, the effective size of this population would naturally be very small. For Phase 2, the high number of unrelated individuals would drive the size estimate upwards.

In the main text, we briefly describe these findings and reference the new supplementary note in the end of the paragraph on IBD sharing which now reads:

We also estimated the effective population sizes for each phase projected back in time using this IBD data (Supplementary Figure 6.17 and Supplementary Note 6.4). From this analysis, we found that the two phases followed separate population size trajectories, and that the effective population size of Phase 1 was markedly smaller than that of Phase 2 at the point of sampling. Importantly, these estimates also revealed a very recent population size contraction in Phase 1. Taken together, these IBD-based findings provide further support for considering the individuals from the two phases as separate genetic groups.

Reviewer #4 (Remarks to the Author):

I would like to thank the authors for having redefined the scope of the manuscript, providing a valuable discussion on population decline at the end of the Neolithic without making major claims based on a single case study. It is also evident that a considerable effort has been made to improve the manuscript and to address the previous suggestions. I sincerely believe that the manuscript has improved in both quality and scientific discussion.

The mixture modelling results strongly support the arguments about this demographic decline and contribute significantly to the broader archaeological debate. In particular, they demonstrate a northwards spread of Neolithic Iberians, first moving into the Paris Basin after/during the Neolithic decline. However, I consider that some additional aspects deserve further discussion in relation to this topic:

- This Neolithic Iberian source might not have originated directly from Iberia but rather from the British Isles and Ireland, where this type of ancestry has already been traced in the Middle Neolithic (Olalde 2018 Nature, Brace 2019 Nat Ecol Evol, Cassidy 2020 Nature). Please integrate the chronological evidence suggesting the presence of Iberian ancestry in the British Isles and Ireland into your discussion of the ~2900 BCE arrival. I also suggest that, if your analyses corroborate similar ancestries, these regions should be included in your Figure 3.

Good point. We did in fact explore this previously. With the 'proximal' mixture modelling sources from Figure 3, Neolithic individuals from the British Isles are modelled as close to 100% of the 'Early Neolithic France' ancestry. This indicates that the Neolithic ancestry prevalent in Western Europe before the spread of the 'Iberian' ancestry, was also spread to the British Isles. To confirm that the direction of this spread was not the other way around (e.g. from the British Isles and southwards), we also added a group of Neolithic farmers from the British Isles as source in another round of mixture modelling. As expected, in this modelling run all Neolithic individuals from the British Isles were modelled as close to 100% 'British Neolithic' ancestry. For individuals from France and Iberia on the other hand, the addition of this additional source group did not change the results markedly. Based on this, we conclude that the Neolithic ancestry in the British Isles arrived from the south prior to the spread of the Iberian ancestry.

To illustrate this finding in Supplementary Figure 6.15, we have expanded the map slightly northwards. This new version of the figure includes a large fraction of the genomes from the British Isles, and it should now be clear that these individuals are comprised of almost 100% of the 'Early Neolithic France' ancestry:

We have addressed this in the end of the first paragraph of the discussion:

Importantly, this middle Neolithic spread of Iberian ancestry does not explain the previously established link between Early Neolithic farmers from Iberia and the British Isles⁴⁷, as individuals from modern day Britain and Ireland are modelled almost exclusively as ‘Early Neolithic France’ (Supplementary Figure 6.15). Instead, the arrival of the first farmers to the British Isles may reflect an earlier migration from mainland Europe.

- Please check the subclade distributions of Y-chromosome haplogroup I2a1 in Ireland, Britain and Europe as presented in Cassidy’s 2020. This could also provide clues about the origins/contact populations of phase 1. A similar exercise could be carried out for Y-chromosome haplogroup H2a1 as described in Rohlach et al. 2021 Scientific Reports.

To address this, we plotted the distribution of chromosome Y haplogroups among our samples in a barplot (see below). We found the same three subclades as reported by Cassidy et al. (I2a1a1, I2a1a2, and I2a1b1) in relatively high numbers in our samples. However, in contrast to the results from Neolithic Ireland, we also found haplogroup H2a1 to be dominant in Phase 1. In agreement with the main results of our paper, we find a strikingly different pattern between the two phases: While Phase 1 is diverse and dominated by three different haplogroups (H2a1, I2a1a1, and I2a1a2), Phase 2 is very homogenous, consisting almost exclusively of individuals with haplogroup I2a1a1.

Supplementary Figure 6.19. Distribution of chromosome Y haplogroups within Bury. Haplogroups were called using the ISOGG database. Variants of haplogroup I upstream of the main three “I” haplogroups were not coloured and were classified as ‘NA’.

To contextualise these results in a broader Western European perspective, we ran the chrY classification pipeline used for the Bury samples on all publicly available genomes in our panel. We visualised the distribution of the two main haplogroups from Bury (I and H), together with haplogroup R associated with the spread of Steppe related ancestry across Europe (see below). From this plot, the rarity of haplogroup H and the westwards spread of haplogroup R after ~2,300 BC is very clear. The distribution of the three major subclades within haplogroup I form a less clear pattern. Generally, we find that all three subclades are widely distributed across Western Europe in both the time window of Phase 1 and in that of Phase 2. Furthermore, as indicated in the plot, there is no evidence for a single chromosome Y haplogroup associated with the northward spread of Iberian ancestry discussed in the main manuscript.

Supplementary Figure 6.20. Map of Y chromosome haplogroups distribution in Western Europe. Haplogroups were called using the ISOGG database. Only individuals with chromosome Y haplogroups within H, I and R are shown.

We have summarised the results of these analyses in Supplementary Note 6.3:

Supplementary Note 6.3 - Chromosome Y diversity

We investigated the diversity within the Y chromosome of all human samples analysed in this study by classifying haplogroups using the ISOGG database (see Seersholm et al. 2024, Supplementary Note 2 for details). We found that the majority of samples generated for this study belonged to either major haplogroup I and H, with a single individual belonging to haplogroup G (Supplementary Figure 6.19). In agreement with the results reported in

the main paper, we find a strikingly different pattern between the two phases: While Phase 1 is diverse and dominated by three different haplogroups (H2a1, I2a1a1, and I2a1a2), Phase 2 is very homogenous, consisting almost exclusively of individuals with haplogroup I2a1a1.

To contextualise these results in a broader Western European perspective, we visualised the distribution of the two main haplogroups from Bury (I and H), together with haplogroup R associated with the spread of Steppe related ancestry across Europe (Supplementary Figure 6.20). From this plot, the rarity of haplogroup H and the westwards spread of haplogroup R after ~2,300 BC is very clear. The distribution of the three major subclades within haplogroup I form a less clear pattern. Generally, we find that all three subclades are widely distributed across Western Europe in both the time window of Phase 1 and in that of Phase 2. Furthermore, as indicated in the plot, there is no evidence for a single chromosome Y haplogroup associated with the northward spread of Iberian ancestry discussed in the main manuscript.

Lastly, we created new versions of the pedigrees from Supplementary Figures 6.2 and 6.3 to visualise chrY haplogroups in the pedigrees using the same color scheme as above:

Supplementary Figure 6.21. Pedigrees from Phase 1 colored by Y haplogroup. Circles and squares represent females and males, respectively, while triangles represent unknown sex. Inside each shape, individuals are labelled with their mitochondrial haplogroup (first line), chromosome Y haplogroup (second line) and calibrated median age (last line). Colors represent chromosome Y haplogroup. Furthermore, solid black lines between shapes

indicate well defined first degree relationships, while stippled black and grey lines specify unknown or uncertain first and second degree relationships, respectively.

Supplementary Figure 6.22. Pedigrees from phase 2 colored by Y haplogroup. Circles and squares represent females and males, respectively, while colors represent chromosome Y haplogroup. Inside each shape, individuals are labelled with their mitochondrial haplogroup (first line), chromosome Y haplogroup (second line) and calibrated median age (last line). Furthermore, solid black lines between shapes indicate well defined first degree relationships, while stippled black and grey lines specify unknown or uncertain first and second degree relationships, respectively.

- The discussion of the Neolithic decline and the Bell Beaker influx could be strengthened. Currently, only the date of Iberian influence in the region (~2900 BCE) is clearly stated. In order to better contextualize this debate, I strongly recommend presenting together the chronological ranges of the three phenomena: the Neolithic decline, the Bell Beaker influx (including its possible source regions), and the Iberian influence in the region. Depending on which dates are considered valid for the Bell Beaker expansion, the time ranges might not be so different. This does not necessarily imply that your results support an Iberian origin of the Bell Beaker, but it could highlight the possibility of two independent yet contemporaneous/almost contemporaneous phenomena. I believe this part of the discussion would benefit from a clearer chronological framework.

Good point. To address this, we have added a couple of sentences to the discussion detailing the exact timing of these phenomena:

Our data demonstrate an influx of genetic ancestry from Iberia before the Bell Beaker phenomenon⁴⁵: we show that the builders and first users of the tomb were largely replaced by a Neolithic population coming from southern France and Iberia as early as 2,900 BCE. While this date precedes the first confirmed Bell Beaker influx in the Paris Basin by several hundred years⁴⁶ it fits well with both the timing of the Neolithic Decline (3,100-2,900 BCE)³, and the first archaeological influence from the south of France in the region at around 2800 BCE¹.

Minor comments

- Lines 119–120: the “The Time of Your Life(s)” project is mentioned but not properly referenced. Please clarify what type of project this refers to.

Corrected. The sentence now reads:

It was then detailed and completed with the ERC funded advanced grant ‘The Times of their Lives’.

- Figure 2 caption: please specify what the grey symbols in the PCA represent.

Corrected. In the previous version of the figure point shape was defined by the broader geographical region of each sample. As we did not use this information in the final plot, we decided to use a fixed shape across the reference panel instead. We also updated the PCA to include our most recent reference panel which resulted in the axis being flipped:

- Line 336: remember to include the final reference for “marçais unpublished”.

Corrected

- Line 369: provide the archaeological context and geographical location of the genome RV2039.

Corrected. The updated sentence now reads:

This finding suggests that the plague form from Phase 1 of Bury is similar, and perhaps slightly older, than that of the RV2039 genome isolated from a hunter-gatherer in present-day Latvia.

- Supplementary Figure 6.14 (mixture modelling barplot): the colours used for Iran Neolithic and Early Neolithic France are too similar and difficult to distinguish.

Corrected. We used a light grey color instead for ‘Iran Neolithic’.

Response to reviewer comments

Reviewer #3 (Remarks to the Author):

I'm very grateful to the authors for preparing another comprehensive and detailed response to my comments and they've again put an impressive amount of work into their response. Generally I'm satisfied with this updated version of the manuscript.

Thanks!

'For the title we prefer to keep the term 'the Neolithic Decline', as we believe that the coupling of our data to this hypothesis is central for the paper and essential for evaluating the current evidence supporting this idea.'

I'm happy to leave this up to the Editor to decide - I don't have an issue with the use of 'Neolithic decline' in the title, only that I think it would be better if it was articulated in a way which makes it clear it is not something around which there is total consensus. e.g. 'Population Discontinuity associated with evidence for Neolithic Decline in the Paris Basin.'

Corrected. As suggested we have reworded the title to clearly articulate the uncertainties around the Neolithic Decline. The new version of the title is:

Population discontinuity in the Paris Basin linked to evidence of the Neolithic Decline

'Line 40 'In Scandinavia this "Neolithic decline" instigated a massive population turnover, as farming communities disappeared and were replaced by people with steppe ancestry. In Western Europe, however, farming ancestry persisted beyond the Neolithic decline, and it remains unclear whether it was accompanied by a similar demographic replacement.'

While plausible, I think it's still unclear whether the 'Neolithic decline' instigated massive population turnover so I think the authors should be more circumspect here. Steppe-related ancestry comes to predominate in places where there is more and less evidence for a 'Neolithic decline' therefore it's unlikely shifts towards steppe-related ancestry were only down to this. In addition I understand what the authors are getting at with the term 'Farming ancestry', but there is a lot of assumed knowledge wrapped up in it and it could come across as a little bizarre to any archaeologists reading the paper, after all Neolithic farming groups often carry ancestry from early hunter-gatherer groups and people carrying the steppe ancestries were probably farmers to some extent too. I think the authors could be more precise here, perhaps discussing 'ancestry from preceding farming communities'. I also think the authors could be more careful in talking about 'population turnover' and 'replacement'. I appreciate these are technical terms which

reflect something specific related to ancestry change, but again going back to that point of precision, we see shifts and replacement in genetic ancestry not necessarily literally populations of people. Talking of local populations 'disappearing' also implies something rapid and immediate, when again I don't think the authors can be confident it was so total and rapid.'

Corrected. We have rewritten the sentence to not use the phrase 'farming ancestry':

In Western Europe, however, ancestry associated with Neolithic farmers persisted beyond the Neolithic decline, and it remains unclear whether a similar demographic replacement occurred.

See below for a reply regarding the link between the 'Neolithic Decline' and population turnover

'In Scandinavia, there is substantial evidence for an actual replacement of the local population as the first people with steppe ancestry in the area (the Battle Axe culture) carry no local ancestry of the preceding farming communities. Instead the people of the Battle Axe culture carry Neolithic ancestry from eastern Europe (mainly Poland)''

Apologies, I wasn't clear here - I wasn't questioning the size of the population turnover, only the authors' unequivocal statement that the 'Neolithic decline' instigated this population turnover. While plausible I don't think the authors can state this with such certainty at this stage and I think they could be more cautious with their language e.g. 'In Scandinavia this 'Neolithic decline' is followed by a massive population turnover...'

Corrected. The new version of the sentence now reads:

In Scandinavia this "Neolithic decline" is followed by a massive population turnover, as farming communities disappeared and were replaced by people with steppe ancestry.

Again, these are only minor suggestions and I don't expect the manuscript would need to come to me again. I'd just like to congratulate the authors and thank them again for a an excellent and constructive review process.

Reviewer #4 (Remarks to the Author):

I consider that the authors have done an excellent job, and that after implementing the suggested changes, both the analyses and the discussion of the data are now complete and presented with a very high level of quality.

I only ask the authors to carefully review the manuscript to ensure that all points where supplementary notes, figures, or tables are relevant are clearly indicated. For example, in the caption of Figure 4C, it would be important to explicitly explain what is meant by the A, B, and C family groups and to include a reference to the corresponding supplementary material. Similarly, in lines 457–459, the authors should clarify whether the Sr data derive from a published reference (and cite it accordingly) or from supplementary material and make a call to the relevant section should be added.

Thank you for the suggestion. To clarify what is meant by the single letter abbreviations in Figure 4c, we have added the following sentence to the legend of Figure 4:

For visual clarity, each pedigree group is abbreviated with a single letter (A, B, C, etc.), where, for example, “A” in Phase 1 corresponds to “pedigree 1.A”, while “B” in Phase 2 represents “pedigree 2.B”, and so on (see Supplementary Note 2.2, Extended Data Fig. 6 and Extended Data Fig. 7, and Supplementary Table 2).

Furthermore, to address the comment regarding strontium data (which was indeed generated as part of this study) we have also referenced the raw data presented in Supplementary Table 5:

Furthermore, our strontium isotope data depicts a more stable and sedentary population in Bury than earlier Neolithic groups from the area (Supplementary Note 6 and Supplementary Table 11).

Lastly, in response to this comment and the editorial requests, we have also gone over the entire manuscript adding missing references to the SI where relevant.

I reiterate my congratulations to the authors and I do not suggest any further changes. Thank you for a very constructive review process!